# Lateral parabrachial FoxP2 neurons regulate respiratory responses to hypercapnia

Satvinder Kaur [1], Nicole Lynch[1], Yaniv Sela [1], Janayna D. Lima[1], Renner C. Thomas[1], Sathyajit S. Bandaru[1] & Clifford B. Saper [1] ✉

About half of the neurons in the parabrachial nucleus (PB) that are activated by $CO_2$ are located in the external lateral (el) subnucleus, express calcitonin gene-related peptide (CGRP), and cause forebrain arousal. We report here, in male mice, that most of the remaining $CO_2$-responsive neurons in the adjacent central lateral (PBcl) and Kölliker-Fuse (KF) PB subnuclei express the transcription factor FoxP2 and many of these neurons project to respiratory sites in the medulla. PBcl[FoxP2] neurons show increased intracellular calcium during wakefulness and REM sleep and in response to elevated $CO_2$ during NREM sleep. Photo-activation of the PBcl[FoxP2] neurons increases respiration, whereas either photo-inhibition of PBcl[FoxP2] or genetic deletion of PB/KF[FoxP2] neurons reduces the respiratory response to $CO_2$ stimulation without preventing awakening. Thus, augmenting the PBcl/KF[FoxP2] response to $CO_2$ in patients with sleep apnea in combination with inhibition of the PBel[CGRP] neurons may avoid hypoventilation and minimize EEG arousals.

Patients with obstructive sleep apnea (OSA) have recurring arousals over the course of a night due to loss of upper airway muscle tone during sleep that results in obstruction of the airway, causing a reduction (hypopnea) or cessation (apnea) of ventilation, despite persisting respiratory efforts. The interruption of ventilation causes progressive hypercapnia and hypoxia, which in turn causes increased ventilatory effort. This increased ventilatory effort is measured by increased activity (as measured by EMG) of muscles related to ventilation, including both airway dilators such as the genioglossus, and pump muscles such as the diaphragm[1–6]. Progressive hypercapnia is associated ultimately with cortical arousal that further augments respiratory efforts and re-establishes airway patency and ventilation[5,7–9]. These repeated awakenings cause sleep fragmentation, which is associated with deleterious cognitive, metabolic, and cardiovascular consequences[5,7,8,10–15]. The $CO_2$ blood levels increase rapidly during an apnea leading to increased respiratory efforts, but the fall in blood oxygen saturation typically lags behind the changes in $CO_2$ blood levels and correlates poorly with the onset of the arousal[16,17]. We have therefore developed a mouse model of the repeated periods of exposure to elevated $CO_2$ and brief arousal that mimics the events in OSA[18–20]. This model permits selective genetic targeting and manipulation of brain circuits that mediate the respiratory and EEG responses during the period of $CO_2$ elevation associated with apnea.

Our earlier work supports the hypothesis that the brain circuits that respond to hypercapnia by elevating respiratory efforts may be distinct from those that cause cortical arousal[18,19]. Specifically, we found that glutamatergic neurons (i.e., expressing the vesicular glutamate transporter, Vglut2) in the lateral PB complex showed cFos activation during elevated $CO_2$, and that deletion of the *Vglut2* gene from these neurons in the lateral PB caused both a delay in arousal and reduced ventilation to a $CO_2$ stimulus[19]. However, the two aspects of $CO_2$ responses were not correlated across our deletions, which varied in their precise location, suggesting that the ventilatory and arousal responses were due to separate populations of glutamatergic neurons in the lateral PB region[19]. We later found that a subset of these neurons in the external lateral parabrachial nucleus that express calcitonin gene-related peptide (CGRP, PBel[CGRP] neurons) play a critical role in transmitting the arousal signal to cause EEG desynchronization in response to hypercapnia, but that their inhibition had little effect on the respiratory response to $CO_2$[18]. In addition, we observed a population of non-CGRP neurons that expressed cFos during exposure to $CO_2$. These neurons were located adjacent to the PBel[CGRP] group in the

[1]Department of Neurology, Division of Sleep Medicine, and Program in Neuroscience, Beth Israel Deaconess Medical Center, Harvard Medical School, Boston, MA, USA. ✉e-mail: csaper@bidmc.harvard.edu

central lateral (PBcl) and Kölliker-Fuse (KF) parabrachial subnuclei (with a few in the lateral crescent subnucleus spanning between the PBcl and KF along the lateral margin of the PBel)[21–29]. Earlier studies showed that many neurons in this region adjacent to the PBel[CGRP] group express the transcription factor Forkhead Box protein 2 (FoxP2)[30–32] and project to respiratory premotor and motor regions in the medulla and spinal cord[33]. Therefore, we hypothesized that whereas the PB[CGRP] neurons wake up the forebrain in response to elevated $CO_2$ levels during apneas, the adjacent PBcl/KF[FoxP2] neurons may trigger activation of ventilatory effort.

To test this hypothesis, we examined the role of the PBcl/KF[FoxP2] neurons in respiratory and arousal responses to $CO_2$. We started by investigating their response (cFos expression) to $CO_2$ exposure, as well as in vivo recording of PBcl[FoxpP2] neuronal activity by using intracellular calcium ($Ca_i$) imaging during spontaneous sleep–wake and during exposure to $CO_2$. We also investigated the respiratory effects of optogenetically photo-activating PB[FoxP2] neurons during either normocapnia or hypercapnia and the effects of inactivating them optogenetically or by genetically targeted deletion of PB/KF[FoxP2] neurons on the response to a $CO_2$ stimulus. Our results support the PBcl/KF[FoxP2]

neurons as playing a key role in the respiratory response to hypercapnia.

## Results

### cFos expression in PB/KF[FoxP2] neurons during hypercapnia

We subjected mice to either 2 h of 10% $CO_2$ (10%, $CO_2$, 21% $O_2$, 69% $N_2$; $n = 5$) or normocapnic air (room air: 21% $O_2$, 79% $N_2$; $n = 3$) and then examined brain sections using immunohistochemistry for both FoxP2 and cFos, an immediate early gene, used as a functional marker for activity in neurons. Because both cFos and FoxP2 are transcription factors, they are localized to the nucleus of neurons (Fig. 1). Exposure to $CO_2$ caused a large increase in number of FoxP2 neurons that showed cFos expression in the KF (38.3 ± 5%) compared to room air (8 ± 1%; $F_{1,7} = 21.88$, $P = 0.003$; Fig. 1a–d, f and Table 1). The doubly labeled KF neurons formed a cluster located medial to the ventral spinocerebellar tract along the ventrolateral surface of the rostral PB. Slightly more caudally in the PBcl, 30 ± 4% of PBcl[FoxP2] neurons expressed cFos after $CO_2$ exposure compared to 9 ± 3% in the control mice (room air) ($F_{1,7} = 17.5$, $P = 0.006$) (Fig. 1a_2, d_3, and f and Table 1). Both the KF and the PBcl also had non-FoxP2 neurons that expressed

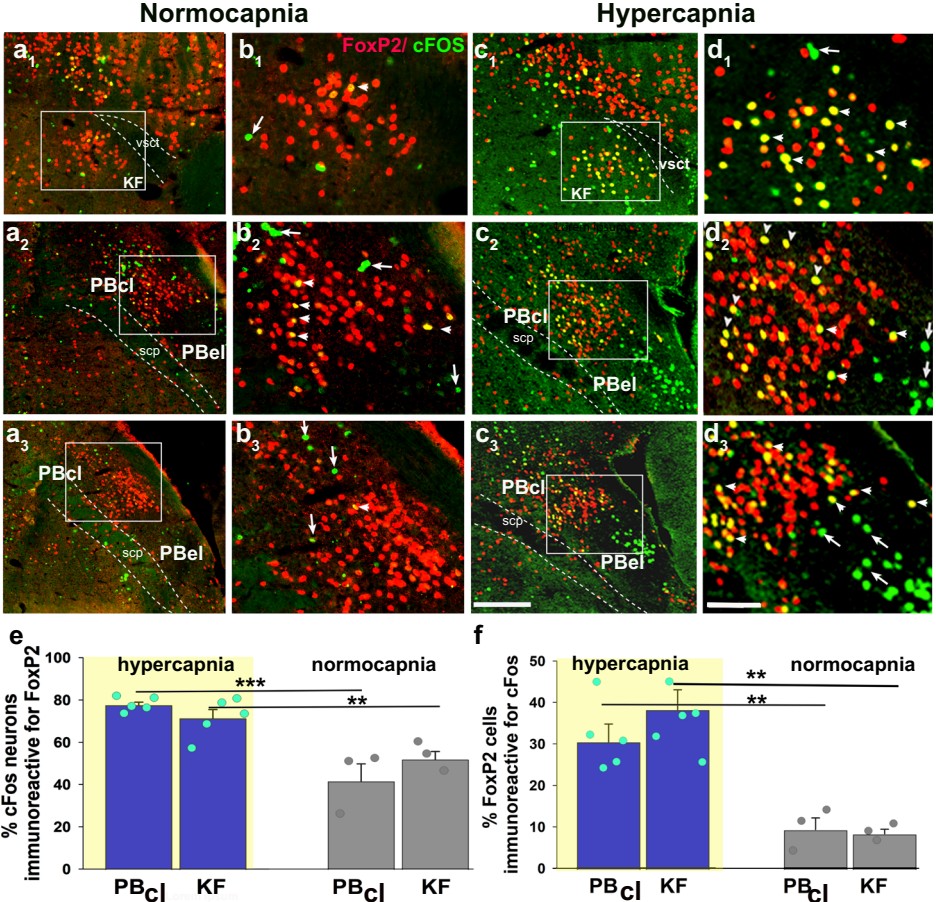

**Fig. 1 | $CO_2$ activates cFos expression in PBcl[FoxP2] and KF[FoxP2] neurons.** Photomicrographs in columns **a**–**d** represent sections at the rostral (level 1, first row), middle (level 2, second row), and more caudal (level 3, third row) portions of the PB labeled immunohistochemically for both the immediate early gene cFos (green) and FoxP2 (red), from a mouse that was exposed to 2 h of either normocapnic room air (columns **a**, **b**) or 10% $CO_2$ (columns **c**, **d**). The insets in **a**, **c** demarcate the areas that are magnified in **b**, **d**. Double-labeling (yellow nuclei) was prominent in the KF and in the central lateral FoxP2 clusters in mice exposed to $CO_2$ but not those breathing normocapnic air. The arrowheads in **b** and **d** point to doubly labeled neurons, while the arrows mark neurons that were only labeled for cFos (green), a large cluster of which represent the CGRP neurons in the PBel (c_2, c_3). The bar

graphs in **e** and **f** compare the percentage of the cFos cells (mean ± SEM) that also expressed FoxP2 (**e**), and the percentage of FoxP2 cells (mean ± SEM) that were also labeled for cFos (**f**) in the PBcl and KF areas, after exposure to 10% $CO_2$ (hypercapnia, $n = 5$) or room air (normocapnia, $n = 3$). All data points for each mouse are also shown in both **e** and **f**. The groups were analyzed using a one-way ANOVA, followed by Holms-Sidak method for multiple comparisons, where ***$P < 0.001$; **$P < 0.01$. Source data are provided as a source data file. Scale in c_3 = 200 μm; d_3 = 100 μm. KF Kölliker-Fuse PB subnucleus, PBcl central lateral PB subnucleus, PBel external lateral PB subnucleus, scp superior cerebellar peduncle, vsct ventral spinocerebellar tract.

**Table 1 | PBcl/KF neurons that express cFos during hypercapnia**

| Treatment groups | FoxP2 and non-FoxP2 cells in the PB and the KF that express cFos | | | | |
|---|---|---|---|---|---|
| | KF$^{FoxP2}$ neurons | KF$^{non-FoxP2}$ neurons | PBcl$^{FoxP2}$ neurons | PBcl$^{non-FoxP2}$ neurons | PBel$^{non-FoxP2}$ neurons |
| Hypercapnia | 38.3 ± 4.8%** | 22.6 ± 1.3% | 30.5 ± 4.3%** | 19.8 ± 0.8% | 51.1 ± 4.7%** |
| Normocapnia-control | 8.3 ± 1.1% | 21.8 ± 1.6% | 9.3 ± 2.8% | 20.9 ± 3.1% | 17.7 ± 3.2% |

Although a small percentage of FoxP2 neurons in the PBcl and KF express cFos in normocapnia (8.3 and 9.3%, respectively), the percentage more than triples after exposure to 10% $CO_2$ for 2 h (39.3% and 30.5%, respectively). There is no significant increase in cFos expression by non-FoxP2 neurons in the PBcl or KF. The groups were analyzed using a one-way ANOVA, followed by the Holms-Sidak method for multiple comparisons, where **$P < 0.01$.

cFos, but their percentages did not differ between the hypercapnia and room air conditions (Table 1). There was a dramatic increase in neurons in the PBel that expressed cFos during $CO_2$ exposure (from 17.7 ± 3.2% to 51.1 ± 4.7%; $F_{1,6} = 24.5$, $P = 0.003$). These neurons, which we had previously shown express CGRP[18,33], did not express FoxP2. Therefore, the PBcl$^{FoxP2}$ and KF$^{FoxP2}$ neurons constitute the bulk of the non-CGRP neurons in the region adjacent to the PBel$^{CGRP}$ neurons that were also responsive to $CO_2$. Interestingly, although Huang and colleagues[34] (and we) could not identify any neurons that stain immunohistochemically for CGRP to also stain for FoxP2, we found a small number of KF$^{FoxP2}$ neurons that expressed mRNA for *Calca*, the gene for CGRP (SFig. 1), by using fluorescence in situ hybridization for *Calca* in FoxP2::L10 mice (*Foxp2$^{tm1.1(cre)Rpa}$*/J crossed with R26-lox-STOP-lox-L10-GFP reporter mice which labels the ribosomes of the Cre expressing neurons with GFP). The FoxP2$^+$ CGRP neurons in the KF (mean diameter 18 ± 0.7 μm) tend to be significantly smaller ($F_{1,18} = 72.4$; $P < 0.001$) than the CGRP neurons that are not FoxP2$^+$ (26.3 ± 0.7 μm). These data correspond well with our recent spatial transcriptomic atlas of the PB region, which shows that the smaller KF neurons that express both *FoxP2* and *Calca* (cluster at1_11) have a distinct genetic expression profile from the larger KF neurons that express *Calca* but not *FoxP2* (cluster at1_10)[35]. The at1_10 neurons in the KF are, however, closely related to the PBel *Calca* neurons that do not express *FoxP2* (cluster at2_2) and may be a rostral continuation of that group.

## In vivo measurement of PBcl$^{FoxP2}$ neuronal activity by fiber photometry

To capture and analyze the pattern of activation of the PBcl$^{FoxP2}$ neurons by $CO_2$ in vivo, we injected an adeno-associated viral vector (AAV) expressing Cre-dependent GCaMP6s into the lateral PB of *Foxp2$^{tm1.1(cre)Rpa}$*/J (FoxP2-Cre) transgenic mice (developed by Dr. Richard Palmiter at the University of Washington, and donated to the Jackson Laboratory)[32,36]. We double-labeled sections through the PB from *FoxP2::L10* reporter mice (in which FoxP2 cell bodies are marked by GFP) with FoxP2 immunohistochemistry (which stained nuclei of FoxP2 expressing neurons magenta) to verify that Cre-recombinase was expressed eutopically, finding FoxP2 immunolabeling in the nucleus of 98.3 ± 1.9% of the GFP-labeled neurons in the KF and 94.4 ± 0.6% in the PBcl (SFig. 2a–d). In FoxP2-Cre mice, all of the neurons transfected by an AAV that expressed Cre-dependent GCaMP6s (which also fluoresces green) also showed FoxP2 expression in their nuclei (red in Fig. 2b). An implanted glass fiber targeting the GCaMP-expressing PBcl$^{FoxP2}$ neurons allowed us to record Ca$_i$ levels, a surrogate for neuronal activity (Fig. 2c). Because the optical fiber ended just dorsal to the PBcl, it is likely that the activity we measured was primarily from the PBcl$^{FoxP2}$ neurons, although some contribution by the KF$^{FoxP2}$ cells cannot be ruled out. The population activity of the PBcl$^{FoxP2}$ neurons was then correlated to the identified sleep–wake states, respiration, and their changes in response to repetitive $CO_2$ exposures.

In $n = 5$ mice, the optical fibers were located in close proximity to the GCaMP-expressing PBcl$^{FoxP2}$ neurons (Fig. 2b). A representative example of a fiber photometry recording across exposure to $CO_2$ is illustrated in Fig. 2d. In these mice, GCaMP fluorescence increased when the mouse transitioned from NREM sleep to either wake or REM

sleep (SFig. 3a, b). The increased activity of PBcl$^{FoxP2}$ neurons when emerging from NREM sleep may contribute to the sudden increase in respiratory effort when an animal exposed to $CO_2$ awakens from NREM sleep.

Cross-correlation analysis of the $\Delta F/F$ and the power in the EEG gamma (γ–30–100 Hz) and delta (δ: 0.5–4 Hz) frequency bands, representative of the wake and sleep states (SFig. 4a–c), respectively, suggests that $\Delta F/F$ correlates positively with the sudden increase in EEGγ (SFig. 4b), and negatively with the fall in EEG δ power at the time of arousal from NREM sleep (SFig. 4c). The peak in $\Delta F/F$ follows the shift in EEGγ power by about 1 s during spontaneous and $CO_2$-induced arousals. In response to hypercapnia, the rise in $\Delta F/F$ began a few seconds after the onset of the $CO_2$ stimulus, (SFig. 4d) similar to the rise in respiratory rate (RR) (Fig. 2h), but underwent a rapid increase beginning just before the EEG arousal and peaking 1 s afterward. Cross-correlation analysis also showed that similar to EEGγ power, the EMG signal (which correlates with the time of arousal) preceded the rise in $\Delta F/F$ by 1 s (SFig. 5c1). On the other hand, cross-correlation analysis between $\Delta F/F$ and RR showed a rise in $\Delta F/F$ that peaked about 0.5 s before the onset of increased RR (SFig 5a, c1), indicating that the activity of the PB$^{FoxP2}$ neurons preceded the change in RR. Interestingly, when this analysis included only the REM sleep episodes, $\Delta F/F$ correlated positively only with RR and not with EMG (SFig. c2). Wild-type control mice injected with AAV-DIO-GCaMP ($n = 4$), showed no GCaMP expression and therefore no change in $\Delta F/F$ under any condition (SFig. 5b, d).

All trials with 30 s exposures to 10% $CO_2$ also resulted in cortical arousal in the latter half of the trial. Therefore, to avoid the confounding effects of significantly higher GCaMP responses caused by the transition to wakefulness, we chose to analyze only the first 15 s segments of the trial, which in all cases preceded cortical arousal. Although the actual level of fluorescence during exposure to $CO_2$ varied between individual exposures (see heatmaps in Fig. 2e), when the level of $\Delta F/F$ (change in GCaMP activity/ background fluorescence) was summated over trials from 15 s prior to and then during $CO_2$ in all trials, there was a robust and statistically significant increase in fluorescence of FoxP2 neurons during exposure to $CO_2$ ($F_{1, 1682} = 62.2$; $P < 0.001$) (Fig. 2f, $\Delta F/F$ normalized to pre-$CO_2$), with the fluorescence reaching a peak increase during $CO_2$ exposure of 6.4 ± 0.9% ($F_{1, 116} = 45.95$; $P < 0.001$) compared to pre-$CO_2$. This increase in $\Delta F/F$ was also accompanied by an increase in respiratory efforts in response to $CO_2$ as measured over 59 trials from 5 mice (Fig. 2g–i). During this period we found a significant increase in RR (40 ± 1% increase for the last 5 breaths of the 15 s interval after onset of $CO_2$ exposure compared to the last 5 breaths prior to the onset of $CO_2$; $F_{1, 1662} = 5.7$, $P < 0.001$, Fig. 2h), and a smaller increase in tidal volume ($V_T$) that lagged behind the increase in RR (Fig. 2h–i) but was statistically significant (30.6 ± 0.89% increase; $F_{1, 1646} = 20.8$; $P < 0.001$) for the last five breaths of the 15 s interval (Fig. 2i). Analysis of the heat maps for individual trials shows that the average latency to when $\Delta F/F$ began to rise in response to $CO_2$ was 4 s (Fig. 2f), while the average latency of the increase in Ca$_i$ to its first peak was 9.6 ± 0.57 s, and in only 13.5% of the trials was this latency to peak less than 5 s (Fig. 2f). This observation aligns with the rise in RR starting at about 5 s and confirms the cross-

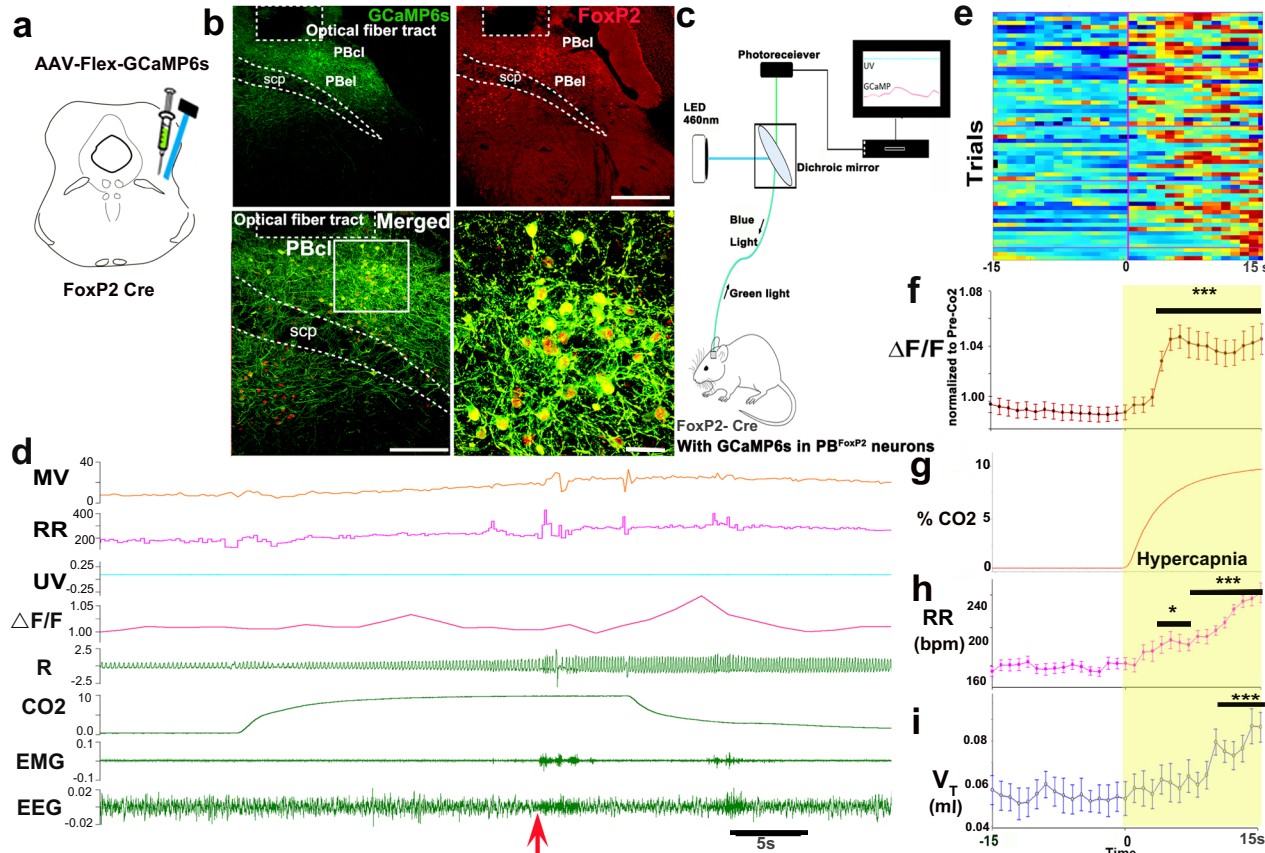

**Fig. 2 | In vivo Ca$_i$ imaging of PBcl$^{FoxP2}$ neurons by fiber photometry during exposure to CO$_2$.** Adeno-associated virus (AAV) expressing Cre-dependent GCaMP6s was injected into the lateral PB of FoxP2-Cre mice (**a**), resulting in eutopic GCaMP6s expression (green) in the PB$^{FoxP2}$ neurons (red) (**b**). The track of the implanted optical fiber just dorsal to GCaMP-expressing PBcl$^{FoxP2}$ neurons is outlined in (**b**) and allowed us to record their calcium activity as shown in the schematic in (**c**). A representative fiber photometry recording **d** shows the activity profile of the PBcl$^{FoxP2}$ neurons ($\Delta F/F$), respiration (R) with simultaneous EEG/EMG signals, and the respiratory rate (RR) and minute ventilation (MV = RR × tidal volume, $V_T$, both of which increased significantly in each CO$_2$ trial. The control UV signal (isosbestic at 405 nm) was also recorded. Note that the GCaMP signal increased slowly during CO2 exposure, but more sharply as the animal woke up

(abrupt change in EMG and EEG about 25 s after CO$_2$ onset, marked by red arrow). The $\Delta F/F$ from 59 trials from $n = 5$ mice is depicted in **e** as a heat map illustrating activity 15 s before and after the onset of CO$_2$ exposure. The graphs below show the mean ± SEM $\Delta F/F$ normalized to the pre-CO$_2$ values (**f**), %CO$_2$ in the plethysmograph (**g**), RR (**h**), and tidal volume ($V_T$, **i**) across the same trials. Two-way ANOVA compared the changes in $\Delta F/F$, RR, and $V_T$ during CO$_2$ exposure (**f**–**i**) compared to the pre-CO$_2$ baseline, followed by the Holms-Sidak method for multiple comparisons, where \*\*\*$P < 0.001$; \*\*$P < 0.01$, and \*$P < 0.05$. Scale in **b** 200 µm (lower left) and 20 µm (lower right). Source data are provided as a source data file. Scale in **b** top right = 400 µm, bottom left = 200 µm; and bottom right = 40 µm. PBcl central lateral PB subnucleus, PBel external lateral PB subnucleus, scp superior cerebellar peduncle.

correlation analysis (SFig. 5c2), where the peak in $\Delta F/F$ preceded the increase in RR by less than a second. The RR increased progressively in all trials for at least the next 10 s (Fig. 2h), and then increased abruptly just after cortical arousal, in a time course parallel to the abrupt rise in $\Delta F/F$ after cortical arousal.

## In vivo measurement of PBcl$^{FoxP2}$ neuron activity by endomicroscopy

To better understand the variability in the Ca$_i$ signal induced in PBcl$^{FoxP2}$ neurons by CO$_2$, we studied the responses of individual PBcl$^{FoxP2}$ neurons in $n = 4$ additional mice, in which we implanted a gradient-index (GRIN) lens for imaging individual neurons into or along the border of the PBcl, after injecting AAV-FLEX-GCaMP6s in the PB of the FoxP2-Cre mice (Fig. 3a). We observed the calcium activity profiles of $n = 42$ cells (8–12 cells per mouse) recorded in freely behaving mice. $\Delta F/F$ values during epochs of 10 s of continual wake, NREM or REM sleep, showed that 35/42 (83.3%) recorded neurons had higher Ca$_i$ during both REM and wake ($F_{2,102} = 5.5$; $P = 0.005$; W > N: $P = 0.014$; R > N: $P = 0.011$), while the remaining 7/42 cells had increased Ca$_i$ only during wake ($F_{2,18} = 4.88$; $P = 0.02$; W > R and N: $P = 0.018$) (SFig. 3c–h). Representative neuronal activity profiles of 8 wake-REM-

active neurons are shown in SFig. 3f–h, which suggests that these neurons show intermittent peaks that often coincide with bursts of active movements on the EMG in waking animals (SVideo 1 wake-active neurons.mp4), resulting in peaks often being synchronized across cells. However, synchronous peaks also occurred during REM sleep (SVideo 2 REM-active neurons.mp4) despite low EMG tone (SFig. 3g). We speculated that these peaks may have correlated with dream activity of movement, while actual movement was suppressed by the atonia generated during REM sleep. Thus even at the level of individual cells, PBcl$^{FoxP2}$ neurons are most active during wake and REM sleep, and particularly during active movement in wake, and may anticipate the need for increased ventilation during active movement[37,38].

We then analyzed the neuronal calcium activity of 28 neurons for the periods where the mice ($n = 3$) were exposed to CO$_2$ (Fig. 3b–e) in a plethysmograph that allowed us simultaneously to record their breathing, as well as EEG and EMG. For this analysis, we wanted to prevent confounds introduced by cortical arousal (EEG desynchronization and increase in motor activity), so we reduced the CO$_2$ concentration to 8% and only used trials when the mouse was in NREM sleep and showed no cortical arousal during 30 s of CO$_2$ exposure and in the subsequent 15 s when CO$_2$ levels in the plethysmograph were still

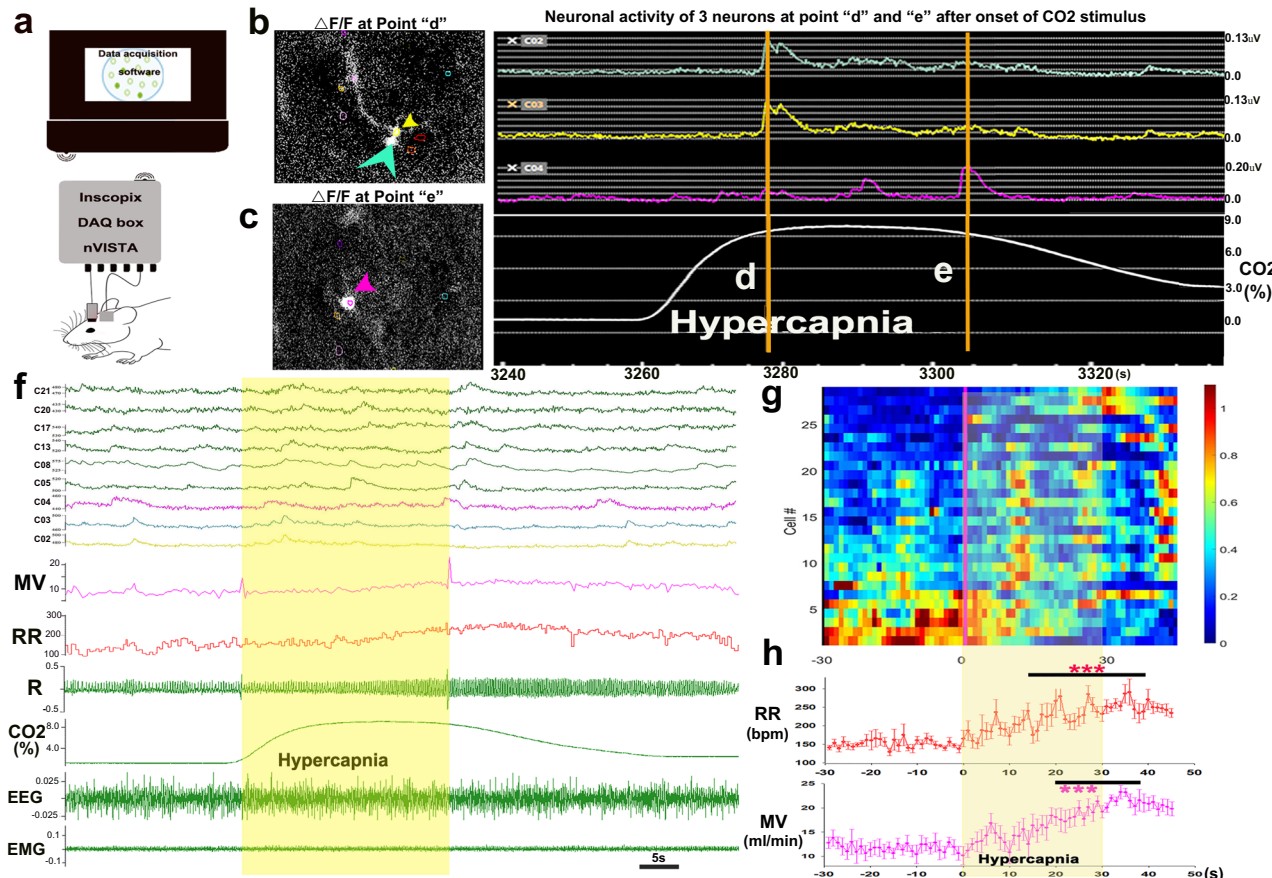

**Fig. 3 | In vivo activity of individual PB^FoxP2 neurons during CO₂ exposure.** GRIN lens was implanted above the injection sites in the PBcl of FoxP2-Cre mice injected with Cre-dependent AAV-GCaMP6s (**a**) and the calcium activity profiles ($\Delta F/F$) of individual PBcl^Foxp2 neurons in response to the $CO_2$ were acquired. Representative Ca_i activity of three neurons (**b**, **c**), of which two neurons peaked ($\Delta F/F$) at point d, early during the hypercapnia stimulus while activity of the neuron shown in (**c**) peaked in the later half of hypercapnia stimulus (shown as point e). Two cells (marked by green and yellow arrowheads whose activity profiles are also plotted in green and yellow) shown in (**b**) had fluorescence that peaked at about 17–19 s after exposure, before the maximal changes in respiration. A third cell (marked by a magenta arrow and activity profile) peaked roughly 50 s after the onset of the $CO_2$ trial (**c**), but $CO_2$ levels were still high. The $\Delta F/F$ from nine cells is also plotted during a 30 s trial of 8% $CO_2$ (with no awakening), shows an overall increase in Ca_i across

the population, with most cells showing peaks at the time of maximal respiration (R), but with substantial variability across neurons (**f**). A heat map of the mean $\Delta F/F$ over 4–7 trials (during which animals exposed to 8% $CO_2$ did not awaken) is shown for all 28 cells in (**g**) (blue to red: shows low to high $\Delta F/F$). Note that although the RR and MV summated over these trials (mean ± SEM) increased relatively smoothly (**h**), the activation of the PBcl^FoxP2 neurons occurred in waves, and of the 17 cells that showed three peaks, ten of them showed second and third peaks that were 17–19 s apart and were synchronous in time after $CO_2$ exposure, but that not every neuron participated in each wave of excitation. Data in (**h**) is acquired from $n = 3$ mice. Source data are provided as a source data file. Two-way ANOVA compared the changes in RR and MV post $CO_2$ to the Pre-$CO_2$ baseline, followed by the Holms-Sidak method for multiple comparisons, where ***$P < 0.001$.

high (see Fig. 3f, also SVideo 3 CO₂ responsive neurons.mp4). In these experiments, 8% $CO_2$ caused a 65 ± 4.4% increase in RR ($F_{1, 530} = 4.95$, $P < 0.001$, pre vs post) to 249 ± 5 bpm 20–40 s after $CO_2$, compared to 152 ± 2 bpm pre-$CO_2$. Minute ventilation also increased significantly (MV, $F_{1, 530} = 5.8$, $P < 0.001$) with a 87.5 ± 4.4% increase 20–40 s after $CO_2$ onset (22.3 ± 0.4 ml/min) compared to pre-$CO_2$ levels (12 ± 0.26 ml/min) (Fig. 3f, h). $V_T$ showed a smaller but still significant increase ($F_{1, 360} = 1.53$; $P = 0.043$) by 28 ± 5% during 20–40 s after $CO_2$ onset (0.094 ± 0.002 ml compared to pre-$CO_2$ 0.074 ± 0.0016 ml). An example of simultaneous neuronal activity profiles for 9 cells during one representative $CO_2$ trial is shown in Fig. 3f. We analyzed the activity of all $n = 28$ cells, for 30 s before and for 45 s after the onset of the $CO_2$ stimulus in 4–7 trials during which the animals remained in NREM sleep for this period of time, and plotted the normalized fluorescence ($\Delta F/F$) as a heat map (Fig. 3g). A small number of cells (4/28, 14%) were more active prior to the $CO_2$ exposure, and were less active during the exposure. We termed these $CO_2$-off cells. The remaining 24 CO2-on neurons all showed periodic increases in Ca_i after exposure to $CO_2$, with 17/24 cells showing three distinct fluorescence peaks during $CO_2$

exposure. 23/24 cells showed a first peak in activity with a mean latency of 11 ± 0.6 s after onset of the $CO_2$, 21/24 showed a second peak at a mean of 27 ± 1.6 s, and then 17/24 at 41 ± 2.8 s; 1/24 only peaked at 33 s after $CO_2$ onset (Fig. 3e). Of the 17 cells that showed all three peaks, 10 cells peaked first at 11 ± 0.21 s after $CO_2$ onset and then showed periodicity of 17–19 s for the appearance of the second and third peak. Smaller numbers of neurons (10–12) also showed peaks in activity at roughly 60 s and 75 s. None of these neurons displayed this pattern of periodic and synchronized activity during NREM sleep prior to $CO_2$ exposure. However, the periodic waves of synchronous activity during $CO_2$ exposure were similar to those we observed in individual PB^FoxP2 neurons in animals breathing room air during wake and REM sleep. These data suggest that PBcl^FoxP2 neurons may have a tendency to be activated in periodic waves that are synchronous across the population during wake and REM, but that these waves of activity may be suppressed during NREM sleep. Exposure to $CO_2$ may then remove this suppression, allowing the resumption of periodic, synchronous waves of activity with a periodicity during $CO_2$ exposure in NREM sleep of about 15 s. Whether these synchronous waves are generated by

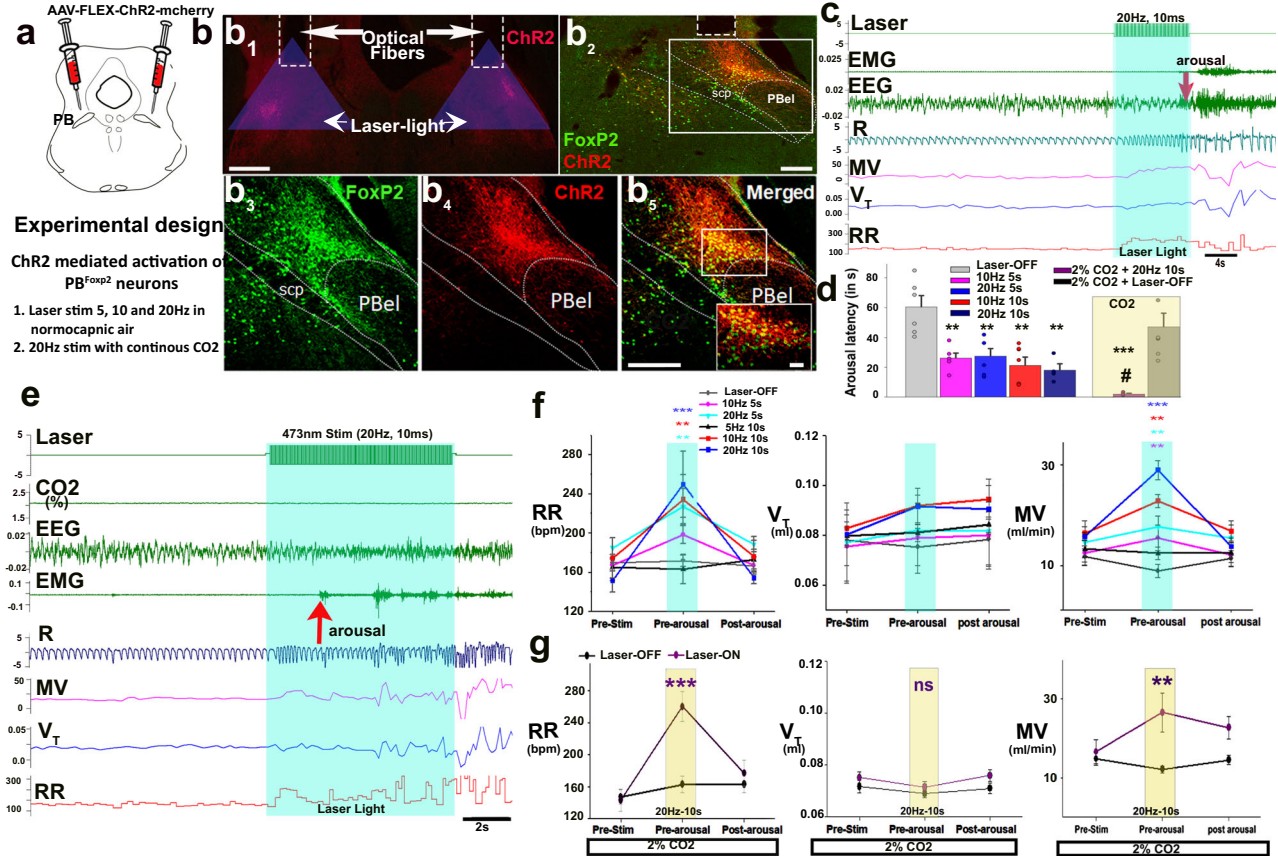

**Fig. 4 | Effect of photoactivation of PB$^{FoxP2}$ neurons on respiration during NREM sleep.** FoxP2-Cre mice were injected bilaterally targeting the PBcl with AAV-Flex-ChR2-mCherry (red **a**, **b**) and sections were immunostained (green) for FoxP2. Implanted optical fibers targeted ChR2-expressing FoxP2 neurons (b$_{1, 2}$). Panel b$_2$ shows the right side of b$_2$ at higher magnification, b$_3$–b$_5$ shows the box in b$_2$, and the inset shows the box in b$_5$, Scale in b$_1$ = 500; b$_2$–b$_5$ = 300 μm; inset = 50 μm. In **c** a representative trial of 20 Hz stimulation in normocapnic air showed gradually increasing respiration (R), with cortical arousal at 4.5 s in this trial. In trials with 5 s stimulation, the animals on average awakened around 15 s after the stimulation stopped (**d**), suggesting that the awakening was not due to the stimulation itself, but may have been elicited by subsequent respiratory efforts. Trials with stimulation for 10 s usually caused EEG arousal either just before or after the termination of stimulation. Arousal latency was decreased dramatically (by 83%) by exposing the mice to continuous 2% CO$_2$ during optostimulation (**d**, shown in a yellow rectangle).

A representative trial of stimulation at 20 Hz for 10 s with continuous 2% CO$_2$ is shown in (**e**). Graphs in **f** compare the RR, $V_T$, and MV during laser stimulation at 5 Hz, 10 Hz, or 20 Hz vs no laser (laser-OFF) for either 5 s or 10 s during normocapnia. Graphs in **g** compare the RR, $V_T$, and MV parameters in the mice subjected to laser stimulation at 20 Hz vs no laser (laser-OFF) for 10 s, in mice continuously exposed to 2% CO$_2$. Values are mean ± SEM for five breaths before the onset of stimulation, during stimulation but before cortical arousal, and in the post-stimulation period after the cortical arousal when stable breathing is attained. Two-way ANOVA was used for statistical comparison, followed by the Holm-sidak test for multiple comparisons, where **$P < 0.01$ (vs pre-stimulation/laser-OFF); ***$P < 0.001$ (vs pre-stimulation/laser-OFF), #$P < 0.05$ (vs laser-ON at normocapnia). Source data are provided as a source data file. PBel external lateral PB subnucleus, scp superior cerebellar peduncle.

recurrent activity in a local network or perhaps by a rhythmic pattern of external input, is not clear. However, there are extensive visceral sensory inputs to the lateral PB, so these synchronous waves of activity by PBcl$^{FoxP2}$ neurons could reflect the convergence of a periodic vagal input, such as waves of peristalsis from the upper gastrointestinal tract, whose swallowing activity has to be coordinated with breathing[39].

**Photo-activation of PB$^{FoxP2}$ neurons and respiration**

To test the effect of activation of PBcl$^{FoxP2}$ cells on respiration, we targeted PBcl in FoxP2-Cre mice ($n = 19$) bilaterally with injections of Cre-dependent AAV-Flex-ChR2-mCherry (Fig. 4a), and implanted them for EEG/EMG recording along with bilateral optical fibers to illuminate the ChR2-expressing FoxP2 neurons in the lateral PB (Fig. 4b). We recorded animals for sleep and breathing at baseline in normocapnic room air and during activation of ChR2 with a blue laser (473 nm) using 10 ms pulses at 5 Hz, 10 Hz, or 20 Hz, for either 5 s or 10 s every 5 min. Trials in which mice were in NREM sleep for at least 30 s before stimulation were analyzed for the effect of optostimulation on respiration. In 12 out of 19

implanted mice, the optical fibers accurately targeted the ChR2-expressing PBcl$^{FoxP2}$ neurons as shown in Fig. 4b and SFig. 6a, with the fiber tip no more than 1 mm from the center of transfection in PBcl. However, there are other neurons dorsal to the PBcl that express FoxP2, and although only the PBcl$^{FoxP2}$ and KF$^{FoxP2}$ neurons project to the ventrolateral medullary region that controls respiration (Fig. 5e), the more dorsal regions may have been stimulated. For this reason, in the optogenetic studies, we will refer to the area of photostimulation as the PB$^{FoxP2}$ neurons. Note that it is also possible that some KF$^{FoxP2}$ neurons may have been included too, but they are far enough from the optical fiber tip (SFig. 6a) that it is unlikely that they made much contribution to the results. The 12 animals with injection and optical fiber placement including the PBcl$^{FoxP2}$ neurons consistently showed respiratory responses during optostimulation; in 69 ± 9.6% of the trials, EEG arousal occurred either late in the stimulation or shortly after offset (within 10 s) while some trials (32 ± 4.8%) showed no EEG arousals (Fig. 4c, d). Interestingly, the maximal RR in the trials where the animals awakened during or immediately after stimulation was 259.9 ± 15.8 bpm; the maximal RR in the trials where the mouse failed to arouse was

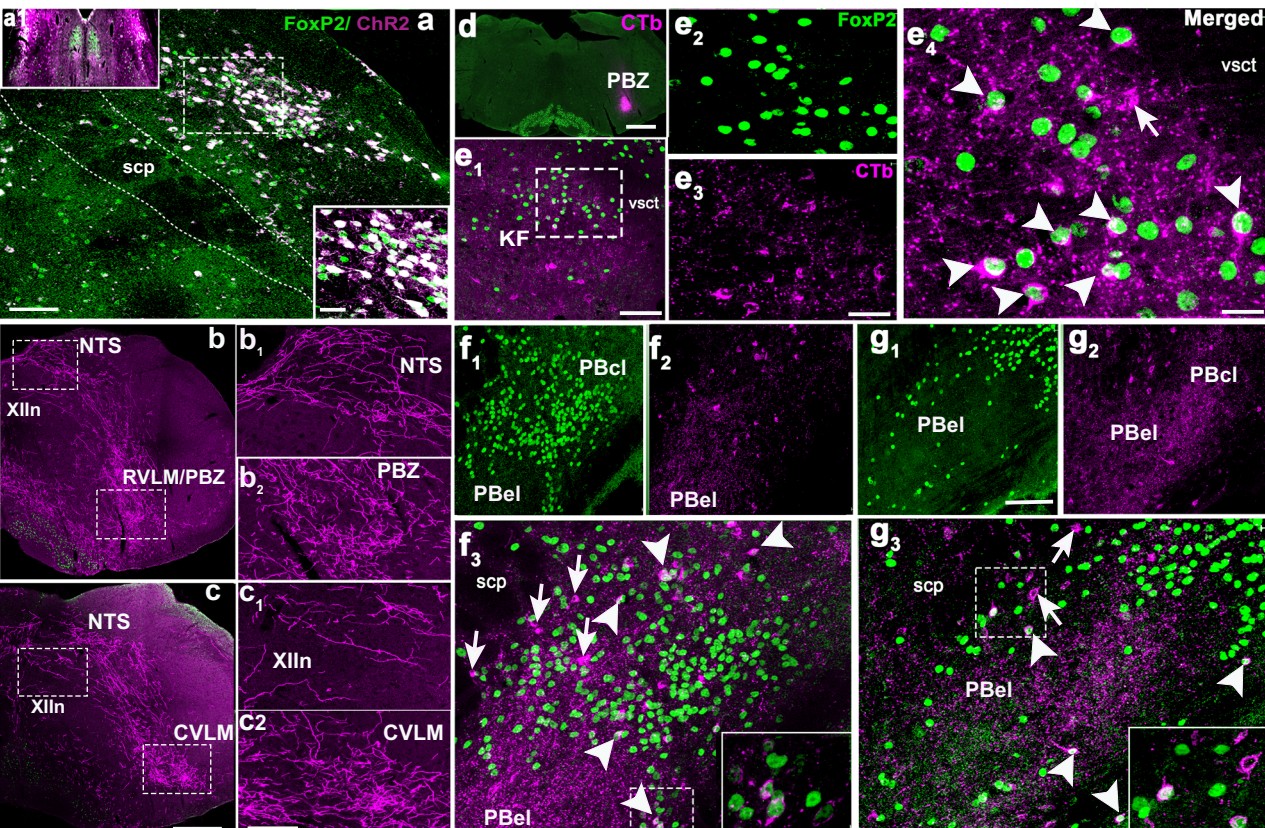

**Fig. 5 | Descending projections of the PB^FoxP2 neurons.** Bilateral injections of Cre-dependent AAV-ChR2 (magenta) into the PBcl in (**a**) FoxP2-Cre mouse (a₁) are shown with immuno-labeled for FoxP2 (green); nearly all of the ChR2-labeled cell bodies are doubly labeled (white) in the PBcl area along the dorsal margin of the PBel (which is unlabeled) (**a**, magnified view of injection site on right). The bottom right inset **a** is a magnified view of the area marked by the dashed rectangle. Descending fibers and terminals in the medulla are shown in magenta in (**b**, **c**) on the right side of the brain. The magnified views (3×) of the areas in the dashed boxes in both (**b**, **c**) are shown as b₁–c₂ to demonstrate that ChR2-mCherry labels only the fibers and does not label any cell bodies in the area. Note that inferior olive neurons (green in **b**, **d**) also express FoxP2. Scale in **a** = 100 μm (inset = 50 μm), in **c** = 500 μm and c₂ = 150 μm. Data shown in micrographs **a**–**c** were consistently observed in *n* = 6 mice. The photomicrograph in **d** shows the injection site of the retrograde tracer CTb (0.2%, magenta) in the pre-Bötzinger area (PBZ) (*n* = 3; scale = 500 μm). This injection retrogradely labeled many FoxP2 neurons in the KF (e₁–e₄) and the PBcl and lateral crescent areas that surround the PBel subnucleus (f₁–g₃). The arrowheads in e₄, f₃, and g₃ mark the cells doubly labeled for FoxP2 and CTb (white), while the arrows mark cells labeled only with CTb (magenta). The insets in f₃ and g₃ are 2× magnified views of the areas encompassed by dashed rectangles, highlighting the double-labeled cells. Scale in e₁ = 100 μm, in e₃ = 60 μm, and in e₄ = 30 μm. Scale in g₁ and g₃ = 100 μm. CVLM caudal ventrolateral medulla, KF Kölliker-Fuse PB subnucleus, NTS nucleus tractus solitarii, PBZ pre-Bötzinger area, PBcl central lateral PB subnucleus, PBel external lateral PB subnucleus, RVLM rostral ventrolateral medulla, scp superior cerebellar peduncle, vsct ventral spinocerebellar tract, XIIn hypoglossal nucleus.

200.38 ± 6.3 bpm. In four cases where fiber implants were placed either medial or lateral to the PBcl^FoxP2 neurons expressing ChR2 and in three cases with correct placements but no ChR2 transfection (SFig. 6a), neither 10 nor 20 Hz stimulation produced any effect on respiration (SFig. 6b, c). To distinguish the increment in ventilation due to waking, we measured RR and $V_T$ for five breaths pre-stimulation, five breaths just before the end of stimulation (or just before the onset of EEG arousal, whichever occurred first), and five breaths after the cortical arousal (if it occurred). Changes in ventilation with stimulation in trials where animals were sleeping before stimulation are shown in SFig. 6b, c. We did not observe any significant changes in respiration in response to stimulation either at 10 Hz or 20 Hz during the waking state when the PBcl^FoxP2 neurons are already active (SFig. 7a–f), and we could not record enough REM sleep state trials using this paradigm to reliably analyze the effects of laser stimulation. We therefore focused on a detailed analysis of the trials in which the animals were in NREM sleep for at least 30 s before the laser stimulation.

While breathing normocapnic air, 10 Hz stimulation progressively increased RR (18.5% at 5 s and 42% at 10 s stimulation, *P* = 0.013; $F_{2, 63}$ = 19.75; *P* < 0.001) and MV (17.9% at 5 s and 26% at 10 s stimulation (for 5s *P* = 0.03; for 10s *P* < 0.001; $F_{2, 63}$ = 10.43; *P* < 0.001) compared to the pre-stimulation period, but had little effect on $V_T$ ($F_{2, 63}$ = 0.35; *P* = 0.71). Stimulation at 20 Hz increased RR by 35% at 5 s vs 69% at 10 s ($F_{1, 106}$ = 122.9, *P* < 0.001) and MV by 25% at 5 s and 40% at 10 s, but again had minimal effects on $V_T$ (Fig. 4f). This pattern was consistent with the fiber photometry results, in which RR began to increase just after the calcium activity of PBcl^FoxP2 neurons, but increases in $V_T$ lagged by about 10 s. Neither sham stimulation (off-target; SFig. 6b, c) nor photo-stimulation at 5 Hz caused any significant changes in respiration (Fig. 4f).

To explore the interaction between optogenetic stimulation and activation of PB^FoxP2 neurons due to elevated $CO_2$, we repeated the 20 Hz, 10 s stimulation in mice breathing 2% $CO_2$ (Fig. 4e, representative example). First, we observed that the presence of 2% $CO_2$ throughout the experiments dramatically decreased the mean latency of awakening to the laser stimulus to 2.22 ± 0.3 s compared to 23.4 ± 2.2 s in normocapnic air ($F_{5, 29}$ = 8.2; *P* < 0.001) (Fig. 4d). Despite the relatively brief period of laser stimulation before awakening in 2% $CO_2$, we also found a large and almost immediate increase prior to awakening in RR (81%; $F_{2, 12}$ = 44.25; *P* < 0.001) and MV (38%; $F_{2, 12}$ = 8.08; *P* = 0.006), with no change in $V_T$ (Fig. 4g), a pattern similar to that found after 10 s stimulation in normocapnic air (Fig. 4f). Plots of instantaneous changes in RR and MV (SFig. 6b–e) also confirm that

during normocapnia, significant increases in RR ($F_{5, 390} = 91.85$; $P < 0.001$) and MV ($F_{5, 390} = 140.5$; $P < 0.001$; SFig. 6b, d) were observed at 4 s and 3 s, respectively, but with 2% $CO_2$, the significant rise in RR ($F_{1, 180} = 238$; $P < 0.001$) and MV ($F_{1, 180} = 112.6$; $P < 0.001$) was already present at 1 s after onset of stimulation (SFig. 6c–e). Interestingly, the EEG arousal occurred at around the same RR with $CO_2$ or optogenetic stimulation alone or with optostimulation at 2% $CO_2$ (i.e., approximately 250–260 bpm), whereas identical optostimulation in trials where the animal failed to arouse had maximal RR in the 200 bpm range. These observations raise the possibility that the arousal caused by stimulation of the PB[FoxP2] neurons may be due to the sensory feedback from the increased ventilatory effort, rather than being a primary effect of the PB[FoxP2] neurons.

Further, we used the mice from these experiments ($n = 6$) to map the medullary projections of the PB[FoxP2] neurons expressing ChR2-mCherry (Fig. 5a). All of these injection sites included the PBcl, as well as the more dorsal lateral PB[FoxP2] neurons (which are known to project extensively to the forebrain[32]), but none included the KF[FoxP2] neurons. Consistent with a previous study of the projections of PB[FoxP2] neurons, we found extensive mCherry labeling of fibers and terminals in the pre-Bötzinger complex (PBZ) and caudal ventrolateral medulla (CVLM), and more moderate numbers of labeled terminals in the nucleus of the solitary tract (NTS) and hypoglossal motor nucleus (XIIn) (Fig. 5b, c)[32]. To identify the origin of these projections, we also injected a retrograde tracer, cholera toxin subunit b (CTb) in the PBZ area ($n = 3$; Fig. 5d) and mapped the retrogradely labeled FoxP2 neurons in the PB and KF. Most CTb-labeled neurons were also immunoreactive for FoxP2 in the KF ($76.2 \pm 7.6\%$) (Fig. 5e1–e3) and in the PBcl and lateral crescent zone along the lateral edge of the PBel between the two major clusters ($70.3 \pm 3.19\%$) (Fig. 5f, g). Retrogradely labeled neurons were not seen in the more dorsal lateral PB populations expressing FoxP2. Thus the PBcl[FoxP2] and KF[FoxP2] populations project to medullary targets (Fig. 5) that are consistent with mediating $CO_2$-evoked changes in ventilation and represent a large proportion of the neurons in the PB complex that project to those targets.

### Effect of inactivation of PB[FoxP2] neurons on respiratory responses to $CO_2$

**Genetic deletion of PB[FoxP2] and KF[FoxP2] neurons.** To test the necessity of PB + KF[FoxP2] neurons in producing respiratory responses to $CO_2$ during sleep, we genetically deleted the FoxP2 neurons in the PB and the KF. We injected AAV-mCherry-DIO-DTA, which stains neurons red if they do not express Cre, but kills Cre-expressing neurons, bilaterally in the PB/KF of the FoxP2-Cre ($n = 8$) and wild-type ($n = 8$) mice (WT-DTA, Fig. 6a). The spread of AAV-DTA resulted in ablation of $91.3 \pm 1.4\%$ of the PBcl[FoxP2] and $87.6 \pm 7.9\%$ of the KF[FoxP] neurons in five mice (PB + KF[FoxP2]-DTA; Fig. 6b). Immunostaining for mCherry showed that other neuronal populations, such as the PBel neurons, which are adjacent to the FoxP2 neurons were intact (Fig. 6b).

Analysis of the respiratory response to hypercapnia in the animals with PB + KF[FoxP2] ablation compared to the WT mice also treated with the AAV-DTA showed that the loss of FoxP2 neurons in PBcl and KF had little effect on the RR increase due to $CO_2$ (Fig. 6c), but significantly reduced ($F_{1, 33} = 4.4$; $P = 0.02$; Fig. 6d) the increase in $V_T$ both prior to arousal (by 55%, $P = 0.028$) and after arousal (by 62%, $P < 0.001$). Similarly, the increase in MV was also significantly reduced ($F_{1, 33} = 10.6$; $P < 0.001$; Fig. 6e) both before (by 54.4%, $P = 0.03$) and after arousal (by 64.2%, $P < 0.001$).

**Acute inhibition of PB[FoxP2] neurons.** To further confirm the role of PB[FoxP2] neurons in the respiratory response to hypercapnia and to rule out compensation due to chronic deletion in the ablation experiment, we also acutely inhibited the PB[FoxP2] neurons during hypercapnia using optogenetics. FoxP2-Cre mice ($n = 12$) and their wild-type littermates (WT, $n = 6$) were injected in the PB with AAV-Flex-ArchT (Fig. 7a) and

implanted with EEG/EMG electrodes and bilateral optical fibers targeting the PB[FoxP2] neurons (Fig. 7c1–c5). Because of limitations of the field of illumination of the optical fibers, these experiments did not attempt to inhibit the KF[FoxP2] neurons although we cannot rule out that they might also have been inhibited. We analyzed the respiratory responses to 10% $CO_2$ given for 30 s every 300 s, with and without photo-inhibition of the PB[FoxP2] neurons using a 593 nm laser light (Fig. 7b). The laser light was on for 60 s, beginning 20 s prior and extending to 10 s after the $CO_2$ stimulus (30 s) (representative trials of $CO_2$ exposure are shown in Fig. 7d, e).

Photoinhibition of PB[FoxP2] neurons significantly reduced the increase in $V_T$ caused by $CO_2$ ($F_{2, 42} = 20.12$; $P < 0.001$) by 42% ($P = 0.022$) pre-arousal and by 34% ($P = 0.010$) post-arousal and also reduced the increase in MV ($F_{2, 42} = 7.58$; $P = 0.002$) by 39% pre-arousal and 37% post-arousal ($P = 0.007$), although the reduction at pre-arousal was not statistically significant ($P = 0.081$) (Fig. 7g, h). The photoinhibition also had no significant effects on respiration during either wake or REM sleep states.

While the animals with chronic ablation of PB + KF[FoxP2] neurons had greater reductions in $V_T$ ($F_{1, 18} = 5.5$; $P = 0.03$) and MV ($F_{1, 18} = 4.6$; $P = 0.04$) at both pre-arousal and post-arousal, compared to the animals with acute inhibition of the PB[FoxP2] neurons, the chronic ablations involved a larger portion of the PB + KF[FoxP2] population. In both groups of animals, though, the loss of activity of PB + KF[FoxP2] neurons had no effect on the RR increases induced by $CO_2$ (Fig. 7f), which presumably can be mediated by medullary respiratory neurons in the absence of PB input, and no effect on latency to arousal (Fig. 7i) which is due to PBel[CGRP] neurons.

## Discussion

Our previous work identified a population of CGRP neurons in the PBel that project to the forebrain and are required for arousal from sleep during $CO_2$ exposure. Their stimulation caused arousal with a latency of less than 2 s and inhibiting them did not affect repiratory efforts in response to hypercapnia[18]. In fact, recent studies provided evidence of their inhibitory role on respiration at least during active wake states[40,41]. Here we report an adjacent complementary population of neurons that is marked by expression of the FoxP2 transcription factor and is required for normal respiratory response to $CO_2$ during sleep, but not arousal. These PB/KF[FoxP2] neurons are located in clusters in the PBcl just dorsal to the CGRP group and in the KF subnucleus, and along a narrow bridge just lateral to the CGRP group (the lateral crescent) connecting the two major clusters. This population of PBcl + KF[FoxP2] neurons projects intensely to medullary respiratory areas. Optogenetic stimulation of the PB[FoxP2] neurons immediately increased ventilation but only caused delayed and inconsistent arousals, and while ablation or photoinhibition reduced the ventilatory increases caused by $CO_2$ it did not affect arousal. Hence, we interpret the PBcl + KF[FoxP2] neurons as primarily mediating the ventilatory response to $CO_2$.

Although optogenetically activating the PB[FoxP2] neurons eventually awakened animals in 69% of the trials, the mean latency to arousal in those trials was longer than the duration of the stimulation, indicating that arousal often occurred after the termination of the stimulus. These arousals typically occurred when the animals reached an RR similar to that at the time of arousal in an animal exposed to 10% $CO_2$. We interpret these findings as indicating that during exposure to hypercapnia, the PB[FoxP2] neurons primarily cause a rapid increase in ventilation. However, as animals increase ventilation to around 250–260 breaths per minute, the mechanosensory feedback from the increased ventilatory effort may activate the PBel[CGRP] neurons, causing EEG arousal. This forebrain arousal itself appears to further activate PBcl[FoxP2] neurons, dramatically further increasing respiratory effort. This relationship underscores the synergy between the behavioral and respiratory motor responses to ensure survival when an animal is either apneic or asphyxiated.

We used three different strategies to identify the role of the PBcl/KF[FoxP2] neurons in the ventilatory response to $CO_2$ during sleep. First,

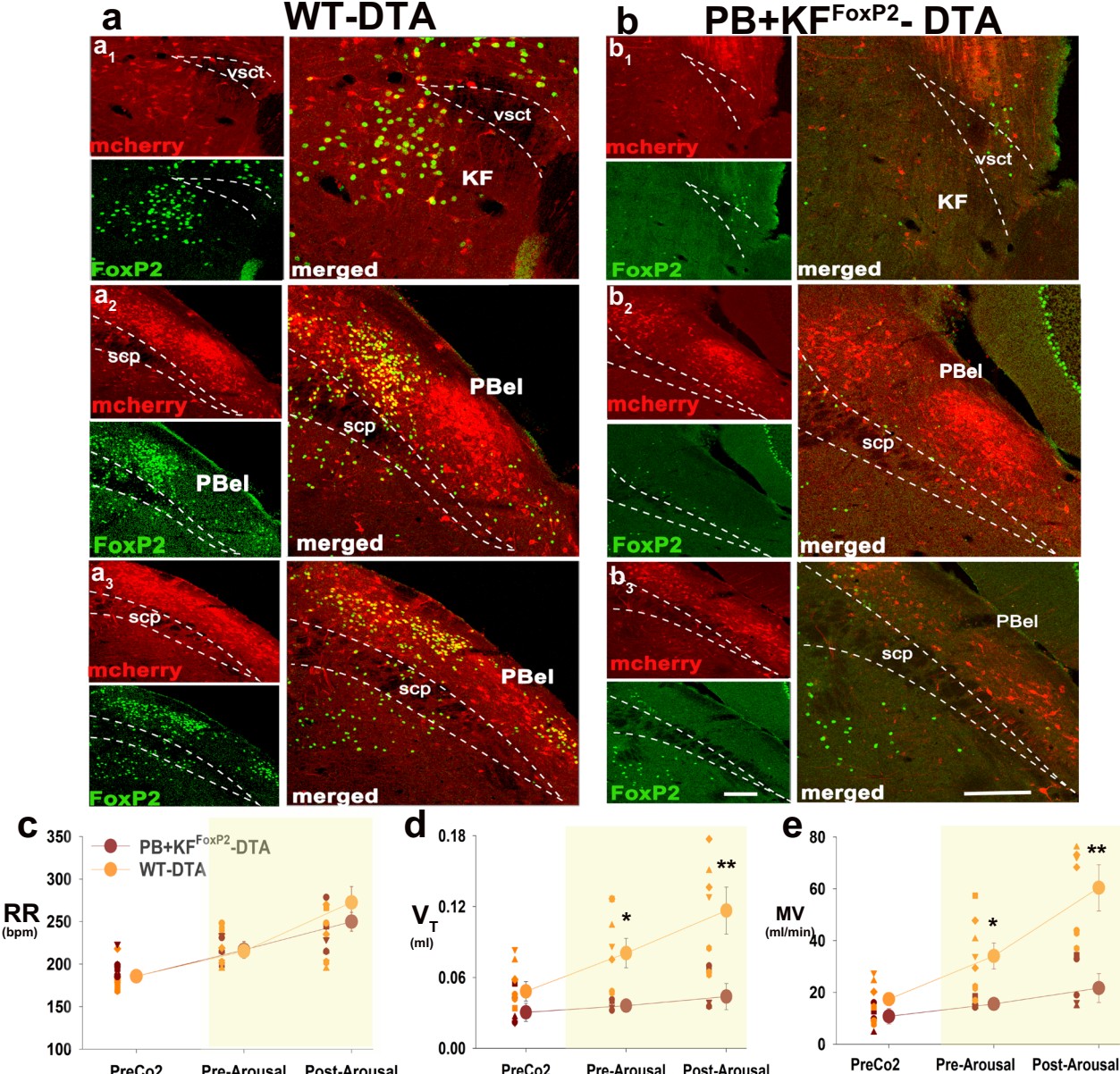

**Fig. 6 | Genetic ablation of the PB/KF$^{FoxP2}$neurons blocks hypercapnia-induced respiratory drive.** AAV-FLEX-DTA injections in PB of the FoxP2-Cre mice and wild-type littermates selectively ablated the FoxP2 neurons in the PBcl and the KF regions of the FoxP2-Cre mice. Photomicrographs (**a**, **b**) from a wild-type mouse (with no Cre expression, **a** WT-DTA) and a FoxP2-Cre mouse **b** injected with AAV-mCherry-Flex-DTA, immuno-labeled for FoxP2 (green) and mCherry (red), at three levels (rostal, middle, and caudal; a$_1$–b$_3$), with magnified merged panels of FoxP2 and mCherry for better view. The photomicrographs show that the spread of the AAV-DTA injection with mCherry labeling covered both KF and PB regions in (**a**, **b**). In the first group of mice **a** that lacked cre-recombinase, AAV-FLEX-DTA did not ablate the KF$^{FoxP2}$ and the PB$^{FoxP2}$ neurons (green); while in **b** where the virus

injections were in the FoxP2-Cre mice, very few FoxP2 cells in the KF and the PBcl were spared (**b** PB + KF$^{FoxP2}$-DTA). However, both groups showed expression of mCherry in neurons in PBcl and PBel which lacked FoxP2. Scale in a$_1$–b$_3$ = 100 μm. KF Kölliker-Fuse PB subnucleus, PBel external lateral PB subnucleus, scp superior cerebellar peduncle, vsct ventral spinocerebellar tract. Genetic ablation of both PB$^{FoxP2}$ and KF$^{FoxP2}$ neurons significantly reduced the increases in $V_T$ (**d**) and MV (**e**) but not RR (**c**) caused by $CO_2$ exposure (hypercapnia shown by a yellow rectangle). Values of RR, $V_T$, and MV are mean ± SEM ($n = 8$ WT-DTA; PB + KF$^{FoxP2}$-DTA = 5). Two-way ANOVA, with *$P < 0.05$; **$P < 0.001$, followed by Holms-Sidak method for multiple comparisons, for PB + KF$^{FoxP2}$-DTA compared to the WT-DTA. Source data are provided as a source data file.

we showed that among the neurons in the PBcl and KF (i.e., outside the PBel$^{CGRP}$ group) that show cFos expression after $CO_2$ exposure, nearly 75% express FoxP2. This PBcl/KF$^{FoxP2}$ population had no overlap with the PBel$^{CGRP}$ neurons (i.e., no PBel$^{CGRP}$ neurons expressed FoxP2), indicating that they are independent cell populations[34]. Interestingly, we found a small number of KF neurons that express *Calca* mRNA (SFig. 1). These neurons are a bit smaller than the CGRP neurons in the PBel, and it is likely that they account for the projection shown by Geerling and colleagues[34] by CGRP neurons to respiratory sites in the medulla, similar to the remainder of KF glutamatergic neurons (see

ref. 33). These small CGRP neurons appear in a recent transcriptomic analysis to be a genetically distinct population from the large PBel CGRP neurons[35]. Interestingly, the transcriptional analysis and our in situ hybridization results reported here (SFig. 1) indicate that many of the KF$^{CGRP}$ neurons also express FoxP2, underscoring that they are a separate population and demonstrating the value of FoxP2 in the PBcl and KF as a marker for identifying the neurons that drive ventilatory responses to $CO_2$. It should be pointed out, however, that FoxP2 is expressed more widely in more dorsally located neurons in the lateral PB, but only the FoxP2 neurons in the PBcl, KF, and lateral crescent

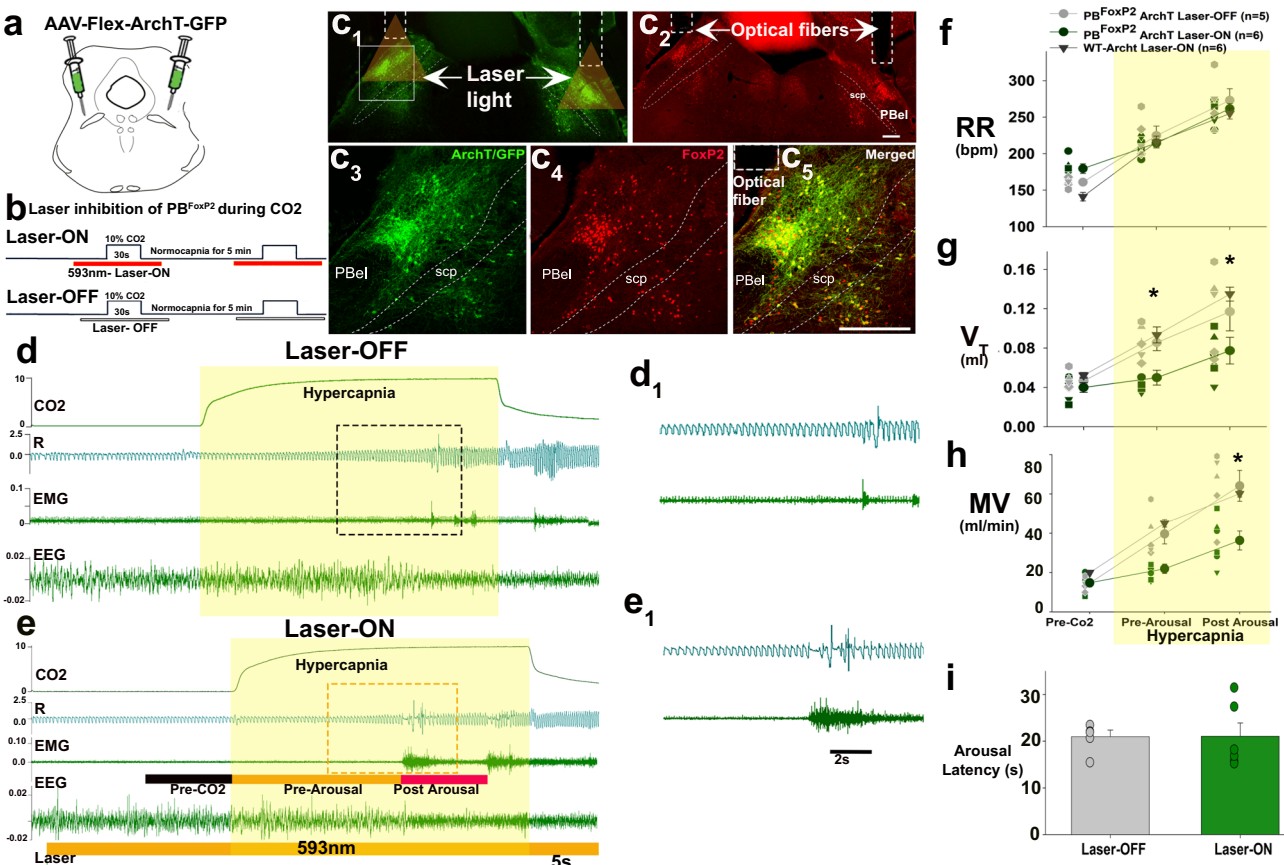

**Fig. 7 | Acute optogenetic silencing of the PB$^{FoxP2}$ neurons blocks hypercapnia-induced respiratory drive.** FoxP2-Cre mice were injected with AAV-FLEX-ArchT (**a**), and implanted for EEG/EMG and with bilateral optical fibers targeting PBcl$^{FoxP2}$ neurons ($c_1$, $c_2$). There was virtually complete co-expression of FoxP2 (red) and ArchT (green, $c_{3-5}$). The experimental strategy to test respiratory responses to 10% $CO_2$ given for 30 s every 300 s, with and without photo-inhibition of the PB$^{FoxP2}$ neurons is shown as a schematic in (**b**). Representative examples of a laser-ON and a laser-OFF trial are shown in (**d**, **e**), with magnified views of the bounding box from each trial shown in $d_1$ and $e_1$. For each stimulation, the laser was switched ON for 60 s beginning 20 s prior and extended for 10 s after the $CO_2$ stimulus (30 s,

illustrated by the yellow box) as shown in (**e**). Photoinhibition of PB$^{FoxP2}$ neurons significantly reduced the increases in $V_T$ (**g**) and MV (**h**) but not RR (**f**) caused by $CO_2$ exposure, although not as much as ablating both the PB$^{FoxP2}$ and KF$^{FoxP2}$ neurons. Values of RR (**f**), $V_T$ (**g**), and MV (**h**) are mean ± SEM. Two-way ANOVA, followed by Holms-Sidak method for multiple comparisons, with $^*P < 0.05$, compared to the laser-OFF and wild type injected with ArchT (controls). The photo-inhibition of PBcl$^{FoxP2}$ neurons did not change the latency to arousal (**i** mean ± SEM, laser-OFF ($n = 5$) vs laser-ON ($n = 6$)). For data shown in **f–i**, '$n$' represents the number of mice. Scale in $c_2$ and $c_5 = 200$ μm. Source data are provided as a source data file.

express cFos in response to $CO_2$. We are currently searching for more selective markers for this neuronal population.

The second strategy we used to investigate the relationship of the PBcl$^{FoxP2}$ neurons with the ventilatory response to $CO_2$ was the use of fiber photometry to study the $Ca_i$ responses of PBcl$^{FoxP2}$ neurons during wake–sleep and during $CO_2$ exposure. The PBcl$^{FoxP2}$ neurons were clearly wake and REM-active and their activity cross-correlated with increased RR during REM sleep and with both increased RR and EMG activation during wake. As suggested by the cross-correlations with the onset of elevated EEG γ-frequency activity (indicative of wakefulness), the cortical arousals preceded calcium activation of the PBcl$^{FoxP2}$ neurons, while increased RR followed their activation. This relationship is consistent with our finding that the activation of the PBcl$^{Foxp2}$ neurons drove increased respiration, which then increased further after EEG desynchronization, whether this occurred during the optostimulation of the PBcl$^{FoxP2}$ neurons, or after it had stopped. The PBcl$^{FoxP2}$ neurons as a group showed a brisk increase in calcium signal beginning a few seconds after the onset of a $CO_2$ stimulus, but persisting for many seconds after the stimulus was removed. When we resolved the responses of individual PBcl$^{FoxP2}$ neurons with GCaMP6s endoscopy during exposure to $CO_2$, the individual PBcl$^{FoxP2}$ neurons showed multiple recurrent peaks of calcium activation, with the first peak at

$11.0 \pm 0.2$ s after the onset of the $CO_2$ stimulus and then further peaks at roughly 17–19 s intervals. The mechanism for these rhythmic pulses of activity in the PBcl$^{FoxP2}$ neurons remains a mystery, although we speculate that it may be due to another input to those neurons, such as a periodic vagal input perhaps due to waves of peristalsis from the oropharynx or esophagus, whose swallowing activity is closely coordinated with breathing[39].

These experiments admittedly involved only a small number of PBcl$^{FoxP2}$ neurons and because of the limited depth of field of the GRIN lens we used, all of them were in a small cluster near the surface of the lens. However, if the larger population of PBcl$^{FoxP2}$ neurons also responds rhythmically to $CO_2$, our results would suggest that the much smoother population response as imaged by fiber photometry is probably a summation of the activity of different subsets of PB$^{FoxP2}$ neurons that are activated at different times in different locations in the PBcl during $CO_2$ stimulation. Our anterograde and retrograde tracing experiments indicate that PBcl$^{Foxp2}$ neurons may mediate these effects on respiration by their descending projections to the medullary targets involved in ventilatory rate and volume[33].

The third strategy we used to characterize the role of the PB$^{FoxP2}$ neurons in the response to $CO_2$ involved optogenetic excitation and

inhibition. When the PB$^{FoxP2}$ neurons were opto-stimulated for 5 or 10 s at 10 Hz or 20 Hz while mice in NREM sleep were breathing normocapnic air, the mice showed a statistically significant increase in RR and MV (but not $V_T$) that was greater both with higher frequency or prolonged duration of stimulation. The rapid onset of elevated RR but not $V_T$ was similar to the response to $CO_2$ stimulation, in which the increase in $V_T$ is delayed by about 10 s. Because the photostimulation was limited to 10 s (because animals would frequently awaken with more prolonged stimulation), there may not have been sufficient time for an increase in $V_T$ to develop. Interestingly, with 10 s photostimulation, the animals only awakened either late in the stimulation episode or immediately afterward, suggesting that the EEG arousal was not a direct effect of the stimulation, in contrast to the almost immediate EEG arousal seen with photo-stimulation of the PBel$^{CGRP}$ population[18]. In this regard, it was interesting that when animals were stimulated optogenetically while breathing 2% $CO_2$, the latency of the arousal was shorter, but occurred at about the same RR as during the optostimulation while breathing room air or in response to $CO_2$ without optostimulation (i.e., awakening in all three conditions occurred at about 250–260 breaths/min, or when RR was about 65% greater than baseline). This observation suggests that the arousal may have been due to somatomotor feedback from the increased ventilatory effort. Alternatively, it is possible that either the PBcl$^{FoxP2}$ neurons or brainstem cell groups downstream from them may have collaterals that activate PBel$^{CGRP}$ or some other arousal neurons. It would be useful to test whether simultaneous suppression of PBel$^{CGRP}$ neurons while activating PBcl$^{FoxP2}$ neurons might permit increased ventilation without EEG arousal.

Our finding that optostimulation of PB$^{FoxP2}$ neurons caused a rapid and robust increase in RR with minimal change in $V_T$ suggests that the RR and $V_T$ responses may depend upon different circuits. The increases in RR are similar to those shown earlier[42–44] for photostimulation of the PBZ region, where low power stimulation (20 Hz at 2 mW) of inhibitory neurons (GABA-ergic and glycinergic) increased the RR but not the $V_T$. Similarly, photo-activation of the catecholaminergic neurons in the rostral part of the ventrolateral medulla which project to the respiratory rhythm-generating neurons located in the PBZ[45–49] also increased the RR in a dose-dependent manner[47]. These findings suggest that the PBcl$^{FoxP2}$ neurons input to the PBZ is likely to cause the increase in RR (Fig. 8).

On the other hand, because there are other $CO_2$-sensitive inputs to the PBZ (e.g., the retrotrapezoid nucleus)[50], the PBcl$^{FoxP2}$ neurons may be sufficient, but not necessary to drive a RR response to $CO_2$. We found that photo-inhibition of the PB$^{FoxP2}$ neurons or deletion of the PB + KF$^{FoxP2}$ neurons did not prevent the rise in RR during a 30 s exposure to $CO_2$ but did diminish the rise in $V_T$ and MV, not only during the $CO_2$ exposure but even after the cortical arousal, when the largest increase in $V_T$ is generally observed. On the surface, it may appear that the increase mainly in RR with photostimulation of the PB$^{FoxP2}$ neurons and decrease mainly in the $V_T$ with photoinhibition of the same cells is inconsistent. However, the photostimulation was done while the animals were breathing room air, so the increase in RR caused by the PB$^{FoxP2}$ neurons was not masked by the contributions of other brainstem sites (such as the retrotrapezoid nucleus) that are also engaged by high $CO_2$ levels, and project to the PBZ (Fig. 8). On the other hand, during 10% $CO_2$ stimulation, when these other $CO_2$-responsive sites were already increasing RR to its maximum[48,51–53], the inhibition or deletion of PB/KF$^{FoxP2}$ neurons revealed that although they are not needed to reach maximal RR, they are necessary to achieve the maximal increase in ventilatory volume. In other words, brief (5–10 s) activation of the PB/KF$^{FoxP2}$ neurons is sufficient to drive RR (but not $V_T$, which may take longer to engage) when breathing room air, and necessary to drive increased $V_T$ during more prolonged (>15 s) exposure to high $CO_2$ levels[54–57]. This response may be driven by inputs from the PBcl/KF$^{FoxP2}$ neurons to respiratory premotor and motor sites, including the nucleus ambiguus, CVLM, hypoglossal nucleus, and

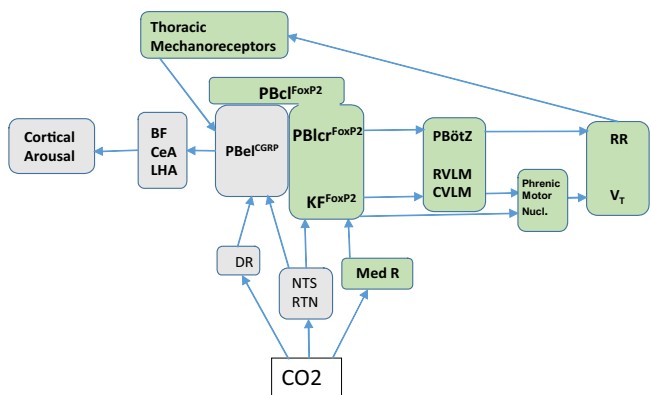

**Fig. 8 | Proposed neural circuit for mediation of hypercapnia-induced increases in ventilation.** PB neurons receive information about $CO_2$ levels via relays from the carotid body through the NTS (nucleus of the solitary tract), and directly from chemosensory neurons including the RTN (retrotrapezoid nucleus)[20,45,49,51], medullary (Med R), and dorsal (DR) raphe nuclei[47,58,64]. The projections of the PB$^{FoxP2}$ neurons to the PBZ, rostral ventrolateral medulla (RVLM), and the CVLM may mediate the increase in ventilation by increasing RR[23,56,67,68] and tidal volume ($V_T$) or directly by the projections from the KF FoxP2 neurons to the phrenic motor nucleus as previously shown by others[53,65]. Mechanical sensory feedback by thoracic stretch receptors during increased ventilatory effort may activate PBel$^{CGRP}$ neurons to cause cortical arousal, via activation of neurons in the basal forebrain (BF), central nucleus of the amygdala (CeA), and lateral hypothalamic area (LHA)[18,59]. Selective activation of PB$^{FoxP2}$ neurons may be effective for augmenting breathing in apneas if combined with ways to suppress the EEG arousal.

phrenic motor nucleus. A schematic model for the role of PB + KF$^{FoxP2}$ neurons in the proposed neuronal circuit for hypercapnia-induced increase in ventilation is shown in Fig. 8.

In conclusion, our experiments reveal that PBcl/KF$^{FoxP2}$ neurons likely contribute both to the increases in RR and $V_T$ during exposure to $CO_2$ in NREM sleep, but that they are not the only pathway driving these responses (Fig. 8). On the other hand, if ways could be found to augment the PBcl/KF$^{FoxP2}$ response to $CO_2$, this may be sufficient to avoid hypoventilation and minimize EEG arousals in patients with OSA. However, activating the PBcl/KF$^{FoxP2}$ neurons to increase RR by 65% or more above baseline may by itself contribute to EEG arousal. It is not known whether this arousal is due to the mechanical sensory feedback of increased ventilatory effort, or might represent collaterals from the PB$^{FoxP2}$ neurons that indirectly activate PBel$^{CGRP}$ neurons or some other arousal pathways. However, for the activation of PB$^{FoxP2}$ neurons to be effective in treating patients with OSA, it will be important to find ways to suppress the EEG arousal while augmenting the respiratory response to $CO_2$.

## Methods
### Animals
We employed *Foxp2$^{tm1.1(cre)Rpa}$*/J transgenic mice originally prepared by Dr. Richard Palmiter, University of Washington, and obtained from Jackson Laboratories, in which IRES-Myc tag-nuclear localization signal (NLS)-Cre-GFP-frt-neomycin-frt was introduced just after the termination codon of the mouse Foxp2 gene via homologous recombination. The mutation was created via homologous recombination in (129S6/SvEvTac × C57BL/6) F1-derived G4 embryonic stem cells. The frt-flanked neomycin cassette was excised through crosses with animals that broadly expressed Flp recombinase. The GFP is believed to be nonfunctional. Resultant mice were backcrossed to C57BL/6J for nine generations by the donating laboratory to the Jackson laboratory (Strain #:030541; RRID: IMSR_JAX:030541). All transgenic mice used here were heterozygous for the transgene and backcrossed to the C57BL6 strain and their littermates with no Cre expression (wild type) were used as controls. We bred these mice in our animal facility and confirmed their genotype by using a Red

Extract N-amp Tissue PCR kit (Sigma Aldrich; Catalog# XNAT-1000RXN) and specific primers for FoxP2Cre. Their wild-type littermates were used as controls, in each experiment.

All mice used in these experiments were male (8–10 weeks old at the time of surgery) because female mice of the same age are smaller, and including animals of various sizes would introduce noise into the analysis of respiratory volumes (which scale with body size) across groups. Animals were maintained on a 12 h light/dark cycle with ad libitum access to water and food and were singly housed after surgery, with an ambient temperature of 21–23 °C and humidity levels between 40% and 60%. Male littermates were randomly assigned to the experimental groups. All animal procedures met National Institutes of Health standards, as described in the Guide for the Care and Use of Laboratory Animals, and all protocols (protocol #032-2020-23) were approved by the Beth Israel Deaconess Medical Center Institutional Animal Care and Use Committee.

### Validation of mice

To test if the Cre expression was eutopic with FoxP2 expression, FoxP2-Cre mice from Jackson Labs were crossed to the L10 reporter line. The Cre-positive neurons were labeled with a green fluorescent protein (GFP), and when the tissue was immuno-stained for FoxP2, we could observe almost all green (GFP) neurons expressed labeling for FoxP2 (red) in their nuclei (SFig. 2a–d). This confirmed that we could reliably use these mice for expressing Cre-dependent viral vectors in the FoxP2 neurons in the PB (examples of Cre-dependent transfections Figs. 2b, 4b, and 6c). $PB^{Foxp2}$ neurons did not overlap with the $PB^{CGRP}$ neurons in the lateral PB.

### Vectors

For fiber-photometry, we used AAV-pSyn-GCaMP6s from Addgene_100843. For optogenetic experiments, we used the excitatory opsin (AAV-EF1a-DIO-hChR2(H134R)-mCherry-AAV-serotype8) and the optogenetic neural silencer AAV-CAG-FLEX-ArchT-GFP (AAV-serotype-8). A Cre-dependent viral vector expressing diphtheria toxin subunit-A (AAV-mCherry-FLEX-DTA) was used for genetic ablation, as validated earlier by us for selective Cre-dependent neuronal ablation[18,58]. These viral vectors were procured from the University of North Carolina (UNC) vector core and the optogenetic vectors have been previously used for excitation, as well as for silencing neurons and their terminals in our lab[18,58].

### Surgery

Under surgical anesthesia, mice were instrumented for sleep with implantation of EEG and EMG electrodes in addition to implanting either the fiber photometry cannula (unilateral), integrated GRIN lens with baseplate (unilateral) or bilateral optical fibers targeted to the PB area (AP: −5.1 mm to −5.3 mm; DV: 2.6 mm; ML: ±1.3 mm). All implants were done 5 weeks post injection of the viral vectors in the PB, to ensure optimal gene expression. After recovery from surgery, mice were recorded for sleep and respiration after acclimatizing them to the recording apparatus for at least one week before the actual recordings were performed. We recorded the arousal and respiratory responses to $CO_2$ by placing them in the plethysmograph and recording them for both sleep and breathing by the procedure described previously[18,20,58].

**Experiment 1a.** cFos experiments: C57BL/6 J mice ($n = 8$) were moved from the animal housing room and placed in the plethysmographs where they were first habituated to the chamber in room (normocapnic) air for 2 h, then exposed continuously to either hypercapnic (21% $O_2$; 10% $CO_2$; 69% $N_2$) or normocapnic air (21% $O_2$; 0% $CO_2$; 79% $N_2$) for 2 h between 10:00 and 12:00. Mice were then deeply anesthetized with chloral hydrate (500 mg/kg, ip) and transcardially perfused with saline, followed by 10% formalin. Brains were removed, post-fixed overnight in 20% sucrose, and cut into four alternate series of 30-

micron frozen sections. After immunostaining the tissue for cFos and FoxP2 as described below, sections were scanned using an Olympus VS200 slide scanner, and images were analyzed using Olympus OlyVia 3.3 software. Nuclei of cells that were singly or doubly labeled for cFos and FoxP2 were counted in the PBcl (0.5 mm × 0.6 mm placed dorsal to the single labeled cFos in the PBel, as in Fig. $1a_2$–$d_3$) and KF (0.5 mm × 0.5 mm placed ventral to the ventral spinocerebellar tract, as in Fig. $1a_1$–$d_1$) subnuclei. To avoid counting bias, the cell counts were performed by investigators blind to the treatment groups (NL and JDL). Counts were corrected for the nuclear size using the Abercrombie correction factor.

**Experiment 1b.** Fiber photometry implants: in vivo, calcium monitoring during sleep–wake and with $CO_2$ exposures were performed using fiber photometry. Five weeks after the injection of AAV-pSyn-GCaMP6s in the PB, the optical fiber was connected via a metal sleeve (Doric Lenses, MFC_400/430-0.48_5mm_MF-1.25) to a low-fluorescence fiber optic patch cord (0.48NA, Doric Lenses). After allowing the animal to adapt to the environment, we simultaneously delivered light via LED drivers at 465 nm and 405 nm (Doric Lenses, CA), to measure the calcium-dependent and calcium-independent (UV, isosbestic), excitation of the GCaMP. Emission from the GCaMP protein passed back through the fiber optic patch cord and through the fluorescence Mini Cube (Doric Lenses) and was detected by a photo-receiver (Doric Lenses). Signals detected by the photo-receiver were transmitted to an Axon Digidata 1322 A analog-to-digital converter and the signals were acquired using Axoscope software v10 (Molecular Devices, Foster City, CA, USA), alongside the EEG/EMG and breathing signals. We exported files to Spike2 and analyzed the respiratory signals using the Spike respiratory scripts (Resp80t, Spike2, CED, UK) that were then correlated with the GCaMP activity.

The GCaMP6 raw data was normalized to the baseline fluorescence of each trial to obtain the $\Delta F/F$ (change in fluorescence intensity relative to the baseline fluorescence intensity) for different animals. The GCaMP $\Delta F/F$ values per second was calculated for 15 s before and during $CO_2$ for each trial and statistically compared, along with RR and $V_T$ values for similar periods. The peak values of GCaMP $\Delta F/F$, and the latency to peak during the entire 15 s with $CO_2$ exposure were also calculated. Two-way ANOVA was performed to compare the effects between pre and post-$CO_2$ exposures on the GCaMP $\Delta F/F$ and also for RR and $V_T$. Cross-correlation analysis between the $\Delta F/F$ changes with RR, EMG, and power bands in EEG (delta and gamma) as reference waveforms were calculated across animals using Spike2, CED, UK.

**Experiment 1c.** Calcium fluorescence endoscopy, GRIN lens implants: $Foxp2^{tm1.1(cre)Rpa}$/J mice (heterozygous FoxP2-Cre) were injected in the PBcl with AAV-pSyn-DIO-GCaMP6s. Three weeks later, they were implanted with a microendoscopic ProView™ Integrated Lens 0.6 mm × 7.3 mm (Inscopix Catalogue# 1050-004413) that allowed for visualizing the GCaMP activity during the lens implant. The lens was targeted to be ~200–300 μm above the neurons using the following coordinates −5.0 mm posterior to bregma, −1.4 mm lateral from the midline, and −2.8 to 3.0 mm ventral to the dura mater. The baseplate provided the interface for attaching the miniature microscope during the calcium-imaging experiments, but at other times a baseplate cover (Inscopix Catalogue# 100-000241) was attached to prevent damage to the microendoscopic lens. Out of eight mice injected with GCaMP6s virus, four had successful implants and were used for the study.

Calcium imaging: we imaged the calcium activity at five frames per second, 200-ms exposure time at 20–30% LED power using the miniature microscope from Inscopix (nVista). These parameters caused minimal bleaching and allowed long-term recordings in mice. Mice were recorded for 5 min every hour to correlate the calcium activity to the 12-h sleep–wake behavioral data. For correlating to the $CO_2$-

induced respiratory changes, mice were habituated to the plethysmograph recording chamber for 3–4 h and then recorded for 3 h, receiving the repeated stimulus of 30 s of 8% $CO_2$ every 5 min. Imaging was done for four trials (-20 min) every hour for 3 h with a week of separation between subsequent recordings. This protocol caused minimum bleaching and allowed us to do up to 3 recording sessions per animal.

**Experiment 2.** Optogenetic activation of the $PB^{FoxP2}$ neurons: for selective activation of the $PB^{FoxP2}$ neurons, we injected an adeno-associated viral vector expressing the excitatory opsin AAV-FLEX-ChR2-mCherry in the PB (AP: −5.1 mm to −5.3 mm; DV: 2.6 mm; ML: ±1.3 mm) and implanted these mice ($n = 19$) with bilateral optical fibers targeting the lateral PB. We also injected some wild-type mice ($n = 3$) with AAV-FLEX-mCherry as well, which served as a control, as we did not observe any expression of mCherry. We recorded these mice for sleep and respiration when they were exposed to room air or $CO_2$, in a plethysmography chamber[18,19,58,59]. Mice were subjected to 5 s or 10 s of 467 nm laser stimulus with a pulse width of 10 ms and with frequencies of either 5 Hz or 10 Hz or 20 Hz, in random order and with each treatment separated by at least 7–10 days, either in normocapnic conditions or with continuous 2% $CO_2$.

**Experiment 3.** 3a Optogenetic inhibition of the $PB^{FoxP2}$ neurons: a separate set of FoxP2-Cre mice ($n = 12$) and wild type ($n = 6$) were injected in the PB with AAV-FLEX-ArchT-GFP and bilaterally implanted with optical fibers targeting the PB (AP: −5.1 to −5.3 mm; DV: 2.6 mm; ML: ±1.3 mm) for the inhibition of the $PB^{FoxP2}$ neurons. At 5 weeks post-injection, these mice were recorded for sleep and breathing in a plethysmography chamber, where the arousal and respiratory responses were assessed while they were subjected to repetitive $CO_2$ stimulations as shown earlier[18,19,58,59].

**3b Genetic deletion of $PB^{FoxP2}$ neurons.** A set of FoxP2-Cre ($n = 8$) and wild type ($n = 8$) were injected in the PB, bilaterally with AAV-mCherry-Flex-DTA, a vector that has been validated to ablate Cre expressing neurons[18,58]. At 5 weeks post-injection, these mice were recorded for sleep and breathing in a plethysmography chamber, where the arousal and respiratory responses were assessed after subjecting them to repetitive $CO_2$ stimulation. Ablation of the FoxP2 neurons was validated by standard immunohistochemistry for FoxP2 (Fig. 6a–b).

**Histology.** At the conclusion of the experiments, the animals were perfused with phosphate-buffered saline (PBS) followed by 10% buffered formalin while under deep anesthesia. Brains were harvested for analysis of the effective location of the injection site. Brains were kept in 20% sucrose for 2 days and sections were cut at 30 μm using a freezing microtome in four 1:4 series.

**Immunohistochemistry.** Sections from Experiment 1a were processed for the detection of c-Fos in combination with FoxP2. All incubations were performed on free-floating tissue sections at room temperature. Sections were first incubated overnight in primary antibody. The c-Fos antibody (Oncogene Sciences, Cat# Ab5 was a rabbit polyclonal, antiserum raised against amino acids 4–17 of human c-Fos. This antiserum stained a single band of 55 kDa on Western blots from rat brains (manufacturer's technical information). It was diluted 1:10 K in PBS with 0.2% Triton X-100. After rinsing, sections were incubated in Alexa-488 (green fluorescence) conjugated donkey anti-Rabbit-IgG (Invitrogen, A11055) at 1:200 in PBS containing 0.2% Triton X and 2.5% normal donkey serum for 3 h. After rinsing in PBS, sections were next incubated overnight with sheep anti-FoxP2 (R and D Systems Cat# AF5647, RRID: AB_2107133) diluted 1:5000 in PBS with 0.2% Triton X-100 and 2.5% normal donkey serum. After rinsing with PBS the next day, sections were incubated with biotinylated secondary antibody (Donkey anti-sheep-biotinylated, 1:200, Cat#-713-065-147; RRID: AB_2340716;

Jackson-Immuno Research) for 2 h followed by incubation in the streptavidin-conjugated Cy3 (red fluorochrome) (1:200, Thermo Fischer, Cat# 434315) for 2 h at room temperature. The immunostained sections were mounted and cover-slipped with fluorescence mounting medium (Dako, North America).

In the remaining experiments, mice injected with AAVs expressing either GCaMP6s, ArchT-GFP, ChR2-mCherry or DTA-mCherry were immunostained either for GFP (Rabbit anti-GFP, 1:10 K, Molecular Probes Cat# A-11122, RRID: AB_221569) or mCherry (Rabbit anti-DsRed. 1:2 K, Clontech, Cat-632496) as per standard immunohistochemistry protocols described previously[18,58]. These were then double stained for FoxP2 using Sheep anti FoxP2, 1:5 K (R and D Systems Cat# AF5647, RRID: AB_2107133) using the protocol described above. Sheep polyclonal antiserum is raised against recombinant human FoxP2 isoform 1, Ala640-Glu715 (accession # O15409), shown by the manufacturer to be specific for humans and mice, shows a single band for FoxP2 at approximately 80 kDa, and is also previously used by others[60–62].

Some of the brains ($n = 3$) from WT mice were injected with cholera toxin subunit b (CTb) and were immunostained using Goat anti-CTb (1:30 K, Cat# 703, RRID: AB_10013220, List Biological Laboratories Inc., CA). These were double-labeled for FoxP2, using Rabbit anti-Foxp2, 1:10 K, Abcam Cat# ab16046, RRID: AB_2107107) and the tissue was processed using standard immunostaining protocols. The rabbit polyclonal antibody to FoxP2 was raised against a synthetic peptide made from residues 700 to the C-terminus of human FoxP2 which was conjugated to keyhole limpet hemocyanin. This antibody also showed a single band in Western blots performed by the manufacturers and was specific to human and mouse FoxP2. For double labeling of CTb and FoxP2, brain sections were incubated in fluorescent-labeled secondary antibodies donkey anti-goat-Alexa-555 (red) at 1:200 (Cat# A 21432, RRID: AB_2535853) or donkey anti-rabbit-Alexa-488 (green) at 1:200 (Cat# A 21206, and RRID: AB_2535792) (Molecular probes, Thermo Fischer Scientific) after overnight incubations in the respective primary antibodies at room temperature. After rinsing with PBS the next day, sections were then incubated in the respective secondary antibodies for 2 h. After mounting and air drying the sections, the glass slides were cover-slipped with a fluorescence mounting medium (Dako, North America). When acquiring confocal images, sometimes pseudocolors were used to enhance clarity.

**Fluorescent In situ hybridization (FISH using RNA Scope).** We identified FoxP2 neurons by using *Foxp2tm1.1(cre)Rpa*/J mice crossed with R26-lox-STOP-lox-L10-GFP reporter mice (FoxP2-L10, $n = 3$), and labeled for *Calca* (the gene for CGRP) by using a set of FISH probes with RNAScope in brain sections from the KF and PBcl areas. The brain was sectioned at 30 μm and sections were mounted on glass slides in RNAase-free conditions, and RNA scope was performed using the multiplex fluorescent reagent Kit V2 (Cat# 323100, Advanced Cell Diagnostics, Hayward, CA). Brain sections on the slides were pretreated with hydrogen peroxide for 20 min at room temperature and then with target retrieval reagent for 5 min (at a temperature above 99 °C), followed by dehydration in 90% alcohol and then air-dried for 5 min. This was followed by a treatment with protease reagent (Protease III) for 30 min at 40 °C. After rinsing in sterile water, sections were hybridized with the *Calca* FISH probes (RNAscope probe: Mm-*Calca*-tv2tv3-C1-Mus musculus, calcitonin-related polypeptide alpha, transcript variant 2 mRNA; Catalog# 420361; Advanced Cell Diagnostics, Hayward, CA) for 2 h at 40 °C. This probe targets Mus musculus (mouse) calcitonin/calcitonin-related polypeptide, alpha (*Calca*), mRNA region (accession #NM_001033954.3) from 63–995 bp (total length of Calca mRNA is -1021 bp), and has been used previously to selectively label *Calca* expressing neurons on the brain tissue[34,63]. Sections were then incubated in three amplification reagents (AMP) at 40 °C (AMP1 for 30 min, AMP2 for 30 min, and AMP3 for 15 min) followed by horseradish peroxidase (HRP)−C1amplification at 40 °C for

15 min. Sections were then incubated in tyramide signal amplification reagents with a Cy3 fluorophore (Cat# NEL744001KT, Perkin Elmer, 1:1000) for 30 min to amplify and visualize CGRP mRNA in red. In the final step, sections were subjected to HRP blocking using the blocker from the reagent kit (Cat# 323100, Advanced Cell Diagnostics, Hayward, CA) for 15 min at 40 °C. After each step, sections were washed with 1× wash buffer provided in the kit. Following the CGRP RNAscope in situ hybridization, immuno-labeling of GFP was performed on the same sections, as the in situ hybridization procedure quenched the native GFP fluorescence. For this, the brain sections were incubated in rabbit anti-GFP (1:1500), (Cat# A6455; Lot# 1220284; Molecular probes) overnight at 4 °C, washed in PBS (3 × 2 min) and then incubated in secondary antibody (Alexa Fluor-488 Donkey anti Rabbit, Life Technologies, Cat# A-21206) for 2 h at room temperature. Finally, the slides were dried and cover-slipped with Dako fluorescence mounting medium (Cat# S302380-2, Agilent, CA), and scanned for analysis.

## Data acquisition

All recordings were done five weeks after the injection of the viral vectors. All sleep and respiration recordings were done in a plethysmography chamber (unrestrained whole-body plethysmograph, Buxco Research Systems) which allowed us to record the breathing of the mouse while continuously monitoring the gas in the chamber. Electroencephalogram (EEG) and electromyogram (EMG) were recorded using Pinnacle preamp cables connected to an analog adapter (8242, Pinnacle Technology). Gas levels in the chamber were continuously monitored using $CO_2$ and $O_2$ monitors from CWE, Inc (Ardmore, PA, USA). EEG, EMG, respiration, and $CO_2$ and $O_2$ levels were fed into an Axon Digidata 1322A analog-to-digital converter and the signals were acquired using Axoscope software v10 (Molecular Devices, Foster City, CA, USA), or by acquired by the 1401 (CED, Cambridge, UK) and Spike2 ver.7 (CED, Cambridge, UK). Mice were connected to cables for sleep recording and the fiber optic cables were connected to the pre-implanted glass fibers in mice, for transmitting the laser light.

Mice were placed in the plethysmography chamber beginning at 9:00 a.m. for 6 h during their lights-ON and behaviorally inactive period, on each test day for these recordings. Here, they underwent either the laser-ON or laser-OFF protocols, separated by a week and in random order.

During the laser-ON protocol, with the 473 nm laser, the photo-stimulations were done at either 5 Hz, 10 Hz, or 20 Hz with a pulse width of 10 ms, with a period of 6–7 days between each treatment. During photo-activation using the 473 nm laser, only normocapnia air was used in the chamber. The inhibitory 593 nm laser stimulations were continuous for 60 s, and preceded each 30 s of $CO_2$ stimulus by 20 s and lasted 10 s after the hypercapnia stimulus. In the laser-OFF condition, everything was the same, except that the laser light was not turned on. The gas input for the plethysmograph was switched either to normocapnic air (21% $O_2$, 79% $N_2$) or hypercapnic air (10% $CO_2$, 21% $O_2$, and 69% $N_2$) for 30 s with 5 min in between the two hypercapnic stimuli. For photo-activation, inhibition or genetically targeted ablation experiments, trials were analyzed for latency to arousal and respiratory changes only for those epochs where the mouse was in NREM sleep for at least 30 s before the stimulus onset. In optostimulation experiments, we also analyzed trials where animals were awake for at least 30 s prior to.

## Laser light

Mice were allowed at least 2 days to acclimate to fiber optic cables (1.5 m long, 200 μm diameter; Doric Lenses, Quebec, QC, Canada) and connecting interfaces coated with opaque heat-shrink tubing before the experimental sessions. During laser-ON experiments, light pulses were programmed using a waveform generator (Agilent Technologies, catalog #33220 A, CA, USA) to drive either 10 ms pulses of 473 nm blue laser light (Laser Glow, Toronto, ON, Canada) at 5 Hz, 10 Hz, or 20 Hz, or drive the orange-yellow laser (593 nm; Laser Glow, Toronto, ON, Canada) which was continuously on

for 60 s beginning 20 s before the onset of the $CO_2$ stimulus. We also used a splitter–TM105FS1B (Thorlabs, NJ) to split the laser stimulus for bilateral activation or inhibition. We adjusted the laser such that the light power exiting the bilateral fiber optic cables was 8–10 mW, and this was checked before and after the experiment. The light power estimated at the PB is less than 10 mW/mm² (www.stanford.edu/group/dlab/cgi-bin/graph/chart.php), and a similar range has been used by most researchers and by us earlier[18,58,64,65]. Note that this is probably a high estimate because some light is probably lost at the interface between the fiber optic cable and the implanted optic fiber.

## Data analysis

**Latency and respiratory data analysis.** EEG arousals in response to $CO_2$ were identified by EEG transition from NREM (dominated by high voltage delta activity) to a waking (desynchronized) state, which was usually accompanied by EMG activation[19,66]. The latency of all the EEG arousals after the onset of stimulation was scored and compared across the laser-ON and laser-OFF days. Respiratory data was analyzed by running the respiratory script in the Spike2 (CED, UK) software, which performs breath-by-breath analysis for many respiratory parameters such as RR, $V_T$, and MV. For both latency and respiratory data, the ranges for analysis were selected by individuals who were blind to the treatment groups, who based the selection on the following criteria: 1. trials with at least 30 s of NREM sleep before $CO_2$/stimulation; 2. select five breaths (at 1–2 s) before $CO_2$ or laser stimulation, during $CO_2$/stimulation before arousal (at 1–2 s before stimulation ends or arousal whichever occurs first), and then at post-arousal (5–10 s post-stimulation) for respiratory analysis; and 3. exclude trials with REM sleep.

**Analysis of the histology for viral transfection and optical fiber tracks.** In all cases, the histology was reviewed by an investigator who was blinded to the physiological result, and cases with off-target placement of optical fibers or injections were considered to be anatomical controls (SFig. 6a and Fig. 7).

**Fiber photometry data.** The GCaMP and UV signals were Gaussian low-pass filtered at 4 Hz, and saved for offline analysis. Both signal channels (465 nm and 405 nm) were monitored continuously throughout recordings, with the 405 nm signal (UV) used as an isosbestic control. Signals detected with 405 nm wavelength light are not calcium-dependent and are indicative of background fluorescence or motion artifacts. A change in fluorescence ($\Delta F/F$) was calculated by normalizing to baseline fluorescence. For generating heat maps, min–max normalization was performed that causes linear transformation and the data is scaled in the range (0–1).

**Inscopix calcium-image processing.** Calcium recording files were spatially filtered and motion corrected for brain movements using the Inscopix data processing (IDPS ver1.6). To extract the calcium activity traces from the individual cells, we used manually drawn small regions of interest. Raw traces were converted to $\Delta F/F$ ($F - F_{baseline\ average}/F_{baseline\ average}$), where $F$ was the fluorescent at any given point and $F_{baseline\ average}$ was the average baseline fluorescence. These baseline image calculations were performed by IDPS to derive the $\Delta F/F$ values for each cell.

**Statistical analysis.** All statistical analyses were performed using SigmaPlot 14.5 (Systat Software, Inc.). For statistical comparisons, we first confirmed if the data meets with the assumptions of the ANOVA, then either one-way or two-way ANOVA was performed to compare the effects between various treatment groups. If differences in the mean values among the treatment groups were greater than would be expected by chance, then all pairwise multiple comparisons were performed using the Holm-sidak method. The $F$ and $P$ values are described in the results section with details of the statistical tests also

given in the respective figure legends and represented in the figures. The '*n*' is reported in the figures and the results represent the number of animals, and the error bars represent mean ± SEM. A probability of error of less than 0.05 was considered significant at alpha = 0.05.

## Reporting summary

Further information on research design is available in the Nature Portfolio Reporting Summary linked to this article.

## Data availability

Source data are provided with this paper. All data generated to support the findings of this study are also available from the corresponding author on request. Source data are provided with this paper.

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

## Acknowledgements

We thank Dr. Richard Palmiter for the generation of *Foxp2$^{tm1.1(cre)Rpa}$*/J transgenic mice used in the study, and for donating them to the Jackson Laboratory. We also thank Quan Ha, Sam Sailesh, and Rayna Jacob for their excellent technical support and help with scoring data. This research work was supported by funding from USPHS grants 1P01 HL149630-01 (CBS), NS 072337 (CBS), and NS112175 (SK).

## Author contributions

S.K.: experimental design, data collection, analysis, and manuscript writing; N.L., J.D.L., and R.C.T.: data collection and analysis; Y.S.: data analysis for the fiber photometry and calcium imaging; S.S.B.: maintaining the mouse breeding program and conducting in situ hybridization; and C.B.S.: experimental design, data analysis, and manuscript writing.

## Competing interests

All authors declare no competing interests.
