## [Peer Review File · Nature Communications]

Lateral parabrachial FoxP2 neurons regulate respiratory responses to hypercapniaREVIEWER COMMENTS

Reviewer #1 (Remarks to the Author):

MAJOR

The authors identify a new subclass of neurons in the parabrachial nuclear complex, including the adjacent Kolliker-Fuse, which is involved in increasing respiratory drive during chemosensory challenge, i.e., elevated carbon dioxide (hypercapnia). This new subclass is identified by transcription factor FoxP2, rather than expression of peptide CGRP, which the authors previously identified. The CGRP population is involved in arousal, but not respiratory drive. The FoxP2 subclass participates in both respiratory drive and arousal. These FoxP2 neurons are associated with REM sleep and wakefulness.

The experiments are convincing and complement the authors former discovery of the CGRP population and this study does advance understanding.

There are a few aspects that dampen enthusiasm:

- 1) It is unclear how the FoxP2 neurons get chemosensory information. Their activity in response to hypercapnia comes at a delay, and the latency to activation is heterogeneous (Figure 3d) so it is unlikely intrinsic chemosensitivity is at work. It seems likely that these neurons are connected to central chemosensory neurons in the retrotrapezoid nucleus / parafacial ventral (RTN/pFV). The authors should discuss how FoxP2 parabrachial / Kolliker-Fuse neurons receive chemosensory information.
- 2) The figures are difficult to understand in many respects. Figure 1 uses bar plots with standard error, rather than showing all the data points and plotting standard deviation or confidence intervals, which have much greater fidelity to the raw data and give the reader more evidence of effect size. Figure 2 uses inconsistent tick and typeface size, which make it extremely hard to gauge effect sizes. Figure 3 has similar problems regarding ticks and inconsistent sizes of typefaces. Panels b and c in Figure 3 are actual screen shots that cannot be read with any accuracy, straining the reader. Figure 3 panels b and c show the same 3 neurons and basically the same time series. Certainly those data can be taken from acquisition and analysis software and made into a presentable figure in a single panel. Figure 4 has the same problems with inconsistent typeface and ticks. Figure 5 is a patchwork of panels that contain valuable information but are packed so tightly together with all labels within the panels, again, straining the reader. Figure 6i is the type of "show the data" plot I would recommend for Figure 1e and f, but here a standard error or confidence interval would be preferable. Also, here the typeface in Figure 6b would be nearly unreadable in typical PDF size for reading (the reader will have to expand the illustration to get the information.)
- 3) The written manuscript overall suffers haphazardness. Time is sometimes "s" and other times "sec". Sometimes the units are separated from numerals, e.g. "20 Hz" and sometimes not "20Hz". Figure callouts are inconsistent too: "Fig. 1" or "Fig.1" or "Fig 1". These inconsistencies are rampant. While this critique does not have any bearing on scientific content, it does not, in my view, show attention to detail.

MINOR

Lines 37, 107: CGRP should be defined.

Line 91: how can ventilation increase during an apnea, when apnea -- by definition -- is a pause in breathing (no ventilation).

Figure 1e & f: the bar charts should be replotted to show all the data points as well as either 95%

confidence intervals or standard deviation rather than standard error of the mean, to show the data (see: Drummond GB, Vowler SL. Show the data, don't conceal them. *J Physiol (Lond)* 589:1861–1863, 2011)

Line 141: the number of the figure is missing.

Figure 2d, f-i: the y-axes are labeled with too many and often small ticks and too small type face. It is difficult to read these axes and make sense of the data. The x-axes are inconsistently labeled with different size type faces and time is "s" or "sec". The graphics should be more elegant and easier to read (see: Tufte ER. *The visual display of quantitative information*, 2nd ed. Cheshire, Conn: Graphics Press, 2001.)

Lines 180-204: The figure callouts are sloppily labeled, sometimes missing a period (180, 181, 190) and other times lacking a gap between the period and the numeral (192, 198, 204). While these simple typos have no bearing on scientific content, the sloppiness is disturbing for its excessive frequency.

Figure 4e: Should be "minute ventilation" not "minute volume".

Line 209: missing "P="

Line 218: "movement" is misspelled.

Line 242: I think the callout should be to Fig. 3e (not 3f).

Line 302: "CTb" should be defined in the main text when it appears. It is defined in the figure legend, but one expects a definition in the main narrative too.

Reviewer #2 (Remarks to the Author):

Summary of key results:

This study evaluates whether Foxp2-expressing neurons in the lateral parabrachial nucleus (LPBN) or nearby Kölliker-Fuse nucleus (KF) play a role in the hypercapnic ventilatory response (HCVR) in mice. HCVR is the increased breathing rate and volume triggered reflexively in response to increased CO₂ (and the resulting drop in blood pH). HCVR is important for survival, particularly in patients with obstructive sleep apnea, in whom this reflex (and a parallel surge in arousal) kick-starts breathing after prolonged apneas. Understanding the neurologic mechanisms of HCVR is an important goal, so this topic is highly significant.

The premise of this new study stems from two previous articles by the same investigators, published in 2013 (*J Neurosci*) and 2017 (*Neuron*). Understanding the premise and conclusions of the new study requires understanding those two papers. In the 2013 study, the authors eliminated a glutamate transporter from a large extent of the LPBN, or selectively ablated glutamatergic neurons in this region, and found a marked reduction in the HCVR and arousal responses to elevated CO₂. Next, in the 2017 study, they focused on a separate group of neurons in the LPBN and proposed that CGRP neurons in the LPBN are important for hypercapnic arousal but not the HCVR. Finally, in the present study, they are advancing an elegant double-dissociation hypothesis by proposing that a separate, large population of FoxP2 neurons in this region of the brainstem is necessary for the breathing response but not for the corresponding arousal. The idea is elegant, but the experimental evidence did not support this conclusion.

Major critiques:

- The main problem is that the results do not directly or fully support the claim. The authors conclude that LPBN/KF FoxP2 neurons are necessary for the HCVR (lines 332-333: "required for normal respiratory response to CO₂ during sleep"; lines 340-341: "we interpret the PB-FoxP2 neurons as primarily mediating the ventilatory response to CO₂."), but their ArchT experiment (attempting to inhibit these neurons) had no effect on respiratory rate, and only a small effect on volume (conversely, photostimulating FoxP2 neurons produced an increase in breathing rate, but not volume – and no explanation is advanced for the discrepancy). Importantly, no attempt was made to simply ablate Foxp2-expressing neurons in LPBN; this is the most straightforward way to test the authors' hypothesis. Robust methods for ablating specific cell types have been in use for over a decade (Cre-dependent expression of diphtheria toxin A; Cre-dependent expression of modified caspase/phosphatase), and the authors were among the first to use one of these techniques (ablating glutamatergic neurons in the LPBN in 2013). Especially given the unconvincing results with ArchT inhibition, the authors must show what happens when they simply remove Foxp2-expressing neurons from the LPBN/KF region. These concerns are particularly important with HCVR, which is highly variable and state-dependent (high propensity for false-positive and false-negative conclusions).

- Also not supported in this manuscript, or in any previous studies, to my knowledge (apologies if I missed it, but it is not cited in this paper), was a claim made in the first sentence of the abstract: that activating CGRP neurons in LPBN "has little effect on respiration." The authors showed in a previous article that activating CGRP neurons increases arousal, but they did not show any respiratory measurements in those experiments (not that I could find in the 2017 Neuron paper). No such data are provided for CGRP neurons in the present study. The proposed double-dissociation (CGRP neurons mediate arousal & FoxP2 neurons mediate hyperventilation) depends heavily upon the authors' claim that CGRP neurons have "little effect on respiration." If this is not true (if stimulating CGRP neurons does in fact increase MV i.e. the respiratory rate and/or volume), such claims should be removed or heavily re-phrased at least.

- Further contradicting the claimed double-dissociation is the author's finding that stimulating FoxP2 neurons in the LPBN triggered arousal in 70% of their ChR2/photostimulation trials. The authors downplay this observation because the arousal happened with a bit of a delay, relative to the HCVR, but it undercuts the idea that FoxP2 neurons specifically trigger HCVR and not arousal. The lack of direct comparisons between arousal and HCVR latency after photostimulating CGRP vs. FoxP2 neurons (and the prior and current lack of information on HCVR after activating CGRP neurons) weakens the conclusions of this study.

- The bottom part of Supplemental Figure 2, and the entire Supplemental Figure 3 was missing. I tried opening or importing the Supplement PDF in several different applications and on multiple computers, with the same problem each time. I cannot complete this review without seeing the full paper.

- A major flaw in this study has to do with the non-specificity of recording and stimulating neurons that express the gene Foxp2. Anatomical work cited in this new study identified heterogeneous populations of FoxP2 neurons in and around the LPBN (including GABAergic neurons in the caudal KF, glutamatergic neurons in rostral KF, and several more glutamatergic populations in LPBN). Another paper, which just appeared in Neuron, identified LPBN Foxp2-expressing neurons that transmit chemical and mechanical itch (Ren et al. 2023). The authors do not go into detail about this complexity and heterogeneity, instead (Fig. 7) treating these aggregate FoxP2 populations as if they had a uniform function (HCVR). If the loss-of-function result had been compelling (see above), this would be less of a problem, but the results presented leave it unclear what concrete advance was made, relative to previous results.

Other critiques

- Insufficient attention is given to counterfactuals (negative-control considerations are lacking, or are

glossed over). For example, the number (and to a lesser extent, the full distribution) of neurons with a developmental history of Foxp2 expression is more than adult expression of this gene (and of adult FoxP2 immunofluorescence labeling). The authors use a Cre-reporter strategy (fate-mapping developmental expression plus subsequent/adult expression) as a surrogate for actual expression, and justify this with a blanket, qualitative statement about co-expression of the reporter with nuclear immunostaining, but they should at least count the numbers of FoxP2-immunoreactive and GFP-expressing neurons and give the actual numbers and proportions. This is worth pointing out because the authors made clear, black-and-white claims of “validation” and “eutopic” adult expression (lines 466, 1103).

- If the authors wish to claim that the two genes or their protein/peptide products are expressed together in adult neurons, they should co-label mRNA (for Foxp2 and Calca) or immunostain (FoxP2 and CGRP), or some combination of the two. Labeling mRNA in cells expressing a (developmental fate-mapping) Cre-reporter leaves it unclear whether and to what extent there is any adult co-expression, which is what the authors are claiming (lines 148-152).

- The relative functions of LPBN versus KF neurons in this study are not addressed. The development, connections, and functions of KF FoxP2 neurons appear to be quite different from FoxP2 neurons in the LPBN. The senior author’s previous work indicated that neurons in the KF location exert a timing-dependent disruption in the respiratory rhythm by prolonging the post-inspiratory phase (apneusis), and hyperventilatory responses in rats were triggered by neurons located ventrally. Now, however, it seems they are pinning hyperventilatory responses on neurons located dorsally (in the LPBN), without functionally re-mapping this region in mice. This seems like a step backwards and is likely to leave readers confused by the un-addressed discrepancies.

- I do not have expertise or experience with the long-latency hypercapnic responses of neurons in the LPBN/KF region. Therefore, I will not critique the calcium-imaging portion of the paper. However, if a 10-second delay in neuronal response is typical for this type of recording, that should be explained to the reader (presumably involving delayed changes in blood pH buffering? I would expect the polysynaptic neuronal responses to be fairly rapid, sub-second, once the RTN or carotid-body chemoreceptors are activated by even mild acidity).

- Lines 298-310 (and 334-335): the authors report “extensive” axonal projections from LPBN FoxP2 neurons to the medulla. In contrast, two papers they cited used the same strain of Foxp2 mice (and a different strain of CGRP mice) and reported that FoxP2 projections to the medulla was minimal and that CGRP neurons in LPBN project more widely in the medulla than FoxP2 neurons. How do the authors explain these discrepancies between their new work and the previous published anatomic studies? The retrograde tracing images shown in this study, like previous results in rodents (including the senior author’s previous results in rats) give the impression of very few LPBN neurons (in contrast to KF) projecting axons to respiratory nuclei in the medulla.

- Line 614-615: how can a primary antibody produce immunostaining if it is omitted? Are the authors instead referring to the secondaries (not mentioned in the preceding lines)?

- Line 740: “A probability of error of less than 0.05 was considered significant” – inaccurate description of p/α ; rephrase (alpha vs. beta error).

Minor / stylistic suggestions

- Lines 40-41 “second group of non-CGRP neurons” is confusing (also line 109). Grammatically, it implies that there was already first group of non-CGRP neurons (after the CGRP neurons). There are of course several groups of non-CGRP neurons in the LPBN, but after reading the paper it became apparent that the authors meant that the CGRP neurons were the first group and they are referring to “a second group of LPBN neurons” as one that lacks CGRP. I recommend re-phrasing to eliminate

ambiguity.

- Lines 45-46 "which has recently been found in this region." This is not a recent finding – it has been 15 years since Foxp2 expression was shown in the LPBN region ("Conservation and diversity of Foxp2 expression in muroid rodents: Functional implications" Campbell et al. J Comp Neurol 2008; also the Gray 2008 review cited by the authors), with many other papers using this LPBN marker in the interim.

- Reference 22 appears to be a conference abstract with contents redundant to the author's published work. Remove. (reference 18 also appears to be a conference abstract rather than a published, peer-reviewed reference; remove)

- Line 159 "FoxP2-Cre::L10" is a non-standard way of describing a Cre-reporter strain, but whatever abbreviation is used, I recommend keeping it consistent throughout the manuscript to avoid confusing readers. In at least two other spots, different terms were used: line 150 ("FoxP2-L10") and line 1095 ("FoxP2-cre-L10").

- Typo Line 218 "movwement"

- More specific criteria (proximity/distance) are needed (preferably with supplementary plots and example images), showing optic fiber targeting in ChR2 and ArchT experiments relative to transduced FoxP2 neurons. For example (lines 268-270), it would be helpful to see examples of successful vs. missed targeting that did or did not produce the reported effects. Most importantly, the reader should understand why some cases were excluded and others were kept. Was the post hoc decision to exclude a case made after reviewing the experimental results, or by someone blinded to the analysis?

- Lines 314-315: "ventral part of the PB" – it would be better to provide supplementary plots of these injection sites so the reader knows the exact distribution of neurons that were transduced in each case

- Line 351: "intersecting" is unnecessary here; also, consider adding "neurons" after "the PB-FoxP2"

- Line 460 & 463: "wildtype littermates" – C57 x Cre littermates are not "wildtype"

- Line 470 typo "Fox-P2"

- Line 509: typo "immune-staining"

- Line 514: "using square grids" – unclear; does this mean that images were tiled together?

- Line 1086: "to the phrenic motor [neurons?]" – there is no evidence in this study (or in any others that I could find) that FoxP2 LPBN or KF neurons project axons to the cervical spinal cord. It may be the case, but the authors should make it clear that this is speculative (or provide evidence).

- Line 1094-1101: the authors labeled Calca (mRNA transcript) yet referred to it as CGRP (peptide). This is confusing and should be simplified for both accuracy and clarity. Also, they used a Cre-reporter strategy (tags developmental+adult expression) yet referred to it as "FoxP2." They should co-label the mRNAs (Calca, Foxp2) and/or protein products (CGRP, FoxP2) as above, and should be more careful with terminology so that the reader can tell what was and wasn't labeled.

- Line 1117: as above, there was no Supplementary Figure 3 to evaluate.

- Figure1 : bar graphs should be replaced by showing the individual data points.

- Figures 4&6: replace the white bars, which obscure the underlying images with thin or dashed lines

that indicate possible locations of optic fiber tracts

- Figure 5: axonal labeling in panels b&c is highly over-contrasted, such that many axons look like blobs the diameter of cell bodies. Images should be replaced with less severe contrast/brightness settings.

Reviewer #3 (Remarks to the Author):

Kaur et al., perform a series of technically challenging experiments to investigate the role of FoxP2 expressing parabrachial neurons for respiration and arousal from sleep. Based on these experiments, and ones they have performed previously on neighboring CGRP neurons, they conclude that FoxP2 neurons are implicated in the respiratory response to CO₂ but not arousal, whereas CGRP neurons drive arousal, but not changes in respiration. For the most part, the data are compelling, but the strength of the conclusions could be significantly enhanced in a revised version with additional analysis and clarification.

Comments and suggestions:

Major:

Directly comparing the effects of CGRP and FoxP2 populations on breathing and arousal would provide much clearer insight into how functionally distinct these populations are.

FoxP2 neurons appear most active in wake and REM states. However, data showing how activation/inhibition of these neurons may affect breathing in wake/REM are not provided. Thus, it is unclear whether these neurons generally increase breathing (across states) or if they do indeed play a specific role in the respiratory response to CO₂ during NREM. Further, the fiberphotometry data (Supplementary Fig 3a), suggest that FoxP2 neuron activity may be correlated with specific types of breathing in the awake state, but no analysis is performed. This would be interesting compare with has been shown for other PB neuron populations (e.g. PMID: 34921781; PMID: 36810601).

It is also suggested that the activity of FoxP2 neurons correlates with movement (EMG), but no analysis is performed to confirm this. Cross-correlation analysis (or something similar) of FoxP2 activity vs EMG, and FoxP2 activity vs respiratory rate, would be beneficial.

The authors show CGRP immunohistochemistry in FoxP2 Cre mice to quantify overlap in the KF but not PB. Showing similar data for the PBcl/el to confirm separation of CGRP and FoxP2 neurons, i.e. that the neurons in question are indeed non-CGRP, would help support the conclusions.

It is clearly shown that FoxP2 neurons have projections to the preBötC area, but the functional role of these projections is not tested. This could provide important insight into the authors speculation that feedback from preBötC or increased respiratory effort, rather than the ascending projections of FoxP2 neurons, is what leads to arousal during activation of FoxP2 PB neurons.

The FoxP2 neuron activation and inhibition experiments are not complimentary – activation drives RR without much TV effect, whereas inhibition prevents TV increase but not changes in RR. A potential explanation for this is provided, suggesting that RR can't be reduced because the CO₂ stimulus activates multiple other areas that drive RR. This is plausible, but if this is the case, activation and inhibition of FoxP2 neurons in the absence of CO₂ should have complimentary (opposite) effects on RR and TV. However, this analysis is not shown. Moreover, based on the inhibition experiments (Fig 6g) it is concluded that FoxP2 neurons are necessary for increased Tv in response to CO₂. But, activation of these neurons does not increase Tv (Fig. 4f). Thus, since FoxP2 neurons do not appear to be sufficient to increase Tv, it is unclear how they can be necessary.

Minor:

Line 109: There is some evidence that CGRP neurons actually inhibit respiration, at least in the awake state (PMID: 32856589; PMID: 36810601), which may be worth clarifying here.

Line 148: It isn't definitively shown with this data that they are non CGRP.

Line 150: FoxP2-L10 mice should be defined here since this is the first mention of the FoxP2-cre strain.

Line 162: "All of the neurons transfected by an AAV that expressed Cre-dependent GCaMP6s (which also fluoresces green) also showed Foxp2 expression in their nuclei". This is only shown for the L10 reporter mouse, not the AAV experiments, so it is unclear to what extent the viral expression remained specific to FoxP2 neurons.

Fig 2b: zoomed image isn't taken from a region that will contribute to the majority of Delta F/F signal.

Line 172-174: "The increased activity of PB FoxP2 neurons when emerging from NREM sleep may contribute to the sudden increase in respiratory effort when an animal exposed to CO2 awakens from NREM sleep." This should be more clearly illustrated/analyzed. Supplementary fig 3b shows only a single example of this and it is unclear whether there is an associated change in RR. Additional analysis to better understand relationships between FoxP2 activity (deltaF/F) state transitions (NREM/REM/wake) and respiratory rate, would be important to support this speculation.

Line 175: It would be helpful to clarify here that these are only trials when the mouse was sleeping. Also as suggested above, it would be informative to show whether or not CO2 causes a similar activation of these neurons in the awake state. E.g. repeat Fig 2e when mice are in wake state and compare vs sleep.

Line 187-192: It's unclear how the last 5 breaths etc. aligns with the points on the graphs in Fig 2f-i, which appear to be time bins. Please clarify.

Line 209-213: Cross correlation analysis to better understand the relationship between EMG and FoxP2 neuron activity in awake state? This relationship is not obvious from the data shown in Fig S3f.

Line 217-218: "PB Foxp2 neurons are most active during wake and REM sleep, and particularly during active movement in wake, and may anticipate the need for increased ventilation during active movement." Too much speculation for Results section without sufficient analysis performed to test these ideas. I.e. cross correlation of EMG and FoxP2 activity to see if FoxP2 activation precedes movements as suggested.

Fig 3e: Averaged or summed deltaF/F should be shown. Although it is stated that similar oscillations weren't present in NREM (Line 245-248) this isn't apparent from the data shown. Additional analysis could help strengthen this point. And again, do these neurons also show responses to CO2 when mice are awake? Was the activity of these neurons recorded during normally behaving conditions in awake mice? Do single neurons show activity profiles similar to those observed in the fiber photometry experiments (i.g. Supplementary Fig 3A)?

Line 263-265: What about trials when mice were in wake state or REM? It would be interesting to know how FoxP2 neurons may affect breathing in these conditions as well.

Line 267: FoxP2 activation causes arousal from sleep – it's unclear how different this is from the effect of CGRP neurons?

Line 268-270: A supplemental Fig to show/quantify the included/excluded fiber placement locations would be helpful.

Line 274-283: It would be helpful to quantify instantaneous RR vs time during 5 and 10 sec stimulations to support the point the authors are trying to make (i.e. similar to Fig. 2h-i). 5 breath "bins" does not provide much resolution.

Line 289-292: Similar to the above, if there really is a different time course of the respiratory response in 2% CO2, this should be illustrated in a graph similar to Fig. 2h-i. As currently plotted the response to 20Hz, 10s stim looks identical in normocapnia vs. 2% CO2 (Compare Fig 4 f (blue) and 4g (purple)).

Line 297: Activation of these projections to the preBötC area would be a nice addition to demonstrate whether or not this is the relevant pathway.

Line 299: preBötC or PBC is more conventional notation for preBötzinger Complex than PBZ.

Line 321: How does inhibition of FoxP2 neurons affect breathing in awake and REM?

Line 326: Reference Fig 6i as well.

Fig 6 c2,c5: Scale bars seem mislabeled?

Line 331-333: The specific role of response to CO2 during sleep is not demonstrated because data testing the role of FoxP2 neurons in the awake state is not provided. Thus, these neurons may have a similar role in sleep and wake states.

Line 333-335: could this reflect the "core/shell" configuration of PB Oprm1 neurons as suggested by PMID: 34921781

Line 412: Photostimulation of inhibitory preBotC neurons generally slows RR (e.g. PMID: 25643296; PMID: 33670653; PMID: 33863785) unless done specifically during the inspiratory phase (PMID: 29483589) or under specific conditions (ref 38). The other reference provided (39) describes activation of excitatory neurons, which do consistently increase RR.

Other general comments:

It would strengthen the paper to be clear about whether FoxP2 neurons in PBcl, KF, or both were primarily targeted during optogenetic imaging and manipulation experiments, since these groups of neurons could have distinct functional roles.

In general, the figures could be improved by making the size and font of text consistent (some axis labels are very small, some are large, some text is bold, some not, etc.). There is also a lot of color used which can get somewhat confusing, and it is not always consistent. E.g. Co2 trace is green and deltaF/F is red in Fig 2d but CO2 is orange and deltaF/F is black in Fig 2g.

Parabrachial anatomy could be more clearly/consistently referenced in the text. E.g. PBL vs LPB vs PBel/PBcl

Response to reviewer's comments:

Reviewer #1

Summary of key results:

The authors identify a new subclass of neurons in the parabrachial nuclear complex, including the adjacent Kolliker-Fuse, which is involved in increasing respiratory drive during chemosensory challenge, i.e., elevated carbon dioxide (hypercapnia). This new subclass is identified by transcription factor FoxP2, rather than expression of peptide CGRP, which the authors previously identified. The CGRP population is involved in arousal, but not respiratory drive. The FoxP2 subclass participates in both respiratory drive and arousal. These FoxP2 neurons are associated with REM sleep and wakefulness.

The experiments are convincing and complement the authors former discovery of the CGRP population and this study does advance understanding.

There are a few aspects that dampen enthusiasm:

Major comments:

1) It is unclear how the FoxP2 neurons get chemosensory information. Their activity in response to hypercapnia comes at a delay, and the latency to activation is heterogeneous (Figure 3d) so it is unlikely intrinsic chemosensitivity is at work. It seems likely that these neurons are connected to central chemosensory neurons in the retrotrapezoid nucleus / parafacial ventral (RTN/pFV). The authors should discuss how FoxP2 parabrachial / Kolliker-Fuse neurons receive chemosensory information.

Response: *This is shown in Fig 8 and discussed in the second to last paragraph of the discussion.*

2) The figures are difficult to understand in many respects. Figure 1 uses bar plots with standard error, rather than showing all the data points and plotting standard deviation or confidence intervals, which have much greater fealty to the raw data and give the reader more evidence of effect size. Figure 2 uses inconsistent tick and typeface size, which make it extremely hard to gauge effect sizes. Figure 3 has similar problems regarding ticks and inconsistent sizes of typefaces. Panels b and c in Figure 3 are actual screen shots that cannot be read with any accuracy, straining the reader. Figure 3 panels b and c show the same 3 neurons and basically the same time series. Certainly, those data can be taken from acquisition and analysis software and made into a presentable figure in a single panel. Figure 4 has the same problems with inconsistent typeface and ticks. Figure 5 is a patchwork of panels that contain valuable information but are packed so tightly together with all labels within the panels, again, straining the reader. Figure 6i is the type of "show the data" plot I would recommend for Figure 1e and f, but here a standard error or confidence interval would be preferable. Also, here the typeface in Figure 6b would be nearly unreadable in typical PDF size for reading (the reader will have to expand the illustration to get the information.)

Response: *We have gone through the figures to make them more consistent in the use of typefaces and easier for the reader to digest. In Fig 1 we added data points in all bar plots. The two parts of Fig 3 have now been combined into a single time series.*

3) The written manuscript overall suffers haphazardness. Time is sometimes "s" and other times "sec". Sometimes the units are separated from numerals, e.g., "20 Hz" and sometimes not "20Hz". Figure callouts are inconsistent too: "Fig. 1" or "Fig.1" or "Fig 1". These inconsistencies are rampant. While this critique does not have any bearing on scientific content, it does not, in my view, show attention to detail.

Response: *We have gone over the manuscript carefully to be as consistent as possible in terminology and typography.*

Minor comments:

Lines 37, 107: CGRP should be defined.

Response: *We have made this change.*

Line 91: how can ventilation increase during an apnea, when apnea -- by definition -- is a pause in breathing (no ventilation).

Response: *We have now rephrased this.*

Figure 1e & f: the bar charts should be replotted to show all the data points as well as either 95% confidence intervals or standard deviation rather than standard error of the mean, to show the data (see: Drummond GB, Vowler SL. Show the data, don't conceal them. *J Physiol (Lond)* 589:1861–1863, 2011)

Response: *We added the requested datapoints. We chose to leave the error bars as SEM, as the datapoints now indicate the variability in the data.*

Line 141: the number of the figure is missing.

Response: *We have added this now.*

Figure 2d, f-i: the y-axes are labeled with too many and often small ticks and too small type face. It is difficult to read these axes and make sense of the data. The x-axes are inconsistently labeled with different size type faces and time is "s" or "sec". The graphics should be more elegant and easier to read (see: Tufte ER. *The visual display of quantitative information*, 2nd ed. Cheshire, Conn: Graphics Press, 2001.)

Response: *We are now keeping them consistent and similar.*

Lines 180-204: The figure callouts are sloppily labeled, sometimes missing a period (180, 181, 190) and other times lacking a gap between the period and the numeral (192, 198, 204). While these simple typos have no bearing on scientific content, the sloppiness is disturbing for its excessive frequency.

Response: *We apologize for this sloppiness and have made corrections.*

Figure 4e: Should be "minute ventilation" not "minute volume".

Response: *We have made this correction.*

Line 209: missing "P="

Response: *We have corrected this.*

Line 218: "movement" is misspelled.

Response: *This is now corrected.*

Line 242: I think the callout should be to Fig. 3e (not 3f).

Response: *corrected.*

Line 302: "CTb" should be defined in the main text when it appears. It is defined in the figure legend, but one expects a definition in the main narrative too.

Response: *We have defined CTb.*

Reviewer #2:

Summary of key results:

This study evaluates whether Foxp2-expressing neurons in the lateral parabrachial nucleus (LPBN) or nearby Kölliker-Fuse nucleus (KF) play a role in the hypercapnic ventilatory response (HCVR) in mice. HCVR is the increased breathing rate and volume triggered reflexively in response to increased CO₂ (and the resulting drop in blood pH). HCVR is important for survival, particularly in patients with obstructive sleep apnea, in whom this reflex (and a parallel surge in arousal) kick-starts breathing after prolonged apneas. Understanding the neurologic mechanisms of HCVR is an important goal, so this topic is highly significant.

The premise of this new study stems from two previous articles by the same investigators, published in 2013 (J Neurosci) and 2017 (Neuron). Understanding the premise and conclusions of the new study requires understanding those two papers. In the 2013 study, the authors eliminated a glutamate transporter from a large extent of the LPBN, or selectively ablated glutamatergic neurons in this region, and found a marked reduction in the HCVR and arousal responses to elevated CO₂. Next, in the 2017 study, they focused on a separate group of neurons in the LPBN and proposed that CGRP neurons in the LPBN are important for hypercapnic arousal but not the HCVR. Finally, in the present study, they are advancing an elegant double-dissociation hypothesis by proposing that a separate, large population of FoxP2 neurons in this region of the brainstem is necessary for the breathing response but not for the corresponding arousal. The idea is elegant, but the experimental evidence did not support this conclusion.

Major critiques:

1) The main problem is that the results do not directly or fully support the claim. The authors conclude that LPBN/KF FoxP2 neurons are necessary for the HCVR (lines 332-333: “required for normal respiratory response to CO₂ during sleep”; lines 340-341: “we interpret the PB-FoxP2 neurons as primarily mediating the ventilatory response to CO₂.”), but their ArchT experiment (attempting to inhibit these neurons) had no effect on respiratory rate, and only a small effect on volume (conversely, photostimulating FoxP2 neurons produced an increase in breathing rate, but not volume – and no explanation is advanced for the discrepancy). Importantly, no attempt was made to simply ablate Foxp2-expressing neurons in LPBN; this is the most straightforward way to test the authors’ hypothesis. Robust methods for ablating specific cell types have been in use for over a decade (Cre-dependent expression of diphtheria toxin A; Cre-dependent expression of modified caspase/phosphatase), and the authors were among the first to use one of these techniques (ablating glutamatergic neurons in the LPBN in 2013). Especially given the unconvincing results with ArchT inhibition, the authors must show what happens when they simply remove Foxp2-expressing neurons from the LPBN/KF region. These concerns are particularly important with HCVR, which is highly variable and state-dependent (high propensity for false-positive and false-negative conclusions).

Response: *At the advice of the reviewer, we have now genetically ablated both PBcl and KF FoxP2 populations and reported the results as Fig 6 and in the result section. The results are consistent with, but more profound than the results of opto-inhibition. They show that the ablation of the PBcl/KF^{FoxP2} neurons dramatically reduce the ventilatory response to CO₂, without affecting the latency to arousal.*

2. - Also not supported in this manuscript, or in any previous studies, to my knowledge (apologies if I missed it, but it is not cited in this paper), was a claim made in the first sentence of the abstract: that activating CGRP neurons in LPBN “has little effect on respiration.” The authors showed in a previous article that activating CGRP neurons increases arousal, but they did not show any respiratory measurements in those experiments (not that I could find in the 2017 Neuron paper). No such data are provided for CGRP neurons in the present study. The

proposed double-dissociation (CGRP neurons mediate arousal & FoxP2 neurons mediate hyperventilation) depends heavily upon the authors' claim that CGRP neurons have "little effect on respiration." If this is not true (if stimulating CGRP neurons does in fact increase MV i.e. the respiratory rate and/or volume), such claims should be removed or heavily re-phrased at least.

Response: *The reviewer is correct that we mis-stated this point. In our 2017 Neurons paper (PMID- 29103805), we reported that upon optogenetic stimulation of the PBe^{CGRP} neurons, animals had almost immediate arousal (SFig 1 of the 2017 publication) with average latencies less than 2s in almost all trials. Because respiration increases almost immediately with arousal for any cause, this short interval before EEG arousal makes it difficult to reliably assess the independent effects of stimulation of PBe^{CGRP} neurons on respiration. However, we did show that optogenetic inhibition of these neurons had no effects on the increase in ventilation to hypercapnia (SFig 4, of Kaur et al., 2017). Thus, the corrected sentence, in both the Abstract and Introduction, is that "inhibition of glutamatergic neurons expressing calcitonin gene-related peptide (CGRP) in the external lateral PB (PBe^{CGRP} neurons) prevents cortical arousal in response to hypercapnia, but has little if any effect on the respiratory response." We maintain that the lack of effect of photoinhibition of the PBe^{CGRP} neurons on respiratory response to CO₂ and the lack of effect of ablation of the PB+KF^{FoxP2} neurons on time to arousal to CO₂ establishes the double-dissociation.*

3. Further contradicting the claimed double-dissociation is the author's finding that stimulating FoxP2 neurons in the LPBN triggered arousal in 70% of their ChR2/photostimulation trials. The authors downplay this observation because the arousal happened with a bit of a delay, relative to the HCVR, but it undercuts the idea that FoxP2 neurons specifically trigger HCVR and not arousal. The lack of direct comparisons between arousal and HCVR latency after photostimulating CGRP vs. FoxP2 neurons (and the prior and current lack of information on HCVR after activating CGRP neurons) weakens the conclusions of this study.

Response: *As we point out in the text, the animals do not arouse at all in 30% of the trials even with vigorous optogenetic stimulation of the PB^{FoxP2} neurons that increases RR by 50%, and that when the animals do arouse, it is often after the optostimulation has been terminated (the average latency to arousal in Fig 4d is consistently greater than the duration of stimulation). This compares to arousal within a mean of 2 sec on every trial when the PBe^{CGRP} neurons were stimulated (see Kaur et al., 2017, Fig S1F). Conversely, as indicated in the Response above, the ablation/inhibition of the two neuronal populations yields a crystal-clear double-dissociation.*

4. The bottom part of Supplemental Figure 2, and the entire Supplemental Figure 3 was missing. I tried opening or importing the Supplement PDF in several different applications and on multiple computers, with the same problem each time. I cannot complete this review without seeing the full paper.

Response: *We apologize for this. Please let us know if this happens again, we can provide you with the full resolution images.*

5. A major flaw in this study has to do with the non-specificity of recording and stimulating neurons that express the gene Foxp2. Anatomical work cited in this new study identified heterogeneous populations of FoxP2 neurons in and around the LPBN (including GABAergic neurons in the caudal KF, glutamatergic neurons in rostral KF, and several more glutamatergic populations in LPBN). Another paper, which just appeared in Neuron, identified LPBN Foxp2-expressing neurons that transmit chemical and mechanical itch (Ren et al. 2023). The authors do not go into detail about this complexity and heterogeneity, instead (Fig. 7) treating these aggregate FoxP2 populations as if they had a uniform function (HCVR). If the loss-of-function result had been compelling (see above), this would be less of a problem, but the results

presented leave it unclear what concrete advance was made, relative to previous results.

Response: *The reviewer is correct that there are several different, and anatomically distinct clusters of FoxP2 neurons in the parabrachial complex. However, as we show in Fig. 1, the PB neurons that express cFos after hypercapnia are in the KF and PBcl, and bridging between them along the lateral margin of the PBel^{CGRP} population, and the overwhelming majority of them express FoxP2. On the other hand, the FoxP2 neurons identified by Ren et al. (Neuron 111:1812, 2023) are clearly identified as being in the dorsal part of the lateral PB. For this reason, in the revised manuscript we have more clearly identified the neurons we are dealing with as being the PBcl^{FoxP2} and KF^{FoxP2} clusters.*

Minor critiques

1. Insufficient attention is given to counterfactuals (negative-control considerations are lacking, or are glossed over). For example, the number (and to a lesser extent, the full distribution) of neurons with a developmental history of Foxp2 expression is more than adult expression of this gene (and of adult FoxP2 immunofluorescence labeling). The authors use a Cre-reporter strategy (fate-mapping developmental expression plus subsequent/adult expression) as a surrogate for actual expression, and justify this with a blanket, qualitative statement about co-expression of the reporter with nuclear immunostaining, but they should at least count the numbers of FoxP2-immunoreactive and GFP-expressing neurons and give the actual numbers and proportions. This is worth pointing out because the authors made clear, black-and-white claims of “validation” and “eutopic” adult expression (lines 466, 1103).

Response: *We have now quantified the eutopic expression and have included that in second paragraph of Results section. We also show in our Figs. 2, 4b, 5a, 7c, and S2 that the Cre expression in our FoxP2-Cre mice (as demonstrated by Cre-dependent expression of GCaMP, ChR2-mCherry, ArchT-GFP, and L10-GFP, respectively) is virtually 100% eutopic with the immunohistochemical staining for FoxP2 protein.*

2. If the authors wish to claim that the two genes or their protein/peptide products are expressed together in adult neurons, they should co-label mRNA (for Foxp2 and Calca) or immunostain (FoxP2 and CGRP), or some combination of the two. Labeling mRNA in cells expressing a (developmental fate-mapping) Cre-reporter leaves it unclear whether and to what extent there is any adult co-expression, which is what the authors are claiming (lines 148-152).

Response: *This is provided in Sfig. 1, which combines in situ hybridization for Calca mRNA with FoxP2-L10-GFP expression, which we show in Sfig2. is 100% eutopic with immunohistochemistry for the gene product in adult animals.*

3. The relative functions of LPBN versus KF neurons in this study are not addressed. The development, connections, and functions of KF FoxP2 neurons appear to be quite different from FoxP2 neurons in the LPBN. The senior author’s previous work indicated that neurons in the KF location exert a timing-dependent disruption in the respiratory rhythm by prolonging the post-inspiratory phase (apneusis), and hyperventilatory responses in rats were triggered by neurons located ventrally. Now, however, it seems they are pinning hyperventilatory responses on neurons located dorsally (in the LPBN), without functionally re-mapping this region in mice. This seems like a step backwards and is likely to leave readers confused by the un-addressed discrepancies.

Response: *The previous studies on using microinjections of glutamate to stimulate PB neurons were done by Chamberlin and Saper in rats, in which the anatomy of the neurons that project to respiratory sites in the medulla is quite different. Such neurons are located primarily in the KF and the lateral crescent (along the lateral margin of PBel) in rats, and those were the locations*

in which injections of glutamate were most effective at eliciting increased respiration. The reviewer is incorrect about the location of the apneustic neurons: they were consistently ventral to the bulk of the sites that increased respiration, and mixed with hyperpneic sites in the KF. In mice, by comparison, there is a large population of PBcl^{FoxP2} neurons that project to the respiratory areas of the medulla. We have not encountered an apneustic response from photostimulation in mice, but we have only done optogenetic stimulation of FoxP2 and CGRP neurons, so it is possible that another cell type may cause apneusis when stimulated, or this could be a species difference. We hope that our spatial transcriptomic map of the PB region will allow us (or another group) to identify a different neuron type in the PB that may produce that response.

With respect to potential differences in function between different subgroups of FoxP2 neurons in the PBcl and KF, our data in this paper show that opto-inhibition of the PBcl^{FoxP2} neurons has a very similar effect to ablation of both populations in terms of preventing a CO₂-induced increase in minute ventilation, although the ablation effect is stronger. We do not have data to speculate beyond that.

4. I do not have expertise or experience with the long-latency hypercapnic responses of neurons in the LPBN/KF region. Therefore, I will not critique the calcium-imaging portion of the paper. However, if a 10-second delay in neuronal response is typical for this type of recording, that should be explained to the reader (presumably involving delayed changes in blood pH buffering? I would expect the polysynaptic neuronal responses to be fairly rapid, sub-second, once the RTN or carotid-body chemoreceptors are activated by even mild acidity).

Response: *We have now discussed this in the results section. We performed in vivo recordings in intact mice, which were placed in a plethysmograph, where exchange of gases is gradual (see graphs in Figs. 2d and g). It also takes time for changes in alveolar gas mixtures to reach the arterial circulation (a few seconds). For this reason, we have aligned the changes in respiration to that of CO₂ in the chamber and $\Delta F/F$ (Fig. 2 and 3). We found that “the rise in RR begins at about 5s (approximate time that it takes for CO₂ levels to rise in the chamber and then reach the circulation)”. This rise in RR is preceded by the rise in the $\Delta F/F$. This is now confirmed by the cross-correlation analysis as well (SFig 5).*

5. Lines 298-310 (and 334-335): the authors report “extensive” axonal projections from LPBN FoxP2 neurons to the medulla. In contrast, two papers they cited used the same strain of Foxp2 mice (and a different strain of CGRP mice) and reported that FoxP2 projections to the medulla was minimal and that CGRP neurons in LPBN project more widely in the medulla than FoxP2 neurons. How do the authors explain these discrepancies between their new work and the previous published anatomic studies? The retrograde tracing images shown in this study, like previous results in rodents (including the senior author’s previous results in rats) give the impression of very few LPBN neurons (in contrast to KF) projecting axons to respiratory nuclei in the medulla.

Response: *The reviewer raises an interesting question. We chose not to address this issue in the paper, because the answer is a bit complicated and beyond the scope of this manuscript. But we will answer it here, and if the editor wishes, we can add this discussion to the paper. For FoxP2 neurons in the PB, as we point out above there are several other FoxP2 populations in the dorsal part of the PB. The injections by Geerling and colleagues (Huang et al., JCN 2021 529:657) were targeted toward the more dorsal and rostral temperature sensitive neurons that he was studying. These neurons project primarily to the hypothalamus. They did include some of the PBcl neurons, though, and they did find the projection to the ventrolateral medulla. It is difficult to compare the intensity of staining across experiments, as this depends upon the number of neurons that were labeled that contribute to a projection. As we found in our study (Fig 5), if the injection targets the PBcl FoxP2 neurons, you see quite a robust projection to the*

medullary respiratory areas (in fact, the reviewer, below (last minor point) complained that the projection was so heavily labeled that they could not make out the details of fibers and terminals, causing us to have to reduce the brightness and contrast of the images). We now report in the Results section that over 70% of neurons in the PB and KF that were retrogradely labeled from the pre-Botzinger complex were also FoxP2 positive.

With respect to the CGRP neurons, in our studies we used a Calca-Ires-Cre-ER line, which produces moderate amounts of Cre recombinase. When we inject those mice with anterograde tracer vectors, we see robust forebrain projections, but little if any projection to the medulla. Geerling (Huang et al JCN 2021 529:2911) used Richard Palmiter's Calca-Cre knockin mouse, which has one chromosome with no Calca gene (it is replaced by Cre). This hemizygote animal for Calca has to increase the drive to its one remaining Calca gene to get sufficient CGRP protein expression. This then drives the other allele, which now produces profuse amounts of Cre, instead of Calca. When they injected those mice with anterograde tracer vectors, they did see the medullary projection (which was relatively modest compared to the forebrain projection). The reason for that disparity became apparent when we did our spatial transcriptomic atlas of the PB region. We found that there are two populations of CGRP neurons in the PB region. The large PBel CGRP neurons express Calca mRNA at a high level, and project to the forebrain. A second population of smaller Calca cells turned up in the KF and these express much lower levels of Calca (as well as FoxP2), and project to the medulla. The small, low-expressor neurons were not labeled by Cre dependent vectors in our CGRP-ires-Cre-ER animals.

6. Line 614-615: how can a primary antibody produce immunostaining if it is omitted? Are the authors instead referring to the secondaries (not mentioned in the preceding lines)?

Response: We have now rephrased this.

7. Line 740: "A probability of error of less than 0.05 was considered significant" – inaccurate description of p/alpha; rephrase (alpha vs. beta error).

Response: We have now rephrased this.

Minor

- Lines 40-41 "second group of non-CGRP neurons" is confusing (also line 109). Grammatically, it implies that there was already first group of non-CGRP neurons (after the CGRP neurons). There are of course several groups of non-CGRP neurons in the LPBN, but after reading the paper it became apparent that the authors meant that the CGRP neurons were the first group and they are referring to "a second group of LPBN neurons" as one that lacks CGRP. I recommend re-phrasing to eliminate ambiguity.

Response: We have now rephrased this.

- Lines 45-46 "which has recently been found in this region." This is not a recent finding –it has been 15 years since Foxp2 expression was shown in the LPBN region ("Conservation and diversity of Foxp2 expression in muroid rodents: Functional implications" Campbell et al. J Comp Neurol 2008; also the Gray 2008 review cited by the authors), with many other papers using this LPBN marker in the interim.

Response: We have now rephrased this.

- Reference 22 appears to be a conference abstract with contents redundant to the author's published work. Remove. (reference 18 also appears to be a conference abstract rather than a published, peer-reviewed reference; remove)

Response: We have removed this.

- Line 159 “FoxP2-Cre::L10” is a non-standard way of describing a Cre-reporter strain, but whatever abbreviation is used, I recommend keeping it consistent throughout the manuscript to avoid confusing readers. In at least two other spots, different terms were used: line 150 (“FoxP2-L10”) and line 1095 (“FoxP2-cre-L10”).

Response: *We have corrected this.*

- Typo Line 218 “movvement”

Response: *We have corrected this.*

- More specific criteria (proximity/distance) are needed (preferably with supplementary plots and example images), showing optic fiber targeting in ChR2 and ArchT experiments relative to transduced FoxP2 neurons. For example (lines 268-270), it would be helpful to see examples of successful vs. missed targeting that did or did not produce the reported effects. Most importantly, the reader should understand why some cases were excluded and others were kept. Was the post hoc decision to exclude a case made after reviewing the experimental results, or by someone blinded to the analysis?

Response: *We have now included this as a supplementary figure (SFig 6a) which shows the transfection area of ChR2 and the site of optical fibers in relation to transfection both in off target (description in result section) and in on-target cases. Fig 7 also now includes a control group of wildtype cases injected with ArchT. In all cases, the histology was reviewed by an investigator who was blinded to the physiological result, and cases with off-target placement of optical fibers or injections were considered to be anatomical controls.*

- Lines 314-315: “ventral part of the PB” – it would be better to provide supplementary plots of these injection sites so the reader knows the exact distribution of neurons that were transduced in each case.

Response: *We have now included this as a supplementary figure (SFig. 6a) which shows the transfection area of ChR2 and the site of optical fibers are plotted in relation to the transfection in both the on and off target cases (description in result section).*

- Line 351: “intersecting” is unnecessary here; also, consider adding “neurons” after “the PB-FoxP2”

Response: *We have corrected this.*

- Line 460 & 463: “wildtype littermates” – C57 x Cre littermates are not “wildtype”

Response: *We have rephrased this.*

- Line 470 typo “Fox-P2”

Response: *We have corrected this.*

- Line 509: typo “immune-staining”

Response: *We have corrected this.*

- Line 514: “using square grids” – unclear; does this mean that images were tiled together?

Response: *We have now corrected this description.*

- Line 1086: “to the phrenic motor [neurons?]” – there is no evidence in this study (or in any others that I could find) that FoxP2 LPBN or KF neurons project axons to the cervical spinal cord. It may be the case, but the authors should make it clear that this is speculative (or provide evidence).

Response: *We have rephrased this.*

- Line 1094-1101: the authors labeled Calca (mRNA transcript) yet referred to it as CGRP (peptide). This is confusing and should be simplified for both accuracy and clarity. Also, they used a Cre-reporter strategy (tags developmental+adult expression) yet referred to it as "FoxP2." They should co-label the mRNAs (Calca, Foxp2) and/or protein products (CGRP, FoxP2) as above, and should be more careful with terminology so that the reader can tell what was and wasn't labeled.

Response: *We have gone back through the text to ensure that we use Calca when referring to the gene or mRNA and CGRP when referring to its peptide gene product. As we indicate above, we have done the studies to demonstrate that essentially all of the neurons in our FoxP2-Cre mice that are labeled by either L10-GFP, a developmental Cre-dependent reporter (SFig. 2) or by injection of a Cre-dependent vector (Figs. 2,4, and 5) in neurons that also express FoxP2 as shown by immunohistochemistry for the gene product.*

- Line 1117: as above, there was no Supplementary Figure 3 to evaluate.

Response: *We apologize for that. If the referee cannot find any of our figures or needs higher resolution versions, please contact the Editor, and we will provide them.*

- Figure1: bar graphs should be replaced by showing the individual data points.

Response: *We have provided both the dot plot and bars in this version.*

- Figures 4&6: replace the white bars, which obscure the underlying images with thin or dashed lines that indicate possible locations of optic fiber tracts.

Response: *We have now done this.*

- Figure 5: axonal labeling in panels b&c is highly over-contrasted, such that many axons look like blobs the diameter of cell bodies. Images should be replaced with less severe contrast/brightness settings.

Response: *We adjusted the contrast to better visualize individual fibers, and now also include images showing the magnified parts of the areas shown in b and c (Fig 5) to clarify that ChR2 in the projection sites only labels axon fibers and terminals and not cell bodies.*

Reviewer #3:

Summary of key results:

Kaur et al., perform a series of technically challenging experiments to investigate the role of FoxP2 expressing parabrachial neurons for respiration and arousal from sleep. Based on these experiments, and ones they have performed previously on neighboring CGRP neurons, they conclude that FoxP2 neurons are implicated in the respiratory response to CO₂ but not arousal, whereas CGRP neurons drive arousal, but not changes in respiration. For the most part, the data are compelling, but the strength of the conclusions could be significantly enhanced in a revised version with additional analysis and clarification.

Comments and suggestions:

Major:

1. Directly comparing the effects of CGRP and FoxP2 populations on breathing and arousal would provide much clearer insight into how functionally distinct these populations are. FoxP2 neurons appear most active in wake and REM states. However, data showing how activation/inhibition of these neurons may affect breathing in wake/REM are not provided. Thus, it is unclear whether these neurons generally increase breathing (across states) or if they do indeed play a specific role in the respiratory response to CO₂ during NREM. Further, the fiber photometry data (Supplementary Fig 3a), suggest that FoxP2 neuron activity may be correlated

with specific types of breathing in the awake state, but no analysis is performed. This would be interesting compare with has been shown for other PB neuron populations (e.g. PMID: 34921781; PMID: 36810601).

It is also suggested that the activity of FoxP2 neurons correlates with movement (EMG), but no analysis is performed to confirm this. Cross-correlation analysis (or something similar) of FoxP2 activity vs EMG, and FoxP2 activity vs respiratory rate, would be beneficial.

Response: *The reviewer raises several good points. With respect to comparison of the role of FoxP2 neurons with CGRP neurons in the PB region, we published our results with the CGRP neurons in 2017, and so it would be duplicative to republish them here. Thus comparisons are done in the Discussion, by comparing our two datasets.*

With respect to the request to examine the effects of photo-stimulation of the PBcl^{FoxP2} neurons during wake, that is now provided in SFig. 7a-f. Stimulation had very little if any effect during wake, when these neurons are apparently already maximally active.

With respect to photo-inhibition, we now indicate in the Results that “photoinhibition had no significant effects on respiration during either wake or REM sleep states.” (last sentence, third paragraph) In other words the PBcl^{FoxP2} neurons are part of the brainstem circuit that increases respiration in response to elevated CO₂ during NREM sleep, but that during wake (or REM) forebrain circuits essentially override this homeostatic brainstem system to provide ventilation consistent with behavior (or in REM, perhaps imagined behavior).

We thank the referee for suggesting cross-correlation analysis, which is now included in S Figs. 4 and 5. These show that activation of the FoxP2 neurons follows EEG arousal and the associated EMG activity by about 1 sec, indicating that these neurons are activated by EEG arousal, rather than causing it. By contrast, activation of PBcl^{FoxP2} neurons occurs about 0.5 sec prior to increased RR.

2. The authors show CGRP immunohistochemistry in FoxP2 Cre mice to quantify overlap in the KF but not PB. Showing similar data for the PBcl/el to confirm separation of CGRP and FoxP2 neurons, i.e. that the neurons in question are indeed non-CGRP, would help support the conclusions.

Response: *Huang et al., 2021 (PMID-33715169) clearly showed that “CGRP and FoxP2 immunoreactivity identified mutually exclusive populations of neurons” (their p. 2917). We have also confirmed this observation in our experiments, and in our new spatial transcriptomic atlas, we found FoxP2 expression only by the small neurons in the KF that express low levels of Calca mRNA. We have added this to the first paragraph of the results.*

3. It is clearly shown that FoxP2 neurons have projections to the preBötC area, but the functional role of these projections is not tested. This could provide important insight into the authors speculation that feedback from preBötC or increased respiratory effort, rather than the ascending projections of FoxP2 neurons, is what leads to arousal during activation of FoxP2 PB neurons.

Response: *We agree that the role of the different medullary projections of the FoxP2 neurons is indeed an interesting problem. However, it would require a massive study, using activation and inhibition of the PB-KF^{FoxP2} terminals in at least four regions of the medulla, in wake and sleep states, with and without CO₂, to work this out. This would take at least two years to accomplish, and is far beyond the scope of the current study.*

4. The FoxP2 neuron activation and inhibition experiments are not complimentary – activation drives RR without much TV effect, whereas inhibition prevents TV increase but not changes in RR. A potential explanation for this is provided, suggesting that RR can't be reduced because the CO₂ stimulus activates multiple other areas that drive RR. This is plausible, but if this is the case, activation and inhibition of FoxP2 neurons in the absence of CO₂ should have

complimentary (opposite) effects on RR and TV. However, this analysis is not shown. Moreover, based on the inhibition experiments (Fig 6g) it is concluded that FoxP2 neurons are necessary for increased Tv in response to CO₂. But, activation of these neurons does not increase Tv (Fig. 4f). Thus, since FoxP2 neurons do not appear to be sufficient to increase Tv, it is unclear how they can be necessary.

Response: We now provide an extensive section on this issue in the Discussion (next to last paragraph). The difference between the photo-stimulation and photo-inhibition experiments is that the stimulation was done for 5 or 10 sec while the animal was breathing room air and the inhibition was done during a 30 sec interval while the animal was breathing 10% CO₂. The increase in RR occurred almost immediately during photo-stimulation (Fig. 4). During CO₂ exposure, the increase in RR is also very rapid, but the increase in Vt lags by 5-10 sec (Fig. 2), which may be why we did not see a significant rise in Vt during the 5-10 sec stimulation periods. On the other hand, when the animal is breathing 10% CO₂, the remaining respiratory circuitry is capable of increasing RR to near maximal levels without the PB^{FoxP2} neurons, but the latter are required to increase Vt (Fig. 6). Thus, under different conditions, the PB^{FoxP2} neurons contribute both to the RR and Vt response to CO₂.

Minor:

Line 109: There is some evidence that CGRP neurons actually inhibit respiration, at least in the awake state (PMID: 32856589; PMID: 36810601), which may be worth clarifying here.

Response: It is hard to know what to do with these observations based on opto-stimulation studies, as CGRP neurons cause behavioral arousal, and behavior has enormous secondary effects on respiration. That is why we focused on the LACK of an effect on CO₂-induced respiratory response when the CGRP neurons are inhibited (i.e., they are not needed for that response). Nevertheless, we have now included these references.

Line 148: It isn't definitively shown with this data that they (FoxP2 neurons) are non CGRP.

Response: Please see our answer to this question under Major point 2 above.

Line 150: FoxP2-L10 mice should be defined here since this is the first mention of the FoxP2-cre strain.

Response: We have now defined them in this line.

Line 162: "All of the neurons transfected by an AAV that expressed Cre-dependent GCaMP6s (which also fluoresces green) also showed Foxp2 expression in their nuclei". This is only shown for the L10 reporter mouse, not the AAV experiments, so it is unclear to what extent the viral expression remained specific to FoxP2 neurons.

Response: Please see the answer to Minor critique #1 by the Second Reviewer. We show in our Figs. 2, 4b, 5a, 7c, and S2 that the Cre expression in our FoxP2-Cre mice (as demonstrated by Cre-dependent expression of GCaMP, ChR2-mCherry, ArchT-GFP, and L10-GFP, respectively) is virtually 100% eutopic with the immunohistochemical staining for FoxP2 protein.

Fig 2b: zoomed image isn't taken from a region that will contribute to the majority of Delta F/F signal.

Response: We have now corrected this.

Line 172-174: "The increased activity of PBFoxP2 neurons when emerging from NREM sleep may contribute to the sudden increase in respiratory effort when an animal exposed to CO₂ awakens from NREM sleep." This should be more clearly illustrated/analyzed. Supplementary fig 3b shows only a single example of this and it is unclear whether there is an associated change in RR. Additional analysis to better understand relationships between FoxP2 activity

($\Delta F/F$) state transitions (NREM/REM/wake) and respiratory rate, would be important to support this speculation.

Response: *We agree and thank the reviewer for this suggestion. S Figs. 4 and 5 now show cross correlation analysis of $\Delta F/F$ for the $PBcl^{FoxP2}$ neurons with EEG and EMG associated with Wake, NREM, REM sleep and increase in RR.*

Line 175: It would be helpful to clarify here that these are only trials when the mouse was sleeping. Also as suggested above, it would be informative to show whether or not CO₂ causes a similar activation of these neurons in the awake state. E.g. repeat Fig 2e when mice are in wake state and compare vs sleep.

Response: *We have now provided the cross-correlation analysis of neuronal activity with EEG (EEGγ- representing arousal) and EMG (representing wake state) in the supplementary figures (S Fig 4 and 5).*

Line 187-192: It's unclear how the last 5 breaths etc. aligns with the points on the graphs in Fig 2f-i, which appear to be time bins. Please clarify.

Response: *Fig. 2 f-i shows the responses of neurons during the first 15 sec after onset of CO₂, when the animals were still asleep. The last 5 breaths before awakening therefore were after the interval shown in Fig. 2f-i. A typical example of this is shown in Fig. 2d where the animal awakens about 23 sec after onset of CO₂. We have now clarified this in the text.*

Line 209-213: Cross correlation analysis to better understand the relationship between EMG and FoxP2 neuron activity in awake state? This relationship is not obvious from the data shown in Fig S3f.

Response: *We agree and thank the reviewer for this suggestion. S Figs. 4 and 5 now show cross correlation analysis of $\Delta F/F$ for the $PBcl^{FoxP2}$ neurons with EEG and EMG associated with arousal and increase in RR. Note that onset of EEG arousal leads the increase in $\Delta F/F$ while increase in RR during NREM sleep follows activation of the $PBcl^{FoxP2}$ neurons. EMG activation which accompanies arousal also leads $\Delta F/F$ except during REM sleep, when there is no EMG signal.*

Line 217-218: "PB Foxp2 neurons are most active during wake and REM sleep, and particularly during active movement in wake, and may anticipate the need for increased ventilation during active movement." Too much speculation for Results section without sufficient analysis performed to test these ideas. I.e. cross correlation of EMG and FoxP2 activity to see if FoxP2 activation precedes movements as suggested.

Response: *See response above.*

Fig 3e: Averaged or summed $\Delta F/F$ should be shown. Although it is stated that similar oscillations weren't present in NREM (Line 245-248) this isn't apparent from the data shown. Additional analysis could help strengthen this point. And again, do these neurons also show responses to CO₂ when mice are awake? Was the activity of these neurons recorded during normally behaving conditions in awake mice? Do single neurons show activity profiles similar to those observed in the fiber photometry experiments (i.g. Supplementary Fig 3A)

Response: *All of the GCaMP recordings were done on freely behaving mice, who were exposed to repeated 30 sec bouts of 10% CO₂ separated by 5 min intervals with room air. Summed $\Delta F/F$ during the first 15 sec of CO₂ exposure is shown for the entire population recorded by fiber photometry in Fig. 2. The activity of 28 individual neurons across a series of CO₂ exposures is shown in Fig. 3g. While it is possible to graph the sum of the activities of the neurons shown in panel 3g, we are not sure of the value of this. Unlike panel 2f, which shows the population response of a large number of PB^{FoxP2} neurons, endomicroscopy experiments*

show the activity of only a handful of individual cells which happen to be near the front of the GRIN lens. The waves of activation seen in the neurons in panel 3g is quite different from the population response in panel 2e, which in turn is different from the summed responses over 59 trials in fig. 2f. This suggests that the activity of PB^{FoxP2} neurons is variegated, but that the overall activity of the population across trials is closely correlated with but slightly leads the increase seen in RR and Vt with CO₂ stimulation. To give a better picture of the variability of individual neuron GCaMP fluorescence, we show neuronal activity profiles of individual neurons as videos during both REM sleep, wake and with exposure to hypercapnia. In addition, activity from multiple neurons is shown during different states (wake, NREM and REM sleep) in supplementary Fig.3c-f. During wake, when animals are exposed to hypercapnia, the neuronal activity profile mainly correlates with their movement (EMG) which also coincides with increase in RR (SFig 5) as seen with cross correlation analysis.

Line 263-265: What about trials when mice were in wake state or REM? It would be interesting to know how FoxP2 neurons may affect breathing in these conditions as well.

Response: We show the effect of photo-stimulation during wake on respiration in SFig 7a-f and discuss the effect of photo-inhibition of PB^{FoxP2} on respiration during wake and REM sleep (Result section, last line of next to last paragraph).

Line 267: FoxP2 activation causes arousal from sleep – it's unclear how different this is from the effect of CGRP neurons?

Response: This issue is now clarified in Paragraph 1 of Discussion. In brief, activation of PB^{FoxP2} neurons does not wake the animal up at all in 30% of trials, and in the remaining trials the animal only awakens near the end of the stimulation, or after it has finished. By comparison stimulation of the PB^{CGRP} neurons awakens animals in all trials in less than 2 sec. Inhibiting or ablating the PB^{FoxP2} neurons has no effect on arousal from NREM sleep caused by CO₂, whereas ablating or inhibiting the PB^{CGRP} neurons almost completely prevents arousal from NREM sleep caused by CO₂.

Line 268-270: A supplemental Fig to show/quantify the included/excluded fiber placement locations would be helpful.

Response: We have added SFig 6a in the revised version detailing this information.

Line 274-283: It would be helpful to quantify instantaneous RR vs time during 5 and 10 sec stimulations to support the point the authors are trying to make (i.e. similar to Fig. 2h-i). 5 breath "bins" does not provide much resolution. Line 289-292: Similar to the above, if there really is a different time course of the respiratory response in 2% CO₂, this should be illustrated in a graph similar to Fig. 2h-i. As currently plotted the response to 20Hz, 10s stim looks identical in normocapnia vs. 2% CO₂ (Compare Fig 4 f (blue) and 4g (purple)).

Response: We have included SFig 6b and c to provide this data.

Line 297: Activation of these projections to the preBötC area would be a nice addition to demonstrate whether or not this is the relevant pathway.

Response: This is the same question as Major Critique 3. We agree that the role of the different medullary projections of the FoxP2 neurons is indeed an interesting problem. However, it would require a massive study, using activation and inhibition of the $PB-KF^{FoxP2}$ terminals in at least four regions of the medulla, in wake and sleep states, with and without CO₂, to work this out. This would take at least two years to accomplish, and is far beyond the scope of the current study.

Line 299: preBötC or PBC is more conventional notation for preBötzinger Complex than PBZ.

Response: We wished to avoid confusion with parabrachial complex (also sometimes abbreviated PBC) in our manuscript. PreBötC is much longer (and requires inserting an umlaut over the “o”), which takes additional steps on an English based word processor. In English, most scientists call this the “preBotz”. Hence, we chose the use of PBZ, which is distinctive and short and mimics the way the term is most commonly pronounced by working scientists in English.

Line 321: How does inhibition of FoxP2 neurons affect breathing in awake and REM?

Response: This is the same as the question asked about Lines 263-5. We show the effect of photo-stimulation during wake on respiration in SFig 7a-f and discuss the effect of photo-inhibition of PB^{FoxP2} on respiration during wake and REM sleep (Result section, last line of next to last paragraph).

Line 326: Reference Fig 6i as well.

Response: We have included that.

Fig 6 c2,c5: Scale bars seem mislabeled?

Response: We have now corrected this.

Line 331-333: The specific role of response to CO2 during sleep is not demonstrated because data testing the role of FoxP2 neurons in the awake state is not provided. Thus, these neurons may have a similar role in sleep and wake states.

Response: This is the fourth time the referee has asked this same question. We show the effect of photo-stimulation during wake on respiration in SFig 7a-f and discuss the effect of photo-inhibition of PB^{FoxP2} on respiration during wake and REM sleep (Result section, last line of next to last paragraph).

Line 333-335: could this reflect the “core/shell” configuration of PB Oprm1 neurons as suggested by PMID: 34921781

Response: Virtually all of the $PBeI^{CGRP}$ and $PBcl^{FoxP2}$ neurons express *Oprm1*. So, they are all in what Liu et al. call the “shell”, i.e., an area that expresses *Oprm1* and projects to the PBZ region. We found that more than 70% of the neurons we retrogradely labeled from the PBZ region in both the *PBcl* and *KF* also express *FoxP2*. But they do not really form a “shell”; rather they wrap around the dorsal and lateral margins of the *PBeI*. It is possible that the *Oprm1* neurons that are ventromedial to the $PBeI^{CGRP}$ cluster and project to the medulla may belong to a different group and have a different function. Our recent spatial transcriptomic analysis of the PB region indicates that there are a very large number of different cell types that express *Oprm1*.

Line 412: Photostimulation of inhibitory preBotC neurons generally slows RR (e.g. PMID: 25643296; PMID: 33670653; PMID: 33863785) unless done specifically during the inspiratory phase (PMID: 29483589) or under specific conditions (ref 38). The other reference provided (39) describes activation of excitatory neurons, which do consistently increase RR.

Response: We agree with reviewer that PBZ is a complex area which needs to be very selectively targeted for understanding their responses.

Other general comments:

It would strengthen the paper to be clear about whether FoxP2 neurons in *PBcl*, *KF*, or both were primarily targeted during optogenetic imaging and manipulation experiments, since these groups of neurons could have distinct functional roles.

Response: We have revised the paper to indicate that the experiments requiring an optical fiber, for technical reasons, were all done by placing the optical fiber above the $PBcl^{FoxP2}$ cluster. The KF^{FoxP2} cluster is more rostral and ventral, and placing an optical fiber specifically above these neurons would destroy most of the $PBcl$ and $PBel$, which would confound the experiments. Our new data with genetic ablation includes both the $PBcl$ and KF groups, and this is compared with opto-inhibition that largely targeted only the $PBel^{FoxP2}$ neurons.

In general, the figures could be improved by making the size and font of text consistent (some axis labels are very small, some are large, some text is bold, some not, etc.). There is also a lot of color used which can get somewhat confusing, and it is not always consistent. E.g. CO_2 trace is green and $\Delta F/F$ is red in Fig 2d but CO_2 is orange and $\Delta F/F$ is black in Fig 2g.

Response: We agree and have now improved the figures to make them clearer.

Parabrachial anatomy could be more clearly/consistently referenced in the text. E.g. PBL vs LPB vs $PBel/PBcl$

Response: We have now made these changes.

REVIEWER COMMENTS

Reviewer #1 (Remarks to the Author):

The authors addressed my concerns. They added proposed mechanisms for FoxP2 neurons to receive chemosensory information and they improved the display of quantitative information in their figures. I still object to using standard error in place of standard deviation, but since the authors include the original data points, I will leave it to the editors to decide whether the changes are acceptable. In most circumstances I would advocate for showing the original data points (unless the data set has 100's or 1000's of points), using SD to quantify variability, and then plotting effect size (difference in means) with 95% confidence intervals. In this context, adding the original data points to the figures is an important change; editors can decide if that is enough. Overall the paper is well done experimentally and advances understanding of the Parabrachial / Kolliker-Fuse regions that are important for respiratory neural control.

Reviewer #2 (Remarks to the Author):

The authors' revisions addressed most of my initial critiques. They improved their manuscript by adding a DTA ablation experiment and helpful anatomical details, but I must admit some difficulty in understanding quite how the different pieces of the story fit together. After CO₂ exposure, there was a rise in PB-Foxp2 calcium activity at the time of increased RR, and stimulating PB-Foxp2 neurons increased RR (with minimal change in V_t), yet inhibiting or ablating the same neurons did not alter RR but did reduce V_t and MV (after CO₂ exposure). I get that these neurons may be sufficient to increase RR but not necessary for that same effect after CO₂ exposure, but it is difficult to see how a group of neurons with little to no effect on V_t (when they are photostimulated) would nonetheless be necessary for the majority of an increase in V_t after CO₂ exposure. None of that is to say that I don't think the results should be shared/published, and I do think the results will be of interest to investigators in several related fields.

More importantly, from the abstract through the Discussion, the authors have made overly specific claims about the location(s) of neurons responsible for their results. Ideally, they would have a more precise marker than Foxp2 for their neurons of interest, which appear to be a small minority of neurons in the lateral PB that project axons to the medulla, rather than the forebrain. Separate publications have blamed different aspects of the lateral PB Foxp2 population for itch (Ren et al. Neuron 2023) and for immune-mediated HPA axis activation (Jagot et al. Neuron 2023), and the primary output targets of Foxp2-expressing neurons in the PB region are in the forebrain and midbrain. The present results are interesting, but the anatomical terms for locations targeted and neurons transduced should be provided with more nuance because this tiny brain contains a seemingly continuous population of Foxp2-expressing neurons, spanning several subnuclei (originally named and defined by the senior author of this paper) -- including central lateral, superior lateral, and dorsal lateral PB. Even the medial and external lateral subnuclei have a sprinkling of Foxp2 neurons. Immediately outside the PB, contiguous Foxp2 neurons fill neighboring nuclei, including the "caudal KF" (GABAergic neurons) as well as the KF and most of the NLL (glutamatergic neurons). The authors' AAV injections certainly transduced neurons across multiple PB subnuclei and surrounding nuclei, on one or both sides, in most cases. It is impossible that a stereotaxic injection at the volumes listed, in this tiny region of the mouse brainstem, could have remained confined to a single PB subnucleus like PBcl. Thus, the term PB"cl" is overly restrictive (leaving out, for example, the lateral crescent region), and the authors should refer to their region of neuronal transduction in photometry/imaging, stimulation, ablation, and inhibition experiments as "PBFoxp2" or "PB/KFFoxp2" (or perhaps "LPBN/KFFoxp2"), not PBcl/KFFoxp2.

Other suggestions (at the authors' discretion) which may improve the clarity of their manuscript for readers:

- Abstract: I recommend deleting the line "but has little if any effect on the respiratory response," given that there is a robust respiratory response (associated with arousal). The authors may or may not be correct in guessing that the respiratory response is mechanistically downstream from the arousal response triggered by CGRP neurons in PBel, but it is not clear from their previous publication how this was (or could be) falsified in a way suggested by "little if any".
- Line 136: "neuron" should be plural
- Line 153-157: It is difficult to understand how the authors find Calca mRNA co-localizing with a GFP Cre-reporter for Foxp2, and nearly 100% co-localization of FoxP2-ir with this GFP Cre-reporter, yet they cannot find co-localization of CGRP and FoxP2 directly. Instead of addressing this interesting dichotomy (and instead of showing direct co-localization between Foxp2 mRNA and Calca mRNA, or between Foxp2 mRNA and CGRP-ir, or between FoxP2-ir and Calca mRNA), they discussed spatial transcriptomic data from a separate (unpublished) paper. As in my initial critique, I recommend that the authors put their conclusion to the test by directly comparing the mRNA and protein products of each genetic marker -- and show the results.
- Line 286: waves of periodic activity, with peaks separated by 10+ seconds, make me think of gastrointestinal peristalsis driven by the MMC. In a resting mouse gut, the frequency of spontaneous GI contractions is in the neighborhood of a few per 30 seconds, and increases upon luminal distention. The lateral PB receives the bulk of its input from the caudal-medial NTS region that receives vagal input from the stomach and other parts of the upper GI tract. Pdyn neurons (most of which co-express Foxp2) in this part of the PB have been implicated in sensing gastric fullness (Kim et al Nature 2020), and this may be worth mentioning as an alternative hypothesis for this unusual activity pattern, which does not look like it has much to do with breathing or arousal.
- Lines 347-8: The authors claim to "map the projections of the PBelFoxP2 neurons expressing ChR2-mCherry" but then mention nothing of their (predominantly) rostral projections. Something should be added here to clarify this, at least in passing. Otherwise, non-expert readers would be forgiven for concluding, after reading this paragraph, that the authors' cases had axonal labeling exclusively in the lower brainstem.
- Line 368: delete "such as..." -- no evidence is provided specifically that PBelCGRP neurons survived. That is almost certainly true, but this is stated as if the authors actually labeled and quantified effects on that population, which was not shown or claimed.
- Line 406: the modifiers "immediate" and "with a latency of less than 2s" seem to be fighting each other. Deleting the former would improve clarity.
- Line 429: "insure" should be "ensure"

Reviewer #3 (Remarks to the Author):

The authors have addressed my comments and have made important additions/revisions that have improved the paper. I have no further concerns.

Response to reviewers:

We thank the reviewers for their careful reading of our manuscript. Responses below are in RED.

Reviewer #1 (Remarks to the Author):

The authors addressed my concerns. They added proposed mechanisms for FoxP2 neurons to receive chemosensory information and they improved the display of quantitative information in their figures. I still object to using standard error in place of standard deviation, but since the authors include the original data points, I will leave it to the editors to decide whether the changes are acceptable. In most circumstances I would advocate for showing the original data points (unless the data set has 100's or 1000's of points), using SD to quantify variability, and then plotting effect size (difference in means) with 95% confidence intervals. In this context, adding the original data points to the figures is an important change; editors can decide if that is enough. Overall the paper is well done experimentally and advances understanding of the Parabrachial / Kolliker-Fuse regions that are important for respiratory neural control.

Response: The reviewers comments are well taken. There are a number of ways to display statistical differences, and we are now providing the dot-plots plus the SEM, which is the SD/(square root of n). We find this useful because the 95% confidence interval for each observed mean is 2xSEM, and we prefer p values for the differences in mean, because these indicate the likelihood of obtaining this difference due to random variation rather than systematic differences between the two means. Most recent experimental studies published in Nature Communications seem to use this approach, although we do understand that many human studies with large n's use the method that the reviewer prefers. We are willing to display our statistics in whatever way the editors require.

Reviewer #2 (Remarks to the Author):

The authors' revisions addressed most of my initial critiques. They improved their manuscript by adding a DTA ablation experiment and helpful anatomical details, but I must admit some difficulty in understanding quite how the different pieces of the story fit together. After CO₂ exposure, there was a rise in PB-Foxp2 calcium activity at the time of increased RR, and stimulating PB-Foxp2 neurons increased RR (with minimal change in V_t), yet inhibiting or ablating the same neurons did not alter RR but did reduce V_t and MV (after CO₂ exposure). I get that these neurons may be sufficient to increase RR but not necessary for that same effect after CO₂ exposure, but it is difficult to see how a group of neurons with little to no effect on V_t (when they are photostimulated) would nonetheless be necessary for the majority of an increase in V_t after CO₂ exposure. None of that is to say that I don't think the results should be shared/published, and I do think the results will be of interest to investigators in several related fields.

Response: We appreciate this issue, and tried to address it in the previous version of our manuscript, in lines 323-325 in the Results and 479-481 in the Discussion. Our explanation for the lack of an increase in V_t with photostimulation of the PB-FoxP2 neurons was 1.) that the increase in V_t during CO₂ exposure takes >10 sec to develop (Fig. 2), and 2.) that the duration of the photostimulation was limited to 5 or 10 sec (Fig. 4f), so may not have been sufficient for a

change in V_t to develop. We have added a sentence to the text after line 481 (in the previous version), which is highlighted in the revised manuscript, to better explain this idea.

We also draw to the reviewer's attention that 5 sec of photostimulation failed to cause any change in mean V_t at all, but that by 10 sec the mean V_t during 10 sec of stimulation at 10 or 20 Hz was roughly 15% greater (Fig. 4). However, V_t had a larger variance than RR at all time points during both CO₂ exposure and photostimulation, so that the 15% increase in mean did not reach statistical significance. These data are consistent, though, with our hypothesis that the change in V_t may take longer to develop than the change in RR.

On the other hand, in the deletion experiment (Fig. 6), when the remaining respiratory cell groups can increase RR in response to elevated CO₂, the PB/KF-FoxP2 neurons turn out to be required for the elevation of V_t during CO₂ exposure. In these experiments, with exposure to 10% CO₂, the animals typically wake up between 17 and 20 sec into the exposure, and the pre-arousal V_t measurement is made on the last 5 breaths (about 1.5 sec) before the animal wakes up, so at least 15 sec into the CO₂ exposure. By this time, the V_t has already increased substantially. It goes up further after arousal, presumably reflecting forebrain input to the PB/KF-FoxP2 neurons. We hope that this is clearer now.

More importantly, from the abstract through the Discussion, the authors have made overly specific claims about the location(s) of neurons responsible for their results. Ideally, they would have a more precise marker than *Foxp2* for their neurons of interest, which appear to be a small minority of neurons in the lateral PB that project axons to the medulla, rather than the forebrain. Separate publications have blamed different aspects of the lateral PB *Foxp2* population for itch (Ren et al. *Neuron* 2023) and for immune-mediated HPA axis activation (Jagot et al. *Neuron* 2023), and the primary output targets of *Foxp2*-expressing neurons in the PB region are in the forebrain and midbrain. The present results are interesting, but the anatomical terms for locations targeted and neurons transduced should be provided with more nuance because this tiny brain contains a seemingly continuous population of *Foxp2*-expressing neurons, spanning several subnuclei (originally named and defined by the senior author of this paper) -- including central lateral, superior lateral, and dorsal lateral PB. Even the medial and external lateral subnuclei have a sprinkling of *Foxp2* neurons. Immediately outside the PB, contiguous *Foxp2* neurons fill neighboring nuclei, including the "caudal KF" (GABAergic neurons) as well as the KF and most of the NLL (glutamatergic neurons). The authors' AAV injections certainly transduced neurons across multiple PB subnuclei and surrounding nuclei, on one or both sides, in most cases. It is impossible that a stereotaxic injection at the volumes listed, in this tiny region of the mouse brainstem, could have remained confined to a single PB subnucleus like PBcl. Thus, the term PB"cl" is overly restrictive (leaving out, for example, the lateral crescent region), and the authors should refer to their region of neuronal transduction in photometry/imaging, stimulation, ablation, and inhibition experiments as "PB*Foxp2*" or "PB/KF*Foxp2*" (or perhaps "LPBN/KF*Foxp2*"), not PBcl/KF*Foxp2*.

Response: We appreciate this problem, and had originally called these PB-FoxP2 and KF-FoxP2 neurons in the first version of this manuscript. However, while there are many types of FoxP2 neurons in the lateral PB (see Huang et al. 2021), it is clear from Fig. 1 that the CO₂-responsive neurons in the lateral PB are limited to the most ventral cluster of FoxP2 neurons in the PBcl subnucleus (with a few in the lateral crescent). So, we changed the terminology in the first revision to PBcl^{FoxP2} neurons to reflect that the only cells that show cFos responses to CO₂

and which project to the ventrolateral medulla from lateral PB are in the PBcl group. However, the referee is technically correct that, although the PBcl-FoxP2 group is separated physiologically, connectionally, and spatially from the FoxP2 neurons located more dorsally in the PB (which were the ones studied by Ren et al and Jagot et al), the injections of viral vectors frequently involved the more dorsal parts of the lateral PB as well. We have therefore adopted the terminology suggested by the reviewer (PB^{FoxP2} neurons) for the optogenetic and lesion experiments, where we may have included more dorsally located FoxP2 neurons. This is now explained in the section of “Photo-activation of PB^{FoxP2} neurons...”, with the added sentences highlighted. However, in the fiber photometry experiments we only obtained responses to CO₂ when the optical fiber or endoscope was just above or in the PBcl, which is the only part of the lateral PB that showed cFos responses to CO₂. Hence in those experiments we have a high level of confidence that the results are obtained from PBcl-FoxP2 neurons.

Other suggestions (at the authors' discretion) which may improve the clarity of their manuscript for readers:

- Abstract: I recommend deleting the line “but has little if any effect on the respiratory response,” given that there is a robust respiratory response (associated with arousal). The authors may or may not be correct in guessing that the respiratory response is mechanistically downstream from the arousal response triggered by CGRP neurons in PBel, but it is not clear from their previous publication how this was (or could be) falsified in a way suggested by “little if any”.

Response. The sentence in question reads: “We previously found that inhibition of the glutamatergic neurons expressing calcitonin-gene related protein (CGRP) in the external lateral PB...prevents cortical arousal in response to hypercapnia, but has little if any effect on the respiratory responses.” We stand by that sentence. The referee is misinterpreting it as saying that the PBel-CGRP neurons have little or no effect on respiratory responses. That is not what the sentence says. We agree with the referee that the arousal responses caused by activating the PBel-CGRP neurons result in respiratory responses, but that is not what the sentence was about.

- Line 136: “neuron” should be plural

Response. Agreed. The change has been made.

- Line 153-157: It is difficult to understand how the authors find Calca mRNA co-localizing with a GFP Cre-reporter for Foxp2, and nearly 100% co-localization of FoxP2-ir with this GFP Cre-reporter, yet they cannot find co-localization of CGRP and FoxP2 directly. Instead of addressing this interesting dichotomy (and instead of showing direct co-localization between Foxp2 mRNA and Calca mRNA, or between Foxp2 mRNA and CGRP-ir, or between FoxP2-ir and Calca mRNA), they discussed spatial transcriptomic data from a separate (unpublished) paper. As in my initial critique, I recommend that the authors put their conclusion to the test by directly comparing the mRNA and protein products of each genetic marker -- and show the results.

Response. The FoxP2-ir is very strong, and it colocalizes 100% with the GFP reporter for FoxP2. The latter colocalizes with Calca mRNA in the small, Calca+ KF neurons. Hence these

neurons make both FoxP2 and Calca mRNA, although they do not stain for Calca immunoreactivity.

The reason for this disparity is likely due to the difference in the level of *Calca* mRNA (and hence, CGRP) expressed in the larger PBel-*Calca* neurons compared to the smaller, KF-*Calca* cells. The paper showing the differential expression of *Calca* between these two distinct cell types was cited as a preprint in the previous version, but has since been published in final form, which is now cited (Nardone S, De Luca R, Zito A, Klymko N, Nicoloutsopoulos D, Amsalem O, Brannigan C, Resch JM, Jacobs CL, Pant D, Veregge M, Srinivasan H, Grippo RM, Yang Z, Zeidel ML, Andermann ML, Harris KD, Tsai LT, Arrigoni E, Versteegen AMJ, Saper CB, Lowell BB. A spatially-resolved transcriptional atlas of the murine dorsal pons at single-cell resolution. *Nat Commun.* 2024 Mar 4;15(1):1966. doi: 10.1038/s41467-024-45907-7.) The reviewer is referred to a detailed discussion of this issue is on p. 5, left hand column, of that publication.

As an aside, many peptides are difficult to stain for immunohistochemically, as the peptides are rapidly exported to the axon terminals. Only cells with high levels of expression of a peptide are usually seen by immunohistochemistry, unless one dams up the peptide product in the cell bodies by either cutting the axon or causing disassembly of axonal microtubules with colchicine. For example, in 1985 we showed that CCK neurons in the superior lateral PB project to the VMH (Fulwiler and Saper, *Neurosci Lett* 53:289), by doing a combination of retrograde labeling and immunohistochemistry in colchicine treated animals. In non-colchicine animals, there was little immunostaining for CCK cell bodies in the PB. Zaborszky et al (*Brain Res* 303, 225, 1984) showed the same pathway by placing a lesion in the medial forebrain bundle that caused loss of CCK staining in the VMH, but by preventing transport of CCK down the axon, caused cell body staining in the PB that was not there in intact animals. Garfield et al. (*Cell Metab* 20:1030, 2014) later showed the same PB-CCK neurons, which are not seen by IHC in untreated animals, by using as CCK-Cre mouse crossed with a GFP reporter line. Another example is the presence of AVP and oxytocin in hypothalamic neurons projecting to the spinal cord. Without colchicine, Sawchenko and Swanson (1983) estimated that fewer than 5% of the spinally projecting neurons in the paraventricular nucleus expressed these peptides; with colchicine, Cechetto and Saper (1987) found about 25-35% of these neurons contained each peptide; with in situ hybridization, Hallbeck et al (1981) found that 40% expressed AVP mRNA.

The bottom line is that the large PBel CGRP neurons are exceptional for their immunohistochemical stainability without colchicine, but the small KF-*Calca* neurons are not stained under those conditions (at least in our laboratory or that of Huang et al). We do not think it is worthwhile to do more experiments to show what is widely known (and has repeatedly been shown, e.g., in the experiments cited above): that use of in situ hybridization (or Cre-reporter mice) is more sensitive than immunohistochemistry for identifying peptide expression by brain neurons.

- Line 286: waves of periodic activity, with peaks separated by 10+ seconds, make me think of gastrointestinal peristalsis driven by the MMC. In a resting mouse gut, the frequency of spontaneous GI contractions is in the neighborhood of a few per 30 seconds, and increases upon luminal distention. The lateral PB receives the bulk of its input from the caudal-medial NTS region that receives vagal input from the stomach and other parts of the upper GI tract. Pdyn neurons (most of which co-express Foxp2) in this part of the PB have been implicated in sensing gastric fullness (Kim et al *Nature* 2020), and this may be worth mentioning as an

alternative hypothesis for this unusual activity pattern, which does not look like it has much to do with breathing or arousal.

Response. The reviewer's thoughts on the possible origin of the responses of PBcl-FoxP2 neurons that have long periodicity are interesting. Kim et al were studying PB-*Pdyn* neurons that responded to various oro-pharyngeal and upper gi stimuli. Some of the PB-*Pdyn* neurons are in the PBcl and express FoxP2, although as Kim et al show (and Geerling showed in 2016, *AJPReg*), most are much more dorsally located and belong to the dorsal lateral FoxP2-*Pdyn* population. In addition to Kim et al. observing that some of the dorsal *Pdyn* neurons respond to oropharyngeal and upper gi stimuli, Geerling showed that some dorsal lateral *Pdyn* neurons respond to warming of the skin. It is likely that these constitute two different cell types, but studies to show that have not been done, and convergence of multiple types of stimuli on individual neurons, even of different cell types, is likely. For example, instilling water in the oropharynx in Kim's study may have activated the temperature responsive neurons that Geerling identified, depending upon the temperature of the water.

In our experiments, the GCaMP responses of the PBcl-FoxP2 neurons shown in Fig 3g are clearly CO2 responsive, because they had a sharp increase in their GCaMP fluorescence after onset of a CO2 stimulus. As we said in the previous version of our manuscript (lines 286-288), it is possible that some of the PBcl-FoxP2 neurons may receive other afferents which may give this periodicity to their GCaMP signal when they are activated. It is possible, as the reviewer indicates, that this could be due to convergence of inputs from oropharyngeal or upper gi vagal sources on the same neurons that respond to CO2. We have added this possibility to the Discussion at the end of the section on endomicroscopy of PBcl neurons.

- Lines 347-8: The authors claim to “map the projections of the PBclFoxP2 neurons expressing ChR2-mCherry” but then mention nothing of their (predominantly) rostral projections. Something should be added here to clarify this, at least in passing. Otherwise, non-expert readers would be forgiven for concluding, after reading this paragraph, that the authors' cases had axonal labeling exclusively in the lower brainstem.

Response. The reviewer makes a good point. We have now revised the text in the last paragraph of the section on “Photoactivation of PB^{FoxP2} neurons...” to clarify that this section is only about the PBcl^{FoxP2} projection to the medulla. The ChR2 injections involved the more dorsal lateral PB^{FoxP2} neurons as well as the PBcl^{FoxP2} cells. That was why, after finding medullary projections in those brains, we then did retrograde labeling studies to show that of the lateral PB^{FoxP2} neurons, only those in the PBcl project to the medulla.

- Line 368: delete “such as...” -- no evidence is provided specifically that PBclCGRP neurons survived. That is almost certainly true, but this is stated as if the authors actually labeled and quantified effects on that population, which was not shown or claimed.

Response. The referee is technically correct. The statement has been modified to say that “Immunostaining for mCherry showed that other neuronal populations, such as the PBcl neurons,...are intact.”

- Line 406: the modifiers “immediate” and “with a latency of less than 2s” seem to be fighting each other. Deleting the former would improve clarity.

Response. Agreed. We have made this change.

- Line 429: "insure" should be "ensure"

Response. Agreed. We have made this change.

Reviewer #3 (Remarks to the Author):

The authors have addressed my comments and have made important additions/revisions that have improved the paper. I have no further concerns.

REVIEWERS' COMMENTS

Reviewer #2 (Remarks to the Author):

The authors addressed my critiques and further improved their manuscript.

Response to reviewer's comments:

Reviewer #1

Summary of key results:

The authors identify a new subclass of neurons in the parabrachial nuclear complex, including the adjacent Kolliker-Fuse, which is involved in increasing respiratory drive during chemosensory challenge, i.e., elevated carbon dioxide (hypercapnia). This new subclass is identified by transcription factor FoxP2, rather than expression of peptide CGRP, which the authors previously identified. The CGRP population is involved in arousal, but not respiratory drive. The FoxP2 subclass participates in both respiratory drive and arousal. These FoxP2 neurons are associated with REM sleep and wakefulness.

The experiments are convincing and complement the authors former discovery of the CGRP population and this study does advance understanding.

There are a few aspects that dampen enthusiasm:

Major comments:

1) It is unclear how the FoxP2 neurons get chemosensory information. Their activity in response to hypercapnia comes at a delay, and the latency to activation is heterogeneous (Figure 3d) so it is unlikely intrinsic chemosensitivity is at work. It seems likely that these neurons are connected to central chemosensory neurons in the retrotrapezoid nucleus / parafacial ventral (RTN/pFV). The authors should discuss how FoxP2 parabrachial / Kolliker-Fuse neurons receive chemosensory information.

Response: *This is shown in Fig 8 and discussed in the second to last paragraph of the discussion.*

2) The figures are difficult to understand in many respects. Figure 1 uses bar plots with standard error, rather than showing all the data points and plotting standard deviation or confidence intervals, which have much greater fealty to the raw data and give the reader more evidence of effect size. Figure 2 uses inconsistent tick and typeface size, which make it extremely hard to gauge effect sizes. Figure 3 has similar problems regarding ticks and inconsistent sizes of typefaces. Panels b and c in Figure 3 are actual screen shots that cannot be read with any accuracy, straining the reader. Figure 3 panels b and c show the same 3 neurons and basically the same time series. Certainly, those data can be taken from acquisition and analysis software and made into a presentable figure in a single panel. Figure 4 has the same problems with inconsistent typeface and ticks. Figure 5 is a patchwork of panels that contain valuable information but are packed so tightly together with all labels within the panels, again, straining the reader. Figure 6i is the type of "show the data" plot I would recommend for Figure 1e and f, but here a standard error or confidence interval would be preferable. Also, here the typeface in Figure 6b would be nearly unreadable in typical PDF size for reading (the reader will have to expand the illustration to get the information.)

Response: *We have gone through the figures to make them more consistent in the use of typefaces and easier for the reader to digest. In Fig 1 we added data points in all bar plots. The two parts of Fig 3 have now been combined into a single time series.*

3) The written manuscript overall suffers haphazardness. Time is sometimes "s" and other times "sec". Sometimes the units are separated from numerals, e.g., "20 Hz" and sometimes not "20Hz". Figure callouts are inconsistent too: "Fig. 1" or "Fig.1" or "Fig 1". These inconsistencies are rampant. While this critique does not have any bearing on scientific content, it does not, in my view, show attention to detail.

Response: *We have gone over the manuscript carefully to be as consistent as possible in terminology and typography.*

Minor comments:

Lines 37, 107: CGRP should be defined.

Response: *We have made this change.*

Line 91: how can ventilation increase during an apnea, when apnea -- by definition -- is a pause in breathing (no ventilation).

Response: *We have now rephrased this.*

Figure 1e & f: the bar charts should be replotted to show all the data points as well as either 95% confidence intervals or standard deviation rather than standard error of the mean, to show the data (see: Drummond GB, Vowler SL. Show the data, don't conceal them. *J Physiol (Lond)* 589:1861–1863, 2011)

Response: *We added the requested datapoints. We chose to leave the error bars as SEM, as the datapoints now indicate the variability in the data.*

Line 141: the number of the figure is missing.

Response: *We have added this now.*

Figure 2d, f-i: the y-axes are labeled with too many and often small ticks and too small type face. It is difficult to read these axes and make sense of the data. The x-axes are inconsistently labeled with different size type faces and time is "s" or "sec". The graphics should be more elegant and easier to read (see: Tufte ER. *The visual display of quantitative information*, 2nd ed. Cheshire, Conn: Graphics Press, 2001.)

Response: *We are now keeping them consistent and similar.*

Lines 180-204: The figure callouts are sloppily labeled, sometimes missing a period (180, 181, 190) and other times lacking a gap between the period and the numeral (192, 198, 204). While these simple typos have no bearing on scientific content, the sloppiness is disturbing for its excessive frequency.

Response: *We apologize for this sloppiness and have made corrections.*

Figure 4e: Should be "minute ventilation" not "minute volume".

Response: *We have made this correction.*

Line 209: missing "P="

Response: *We have corrected this.*

Line 218: "movement" is misspelled.

Response: *This is now corrected.*

Line 242: I think the callout should be to Fig. 3e (not 3f).

Response: *corrected.*

Line 302: "CTb" should be defined in the main text when it appears. It is defined in the figure legend, but one expects a definition in the main narrative too.

Response: *We have defined CTb.*

Reviewer #2:

Summary of key results:

This study evaluates whether Foxp2-expressing neurons in the lateral parabrachial nucleus (LPBN) or nearby Kölliker-Fuse nucleus (KF) play a role in the hypercapnic ventilatory response (HCVR) in mice. HCVR is the increased breathing rate and volume triggered reflexively in response to increased CO₂ (and the resulting drop in blood pH). HCVR is important for survival, particularly in patients with obstructive sleep apnea, in whom this reflex (and a parallel surge in arousal) kick-starts breathing after prolonged apneas. Understanding the neurologic mechanisms of HCVR is an important goal, so this topic is highly significant.

The premise of this new study stems from two previous articles by the same investigators, published in 2013 (J Neurosci) and 2017 (Neuron). Understanding the premise and conclusions of the new study requires understanding those two papers. In the 2013 study, the authors eliminated a glutamate transporter from a large extent of the LPBN, or selectively ablated glutamatergic neurons in this region, and found a marked reduction in the HCVR and arousal responses to elevated CO₂. Next, in the 2017 study, they focused on a separate group of neurons in the LPBN and proposed that CGRP neurons in the LPBN are important for hypercapnic arousal but not the HCVR. Finally, in the present study, they are advancing an elegant double-dissociation hypothesis by proposing that a separate, large population of FoxP2 neurons in this region of the brainstem is necessary for the breathing response but not for the corresponding arousal. The idea is elegant, but the experimental evidence did not support this conclusion.

Major critiques:

1) The main problem is that the results do not directly or fully support the claim. The authors conclude that LPBN/KF FoxP2 neurons are necessary for the HCVR (lines 332-333: “required for normal respiratory response to CO₂ during sleep”; lines 340-341: “we interpret the PB-FoxP2 neurons as primarily mediating the ventilatory response to CO₂.”), but their ArchT experiment (attempting to inhibit these neurons) had no effect on respiratory rate, and only a small effect on volume (conversely, photostimulating FoxP2 neurons produced an increase in breathing rate, but not volume – and no explanation is advanced for the discrepancy). Importantly, no attempt was made to simply ablate Foxp2-expressing neurons in LPBN; this is the most straightforward way to test the authors’ hypothesis. Robust methods for ablating specific cell types have been in use for over a decade (Cre-dependent expression of diphtheria toxin A; Cre-dependent expression of modified caspase/phosphatase), and the authors were among the first to use one of these techniques (ablating glutamatergic neurons in the LPBN in 2013). Especially given the unconvincing results with ArchT inhibition, the authors must show what happens when they simply remove Foxp2-expressing neurons from the LPBN/KF region. These concerns are particularly important with HCVR, which is highly variable and state-dependent (high propensity for false-positive and false-negative conclusions).

Response: *At the advice of the reviewer, we have now genetically ablated both PBcl and KF FoxP2 populations and reported the results as Fig 6 and in the result section. The results are consistent with, but more profound than the results of opto-inhibition. They show that the ablation of the PBcl/KF^{FoxP2} neurons dramatically reduce the ventilatory response to CO₂, without affecting the latency to arousal.*

2. - Also not supported in this manuscript, or in any previous studies, to my knowledge (apologies if I missed it, but it is not cited in this paper), was a claim made in the first sentence of the abstract: that activating CGRP neurons in LPBN “has little effect on respiration.” The authors showed in a previous article that activating CGRP neurons increases arousal, but they did not show any respiratory measurements in those experiments (not that I could find in the 2017 Neuron paper). No such data are provided for CGRP neurons in the present study. The

proposed double-dissociation (CGRP neurons mediate arousal & FoxP2 neurons mediate hyperventilation) depends heavily upon the authors' claim that CGRP neurons have "little effect on respiration." If this is not true (if stimulating CGRP neurons does in fact increase MV i.e. the respiratory rate and/or volume), such claims should be removed or heavily re-phrased at least.

Response: *The reviewer is correct that we mis-stated this point. In our 2017 Neurons paper (PMID- 29103805), we reported that upon optogenetic stimulation of the PBe^{CGRP} neurons, animals had almost immediate arousal (SFig 1 of the 2017 publication) with average latencies less than 2s in almost all trials. Because respiration increases almost immediately with arousal for any cause, this short interval before EEG arousal makes it difficult to reliably assess the independent effects of stimulation of PBe^{CGRP} neurons on respiration. However, we did show that optogenetic inhibition of these neurons had no effects on the increase in ventilation to hypercapnia (SFig 4, of Kaur et al., 2017). Thus, the corrected sentence, in both the Abstract and Introduction, is that "inhibition of glutamatergic neurons expressing calcitonin gene-related peptide (CGRP) in the external lateral PB (PBe^{CGRP} neurons) prevents cortical arousal in response to hypercapnia, but has little if any effect on the respiratory response." We maintain that the lack of effect of photoinhibition of the PBe^{CGRP} neurons on respiratory response to CO₂ and the lack of effect of ablation of the PB+KF^{FoxP2} neurons on time to arousal to CO₂ establishes the double-dissociation.*

3. Further contradicting the claimed double-dissociation is the author's finding that stimulating FoxP2 neurons in the LPBN triggered arousal in 70% of their ChR2/photostimulation trials. The authors downplay this observation because the arousal happened with a bit of a delay, relative to the HCVR, but it undercuts the idea that FoxP2 neurons specifically trigger HCVR and not arousal. The lack of direct comparisons between arousal and HCVR latency after photostimulating CGRP vs. FoxP2 neurons (and the prior and current lack of information on HCVR after activating CGRP neurons) weakens the conclusions of this study.

Response: *As we point out in the text, the animals do not arouse at all in 30% of the trials even with vigorous optogenetic stimulation of the PB^{FoxP2} neurons that increases RR by 50%, and that when the animals do arouse, it is often after the optostimulation has been terminated (the average latency to arousal in Fig 4d is consistently greater than the duration of stimulation). This compares to arousal within a mean of 2 sec on every trial when the PBe^{CGRP} neurons were stimulated (see Kaur et al., 2017, Fig S1F). Conversely, as indicated in the Response above, the ablation/inhibition of the two neuronal populations yields a crystal-clear double-dissociation.*

4. The bottom part of Supplemental Figure 2, and the entire Supplemental Figure 3 was missing. I tried opening or importing the Supplement PDF in several different applications and on multiple computers, with the same problem each time. I cannot complete this review without seeing the full paper.

Response: *We apologize for this. Please let us know if this happens again, we can provide you with the full resolution images.*

5. A major flaw in this study has to do with the non-specificity of recording and stimulating neurons that express the gene Foxp2. Anatomical work cited in this new study identified heterogeneous populations of FoxP2 neurons in and around the LPBN (including GABAergic neurons in the caudal KF, glutamatergic neurons in rostral KF, and several more glutamatergic populations in LPBN). Another paper, which just appeared in Neuron, identified LPBN Foxp2-expressing neurons that transmit chemical and mechanical itch (Ren et al. 2023). The authors do not go into detail about this complexity and heterogeneity, instead (Fig. 7) treating these aggregate FoxP2 populations as if they had a uniform function (HCVR). If the loss-of-function result had been compelling (see above), this would be less of a problem, but the results

presented leave it unclear what concrete advance was made, relative to previous results.

Response: *The reviewer is correct that there are several different, and anatomically distinct clusters of FoxP2 neurons in the parabrachial complex. However, as we show in Fig. 1, the PB neurons that express cFos after hypercapnia are in the KF and PBcl, and bridging between them along the lateral margin of the PBel^{CGRP} population, and the overwhelming majority of them express FoxP2. On the other hand, the FoxP2 neurons identified by Ren et al. (Neuron 111:1812, 2023) are clearly identified as being in the dorsal part of the lateral PB. For this reason, in the revised manuscript we have more clearly identified the neurons we are dealing with as being the PBcl^{FoxP2} and KF^{FoxP2} clusters.*

Minor critiques

1. Insufficient attention is given to counterfactuals (negative-control considerations are lacking, or are glossed over). For example, the number (and to a lesser extent, the full distribution) of neurons with a developmental history of Foxp2 expression is more than adult expression of this gene (and of adult FoxP2 immunofluorescence labeling). The authors use a Cre-reporter strategy (fate-mapping developmental expression plus subsequent/adult expression) as a surrogate for actual expression, and justify this with a blanket, qualitative statement about co-expression of the reporter with nuclear immunostaining, but they should at least count the numbers of FoxP2-immunoreactive and GFP-expressing neurons and give the actual numbers and proportions. This is worth pointing out because the authors made clear, black-and-white claims of “validation” and “eutopic” adult expression (lines 466, 1103).

Response: *We have now quantified the eutopic expression and have included that in second paragraph of Results section. We also show in our Figs. 2, 4b, 5a, 7c, and S2 that the Cre expression in our FoxP2-Cre mice (as demonstrated by Cre-dependent expression of GCaMP, ChR2-mCherry, ArchT-GFP, and L10-GFP, respectively) is virtually 100% eutopic with the immunohistochemical staining for FoxP2 protein.*

2. If the authors wish to claim that the two genes or their protein/peptide products are expressed together in adult neurons, they should co-label mRNA (for Foxp2 and Calca) or immunostain (FoxP2 and CGRP), or some combination of the two. Labeling mRNA in cells expressing a (developmental fate-mapping) Cre-reporter leaves it unclear whether and to what extent there is any adult co-expression, which is what the authors are claiming (lines 148-152).

Response: *This is provided in Sfig. 1, which combines in situ hybridization for Calca mRNA with FoxP2-L10-GFP expression, which we show in Sfig2. is 100% eutopic with immunohistochemistry for the gene product in adult animals.*

3. The relative functions of LPBN versus KF neurons in this study are not addressed. The development, connections, and functions of KF FoxP2 neurons appear to be quite different from FoxP2 neurons in the LPBN. The senior author’s previous work indicated that neurons in the KF location exert a timing-dependent disruption in the respiratory rhythm by prolonging the post-inspiratory phase (apneusis), and hyperventilatory responses in rats were triggered by neurons located ventrally. Now, however, it seems they are pinning hyperventilatory responses on neurons located dorsally (in the LPBN), without functionally re-mapping this region in mice. This seems like a step backwards and is likely to leave readers confused by the un-addressed discrepancies.

Response: *The previous studies on using microinjections of glutamate to stimulate PB neurons were done by Chamberlin and Saper in rats, in which the anatomy of the neurons that project to respiratory sites in the medulla is quite different. Such neurons are located primarily in the KF and the lateral crescent (along the lateral margin of PBel) in rats, and those were the locations*

in which injections of glutamate were most effective at eliciting increased respiration. The reviewer is incorrect about the location of the apneustic neurons: they were consistently ventral to the bulk of the sites that increased respiration, and mixed with hyperpneic sites in the KF. In mice, by comparison, there is a large population of PBcl^{FoxP2} neurons that project to the respiratory areas of the medulla. We have not encountered an apneustic response from photostimulation in mice, but we have only done optogenetic stimulation of FoxP2 and CGRP neurons, so it is possible that another cell type may cause apneusis when stimulated, or this could be a species difference. We hope that our spatial transcriptomic map of the PB region will allow us (or another group) to identify a different neuron type in the PB that may produce that response.

With respect to potential differences in function between different subgroups of FoxP2 neurons in the PBcl and KF, our data in this paper show that opto-inhibition of the PBcl^{FoxP2} neurons has a very similar effect to ablation of both populations in terms of preventing a CO₂-induced increase in minute ventilation, although the ablation effect is stronger. We do not have data to speculate beyond that.

4. I do not have expertise or experience with the long-latency hypercapnic responses of neurons in the LPBN/KF region. Therefore, I will not critique the calcium-imaging portion of the paper. However, if a 10-second delay in neuronal response is typical for this type of recording, that should be explained to the reader (presumably involving delayed changes in blood pH buffering? I would expect the polysynaptic neuronal responses to be fairly rapid, sub-second, once the RTN or carotid-body chemoreceptors are activated by even mild acidity).

Response: *We have now discussed this in the results section. We performed in vivo recordings in intact mice, which were placed in a plethysmograph, where exchange of gases is gradual (see graphs in Figs. 2d and g). It also takes time for changes in alveolar gas mixtures to reach the arterial circulation (a few seconds). For this reason, we have aligned the changes in respiration to that of CO₂ in the chamber and $\Delta F/F$ (Fig. 2 and 3). We found that “the rise in RR begins at about 5s (approximate time that it takes for CO₂ levels to rise in the chamber and then reach the circulation)”. This rise in RR is preceded by the rise in the $\Delta F/F$. This is now confirmed by the cross-correlation analysis as well (SFig 5).*

5. Lines 298-310 (and 334-335): the authors report “extensive” axonal projections from LPBN FoxP2 neurons to the medulla. In contrast, two papers they cited used the same strain of Foxp2 mice (and a different strain of CGRP mice) and reported that FoxP2 projections to the medulla was minimal and that CGRP neurons in LPBN project more widely in the medulla than FoxP2 neurons. How do the authors explain these discrepancies between their new work and the previous published anatomic studies? The retrograde tracing images shown in this study, like previous results in rodents (including the senior author’s previous results in rats) give the impression of very few LPBN neurons (in contrast to KF) projecting axons to respiratory nuclei in the medulla.

Response: *The reviewer raises an interesting question. We chose not to address this issue in the paper, because the answer is a bit complicated and beyond the scope of this manuscript. But we will answer it here, and if the editor wishes, we can add this discussion to the paper. For FoxP2 neurons in the PB, as we point out above there are several other FoxP2 populations in the dorsal part of the PB. The injections by Geerling and colleagues (Huang et al., JCN 2021 529:657) were targeted toward the more dorsal and rostral temperature sensitive neurons that he was studying. These neurons project primarily to the hypothalamus. They did include some of the PBcl neurons, though, and they did find the projection to the ventrolateral medulla. It is difficult to compare the intensity of staining across experiments, as this depends upon the number of neurons that were labeled that contribute to a projection. As we found in our study (Fig 5), if the injection targets the PBcl FoxP2 neurons, you see quite a robust projection to the*

medullary respiratory areas (in fact, the reviewer, below (last minor point) complained that the projection was so heavily labeled that they could not make out the details of fibers and terminals, causing us to have to reduce the brightness and contrast of the images). We now report in the Results section that over 70% of neurons in the PB and KF that were retrogradely labeled from the pre-Botzinger complex were also FoxP2 positive.

With respect to the CGRP neurons, in our studies we used a Calca-Ires-Cre-ER line, which produces moderate amounts of Cre recombinase. When we inject those mice with anterograde tracer vectors, we see robust forebrain projections, but little if any projection to the medulla. Geerling (Huang et al JCN 2021 529:2911) used Richard Palmiter's Calca-Cre knockin mouse, which has one chromosome with no Calca gene (it is replaced by Cre). This hemizygote animal for Calca has to increase the drive to its one remaining Calca gene to get sufficient CGRP protein expression. This then drives the other allele, which now produces profuse amounts of Cre, instead of Calca. When they injected those mice with anterograde tracer vectors, they did see the medullary projection (which was relatively modest compared to the forebrain projection). The reason for that disparity became apparent when we did our spatial transcriptomic atlas of the PB region. We found that there are two populations of CGRP neurons in the PB region. The large PBel CGRP neurons express Calca mRNA at a high level, and project to the forebrain. A second population of smaller Calca cells turned up in the KF and these express much lower levels of Calca (as well as FoxP2), and project to the medulla. The small, low-expressor neurons were not labeled by Cre dependent vectors in our CGRP-ires-Cre-ER animals.

6. Line 614-615: how can a primary antibody produce immunostaining if it is omitted? Are the authors instead referring to the secondaries (not mentioned in the preceding lines)?

Response: We have now rephrased this.

7. Line 740: "A probability of error of less than 0.05 was considered significant" – inaccurate description of p/alpha; rephrase (alpha vs. beta error).

Response: We have now rephrased this.

Minor

- Lines 40-41 "second group of non-CGRP neurons" is confusing (also line 109). Grammatically, it implies that there was already first group of non-CGRP neurons (after the CGRP neurons). There are of course several groups of non-CGRP neurons in the LPBN, but after reading the paper it became apparent that the authors meant that the CGRP neurons were the first group and they are referring to "a second group of LPBN neurons" as one that lacks CGRP. I recommend re-phrasing to eliminate ambiguity.

Response: We have now rephrased this.

- Lines 45-46 "which has recently been found in this region." This is not a recent finding –it has been 15 years since Foxp2 expression was shown in the LPBN region ("Conservation and diversity of Foxp2 expression in muroid rodents: Functional implications" Campbell et al. J Comp Neurol 2008; also the Gray 2008 review cited by the authors), with many other papers using this LPBN marker in the interim.

Response: We have now rephrased this.

- Reference 22 appears to be a conference abstract with contents redundant to the author's published work. Remove. (reference 18 also appears to be a conference abstract rather than a published, peer-reviewed reference; remove)

Response: We have removed this.

- Line 159 “FoxP2-Cre::L10” is a non-standard way of describing a Cre-reporter strain, but whatever abbreviation is used, I recommend keeping it consistent throughout the manuscript to avoid confusing readers. In at least two other spots, different terms were used: line 150 (“FoxP2-L10”) and line 1095 (“FoxP2-cre-L10”).

Response: *We have corrected this.*

- Typo Line 218 “movvement”

Response: *We have corrected this.*

- More specific criteria (proximity/distance) are needed (preferably with supplementary plots and example images), showing optic fiber targeting in ChR2 and ArchT experiments relative to transduced FoxP2 neurons. For example (lines 268-270), it would be helpful to see examples of successful vs. missed targeting that did or did not produce the reported effects. Most importantly, the reader should understand why some cases were excluded and others were kept. Was the post hoc decision to exclude a case made after reviewing the experimental results, or by someone blinded to the analysis?

Response: *We have now included this as a supplementary figure (SFig 6a) which shows the transfection area of ChR2 and the site of optical fibers in relation to transfection both in off target (description in result section) and in on-target cases. Fig 7 also now includes a control group of wildtype cases injected with ArchT. In all cases, the histology was reviewed by an investigator who was blinded to the physiological result, and cases with off-target placement of optical fibers or injections were considered to be anatomical controls.*

- Lines 314-315: “ventral part of the PB” – it would be better to provide supplementary plots of these injection sites so the reader knows the exact distribution of neurons that were transduced in each case.

Response: *We have now included this as a supplementary figure (SFig. 6a) which shows the transfection area of ChR2 and the site of optical fibers are plotted in relation to the transfection in both the on and off target cases (description in result section).*

- Line 351: “intersecting” is unnecessary here; also, consider adding “neurons” after “the PB-FoxP2”

Response: *We have corrected this.*

- Line 460 & 463: “wildtype littermates” – C57 x Cre littermates are not “wildtype”

Response: *We have rephrased this.*

- Line 470 typo “Fox-P2”

Response: *We have corrected this.*

- Line 509: typo “immune-staining”

Response: *We have corrected this.*

- Line 514: “using square grids” – unclear; does this mean that images were tiled together?

Response: *We have now corrected this description.*

- Line 1086: “to the phrenic motor [neurons?]” – there is no evidence in this study (or in any others that I could find) that FoxP2 LPBN or KF neurons project axons to the cervical spinal cord. It may be the case, but the authors should make it clear that this is speculative (or provide evidence).

Response: *We have rephrased this.*

- Line 1094-1101: the authors labeled Calca (mRNA transcript) yet referred to it as CGRP (peptide). This is confusing and should be simplified for both accuracy and clarity. Also, they used a Cre-reporter strategy (tags developmental+adult expression) yet referred to it as "FoxP2." They should co-label the mRNAs (Calca, Foxp2) and/or protein products (CGRP, FoxP2) as above, and should be more careful with terminology so that the reader can tell what was and wasn't labeled.

Response: *We have gone back through the text to ensure that we use Calca when referring to the gene or mRNA and CGRP when referring to its peptide gene product. As we indicate above, we have done the studies to demonstrate that essentially all of the neurons in our FoxP2-Cre mice that are labeled by either L10-GFP, a developmental Cre-dependent reporter (SFig. 2) or by injection of a Cre-dependent vector (Figs. 2,4, and 5) in neurons that also express FoxP2 as shown by immunohistochemistry for the gene product.*

- Line 1117: as above, there was no Supplementary Figure 3 to evaluate.

Response: *We apologize for that. If the referee cannot find any of our figures or needs higher resolution versions, please contact the Editor, and we will provide them.*

- Figure1: bar graphs should be replaced by showing the individual data points.

Response: *We have provided both the dot plot and bars in this version.*

- Figures 4&6: replace the white bars, which obscure the underlying images with thin or dashed lines that indicate possible locations of optic fiber tracts.

Response: *We have now done this.*

- Figure 5: axonal labeling in panels b&c is highly over-contrasted, such that many axons look like blobs the diameter of cell bodies. Images should be replaced with less severe contrast/brightness settings.

Response: *We adjusted the contrast to better visualize individual fibers, and now also include images showing the magnified parts of the areas shown in b and c (Fig 5) to clarify that ChR2 in the projection sites only labels axon fibers and terminals and not cell bodies.*

Reviewer #3:

Summary of key results:

Kaur et al., perform a series of technically challenging experiments to investigate the role of FoxP2 expressing parabrachial neurons for respiration and arousal from sleep. Based on these experiments, and ones they have performed previously on neighboring CGRP neurons, they conclude that FoxP2 neurons are implicated in the respiratory response to CO₂ but not arousal, whereas CGRP neurons drive arousal, but not changes in respiration. For the most part, the data are compelling, but the strength of the conclusions could be significantly enhanced in a revised version with additional analysis and clarification.

Comments and suggestions:

Major:

1. Directly comparing the effects of CGRP and FoxP2 populations on breathing and arousal would provide much clearer insight into how functionally distinct these populations are. FoxP2 neurons appear most active in wake and REM states. However, data showing how activation/inhibition of these neurons may affect breathing in wake/REM are not provided. Thus, it is unclear whether these neurons generally increase breathing (across states) or if they do indeed play a specific role in the respiratory response to CO₂ during NREM. Further, the fiber photometry data (Supplementary Fig 3a), suggest that FoxP2 neuron activity may be correlated

with specific types of breathing in the awake state, but no analysis is performed. This would be interesting compare with has been shown for other PB neuron populations (e.g. PMID: 34921781; PMID: 36810601).

It is also suggested that the activity of FoxP2 neurons correlates with movement (EMG), but no analysis is performed to confirm this. Cross-correlation analysis (or something similar) of FoxP2 activity vs EMG, and FoxP2 activity vs respiratory rate, would be beneficial.

Response: *The reviewer raises several good points. With respect to comparison of the role of FoxP2 neurons with CGRP neurons in the PB region, we published our results with the CGRP neurons in 2017, and so it would be duplicative to republish them here. Thus comparisons are done in the Discussion, by comparing our two datasets.*

With respect to the request to examine the effects of photo-stimulation of the PBcl^{FoxP2} neurons during wake, that is now provided in SFig. 7a-f. Stimulation had very little if any effect during wake, when these neurons are apparently already maximally active.

With respect to photo-inhibition, we now indicate in the Results that “photoinhibition had no significant effects on respiration during either wake or REM sleep states.” (last sentence, third paragraph) In other words the PBcl^{FoxP2} neurons are part of the brainstem circuit that increases respiration in response to elevated CO₂ during NREM sleep, but that during wake (or REM) forebrain circuits essentially override this homeostatic brainstem system to provide ventilation consistent with behavior (or in REM, perhaps imagined behavior).

We thank the referee for suggesting cross-correlation analysis, which is now included in S Figs. 4 and 5. These show that activation of the FoxP2 neurons follows EEG arousal and the associated EMG activity by about 1 sec, indicating that these neurons are activated by EEG arousal, rather than causing it. By contrast, activation of PBcl^{FoxP2} neurons occurs about 0.5 sec prior to increased RR.

2. The authors show CGRP immunohistochemistry in FoxP2 Cre mice to quantify overlap in the KF but not PB. Showing similar data for the PBcl/el to confirm separation of CGRP and FoxP2 neurons, i.e. that the neurons in question are indeed non-CGRP, would help support the conclusions.

Response: *Huang et al., 2021 (PMID-33715169) clearly showed that “CGRP and FoxP2 immunoreactivity identified mutually exclusive populations of neurons” (their p. 2917). We have also confirmed this observation in our experiments, and in our new spatial transcriptomic atlas, we found FoxP2 expression only by the small neurons in the KF that express low levels of Calca mRNA. We have added this to the first paragraph of the results.*

3. It is clearly shown that FoxP2 neurons have projections to the preBötC area, but the functional role of these projections is not tested. This could provide important insight into the authors speculation that feedback from preBötC or increased respiratory effort, rather than the ascending projections of FoxP2 neurons, is what leads to arousal during activation of FoxP2 PB neurons.

Response: *We agree that the role of the different medullary projections of the FoxP2 neurons is indeed an interesting problem. However, it would require a massive study, using activation and inhibition of the PB-KF^{FoxP2} terminals in at least four regions of the medulla, in wake and sleep states, with and without CO₂, to work this out. This would take at least two years to accomplish, and is far beyond the scope of the current study.*

4. The FoxP2 neuron activation and inhibition experiments are not complimentary – activation drives RR without much TV effect, whereas inhibition prevents TV increase but not changes in RR. A potential explanation for this is provided, suggesting that RR can't be reduced because the CO₂ stimulus activates multiple other areas that drive RR. This is plausible, but if this is the case, activation and inhibition of FoxP2 neurons in the absence of CO₂ should have

complimentary (opposite) effects on RR and TV. However, this analysis is not shown. Moreover, based on the inhibition experiments (Fig 6g) it is concluded that FoxP2 neurons are necessary for increased Tv in response to CO₂. But, activation of these neurons does not increase Tv (Fig. 4f). Thus, since FoxP2 neurons do not appear to be sufficient to increase Tv, it is unclear how they can be necessary.

Response: *We now provide an extensive section on this issue in the Discussion (next to last paragraph). The difference between the photo-stimulation and photo-inhibition experiments is that the stimulation was done for 5 or 10 sec while the animal was breathing room air and the inhibition was done during a 30 sec interval while the animal was breathing 10% CO₂. The increase in RR occurred almost immediately during photo-stimulation (Fig. 4). During CO₂ exposure, the increase in RR is also very rapid, but the increase in Vt lags by 5-10 sec (Fig. 2), which may be why we did not see a significant rise in Vt during the 5-10 sec stimulation periods. On the other hand, when the animal is breathing 10% CO₂, the remaining respiratory circuitry is capable of increasing RR to near maximal levels without the PB^{FoxP2} neurons, but the latter are required to increase Vt (Fig. 6). Thus, under different conditions, the PB^{FoxP2} neurons contribute both to the RR and Vt response to CO₂.*

Minor:

Line 109: There is some evidence that CGRP neurons actually inhibit respiration, at least in the awake state (PMID: 32856589; PMID: 36810601), which may be worth clarifying here.

Response: *It is hard to know what to do with these observations based on opto-stimulation studies, as CGRP neurons cause behavioral arousal, and behavior has enormous secondary effects on respiration. That is why we focused on the LACK of an effect on CO₂-induced respiratory response when the CGRP neurons are inhibited (i.e., they are not needed for that response). Nevertheless, we have now included these references.*

Line 148: It isn't definitively shown with this data that they (FoxP2 neurons) are non CGRP.

Response: *Please see our answer to this question under Major point 2 above.*

Line 150: FoxP2-L10 mice should be defined here since this is the first mention of the FoxP2-cre strain.

Response: *We have now defined them in this line.*

Line 162: "All of the neurons transfected by an AAV that expressed Cre-dependent GCaMP6s (which also fluoresces green) also showed Foxp2 expression in their nuclei". This is only shown for the L10 reporter mouse, not the AAV experiments, so it is unclear to what extent the viral expression remained specific to FoxP2 neurons.

Response: *Please see the answer to Minor critique #1 by the Second Reviewer. We show in our Figs. 2, 4b, 5a, 7c, and S2 that the Cre expression in our FoxP2-Cre mice (as demonstrated by Cre-dependent expression of GCaMP, ChR2-mCherry, ArchT-GFP, and L10-GFP, respectively) is virtually 100% eutopic with the immunohistochemical staining for FoxP2 protein.*

Fig 2b: zoomed image isn't taken from a region that will contribute to the majority of Delta F/F signal.

Response: *We have now corrected this.*

Line 172-174: "The increased activity of PBFoxP2 neurons when emerging from NREM sleep may contribute to the sudden increase in respiratory effort when an animal exposed to CO₂ awakens from NREM sleep." This should be more clearly illustrated/analyzed. Supplementary fig 3b shows only a single example of this and it is unclear whether there is an associated change in RR. Additional analysis to better understand relationships between FoxP2 activity

($\Delta F/F$) state transitions (NREM/REM/wake) and respiratory rate, would be important to support this speculation.

Response: *We agree and thank the reviewer for this suggestion. S Figs. 4 and 5 now show cross correlation analysis of $\Delta F/F$ for the $PBcl^{FoxP2}$ neurons with EEG and EMG associated with Wake, NREM, REM sleep and increase in RR.*

Line 175: It would be helpful to clarify here that these are only trials when the mouse was sleeping. Also as suggested above, it would be informative to show whether or not CO₂ causes a similar activation of these neurons in the awake state. E.g. repeat Fig 2e when mice are in wake state and compare vs sleep.

Response: *We have now provided the cross-correlation analysis of neuronal activity with EEG (EEG γ - representing arousal) and EMG (representing wake state) in the supplementary figures (S Fig 4 and 5).*

Line 187-192: It's unclear how the last 5 breaths etc. aligns with the points on the graphs in Fig 2f-i, which appear to be time bins. Please clarify.

Response: *Fig. 2 f-i shows the responses of neurons during the first 15 sec after onset of CO₂, when the animals were still asleep. The last 5 breaths before awakening therefore were after the interval shown in Fig. 2f-i. A typical example of this is shown in Fig. 2d where the animal awakens about 23 sec after onset of CO₂. We have now clarified this in the text.*

Line 209-213: Cross correlation analysis to better understand the relationship between EMG and FoxP2 neuron activity in awake state? This relationship is not obvious from the data shown in Fig S3f.

Response: *We agree and thank the reviewer for this suggestion. S Figs. 4 and 5 now show cross correlation analysis of $\Delta F/F$ for the $PBcl^{FoxP2}$ neurons with EEG and EMG associated with arousal and increase in RR. Note that onset of EEG arousal leads the increase in $\Delta F/F$ while increase in RR during NREM sleep follows activation of the $PBcl^{FoxP2}$ neurons. EMG activation which accompanies arousal also leads $\Delta F/F$ except during REM sleep, when there is no EMG signal.*

Line 217-218: "PB Foxp2 neurons are most active during wake and REM sleep, and particularly during active movement in wake, and may anticipate the need for increased ventilation during active movement." Too much speculation for Results section without sufficient analysis performed to test these ideas. I.e. cross correlation of EMG and FoxP2 activity to see if FoxP2 activation precedes movements as suggested.

Response: *See response above.*

Fig 3e: Averaged or summed $\Delta F/F$ should be shown. Although it is stated that similar oscillations weren't present in NREM (Line 245-248) this isn't apparent from the data shown. Additional analysis could help strengthen this point. And again, do these neurons also show responses to CO₂ when mice are awake? Was the activity of these neurons recorded during normally behaving conditions in awake mice? Do single neurons show activity profiles similar to those observed in the fiber photometry experiments (i.g. Supplementary Fig 3A)

Response: *All of the GCaMP recordings were done on freely behaving mice, who were exposed to repeated 30 sec bouts of 10% CO₂ separated by 5 min intervals with room air. Summed $\Delta F/F$ during the first 15 sec of CO₂ exposure is shown for the entire population recorded by fiber photometry in Fig. 2. The activity of 28 individual neurons across a series of CO₂ exposures is shown in Fig. 3g. While it is possible to graph the sum of the activities of the neurons shown in panel 3g, we are not sure of the value of this. Unlike panel 2f, which shows the population response of a large number of PB^{FoxP2} neurons, endomicroscopy experiments*

show the activity of only a handful of individual cells which happen to be near the front of the GRIN lens. The waves of activation seen in the neurons in panel 3g is quite different from the population response in panel 2e, which in turn is different from the summed responses over 59 trials in fig. 2f. This suggests that the activity of PB^{FoxP2} neurons is variegated, but that the overall activity of the population across trials is closely correlated with but slightly leads the increase seen in RR and Vt with CO₂ stimulation. To give a better picture of the variability of individual neuron GCaMP fluorescence, we show neuronal activity profiles of individual neurons as videos during both REM sleep, wake and with exposure to hypercapnia. In addition, activity from multiple neurons is shown during different states (wake, NREM and REM sleep) in supplementary Fig.3c-f. During wake, when animals are exposed to hypercapnia, the neuronal activity profile mainly correlates with their movement (EMG) which also coincides with increase in RR (SFig 5) as seen with cross correlation analysis.

Line 263-265: What about trials when mice were in wake state or REM? It would be interesting to know how FoxP2 neurons may affect breathing in these conditions as well.

Response: We show the effect of photo-stimulation during wake on respiration in SFig 7a-f and discuss the effect of photo-inhibition of PB^{FoxP2} on respiration during wake and REM sleep (Result section, last line of next to last paragraph).

Line 267: FoxP2 activation causes arousal from sleep – it's unclear how different this is from the effect of CGRP neurons?

Response: This issue is now clarified in Paragraph 1 of Discussion. In brief, activation of PB^{FoxP2} neurons does not wake the animal up at all in 30% of trials, and in the remaining trials the animal only awakens near the end of the stimulation, or after it has finished. By comparison stimulation of the PB^{CGRP} neurons awakens animals in all trials in less than 2 sec. Inhibiting or ablating the PB^{FoxP2} neurons has no effect on arousal from NREM sleep caused by CO₂, whereas ablating or inhibiting the PB^{CGRP} neurons almost completely prevents arousal from NREM sleep caused by CO₂.

Line 268-270: A supplemental Fig to show/quantify the included/excluded fiber placement locations would be helpful.

Response: We have added SFig 6a in the revised version detailing this information.

Line 274-283: It would be helpful to quantify instantaneous RR vs time during 5 and 10 sec stimulations to support the point the authors are trying to make (i.e. similar to Fig. 2h-i). 5 breath "bins" does not provide much resolution. Line 289-292: Similar to the above, if there really is a different time course of the respiratory response in 2% CO₂, this should be illustrated in a graph similar to Fig. 2h-i. As currently plotted the response to 20Hz, 10s stim looks identical in normocapnia vs. 2% CO₂ (Compare Fig 4 f (blue) and 4g (purple)).

Response: We have included SFig 6b and c to provide this data.

Line 297: Activation of these projections to the preBötC area would be a nice addition to demonstrate whether or not this is the relevant pathway.

Response: This is the same question as Major Critique 3. We agree that the role of the different medullary projections of the FoxP2 neurons is indeed an interesting problem. However, it would require a massive study, using activation and inhibition of the $PB-KF^{FoxP2}$ terminals in at least four regions of the medulla, in wake and sleep states, with and without CO₂, to work this out. This would take at least two years to accomplish, and is far beyond the scope of the current study.

Line 299: preBötC or PBC is more conventional notation for preBötzinger Complex than PBZ.

Response: We wished to avoid confusion with parabrachial complex (also sometimes abbreviated PBC) in our manuscript. PreBötC is much longer (and requires inserting an umlaut over the “o”), which takes additional steps on an English based word processor. In English, most scientists call this the “preBotz”. Hence, we chose the use of PBZ, which is distinctive and short and mimics the way the term is most commonly pronounced by working scientists in English.

Line 321: How does inhibition of FoxP2 neurons affect breathing in awake and REM?

Response: This is the same as the question asked about Lines 263-5. We show the effect of photo-stimulation during wake on respiration in SFig 7a-f and discuss the effect of photo-inhibition of PB^{FoxP2} on respiration during wake and REM sleep (Result section, last line of next to last paragraph).

Line 326: Reference Fig 6i as well.

Response: We have included that.

Fig 6 c2,c5: Scale bars seem mislabeled?

Response: We have now corrected this.

Line 331-333: The specific role of response to CO₂ during sleep is not demonstrated because data testing the role of FoxP2 neurons in the awake state is not provided. Thus, these neurons may have a similar role in sleep and wake states.

Response: This is the fourth time the referee has asked this same question. We show the effect of photo-stimulation during wake on respiration in SFig 7a-f and discuss the effect of photo-inhibition of PB^{FoxP2} on respiration during wake and REM sleep (Result section, last line of next to last paragraph).

Line 333-335: could this reflect the “core/shell” configuration of PB Oprm1 neurons as suggested by PMID: 34921781

Response: Virtually all of the $PBeI^{CGRP}$ and $PBcl^{FoxP2}$ neurons express *Oprm1*. So, they are all in what Liu et al. call the “shell”, i.e., an area that expresses *Oprm1* and projects to the PBZ region. We found that more than 70% of the neurons we retrogradely labeled from the PBZ region in both the *PBcl* and *KF* also express *FoxP2*. But they do not really form a “shell”; rather they wrap around the dorsal and lateral margins of the *PBeI*. It is possible that the *Oprm1* neurons that are ventromedial to the $PBeI^{CGRP}$ cluster and project to the medulla may belong to a different group and have a different function. Our recent spatial transcriptomic analysis of the PB region indicates that there are a very large number of different cell types that express *Oprm1*.

Line 412: Photostimulation of inhibitory preBotC neurons generally slows RR (e.g. PMID: 25643296; PMID: 33670653; PMID: 33863785) unless done specifically during the inspiratory phase (PMID: 29483589) or under specific conditions (ref 38). The other reference provided (39) describes activation of excitatory neurons, which do consistently increase RR.

Response: We agree with reviewer that PBZ is a complex area which needs to be very selectively targeted for understanding their responses.

Other general comments:

It would strengthen the paper to be clear about whether FoxP2 neurons in *PBcl*, *KF*, or both were primarily targeted during optogenetic imaging and manipulation experiments, since these groups of neurons could have distinct functional roles.

Response: We have revised the paper to indicate that the experiments requiring an optical fiber, for technical reasons, were all done by placing the optical fiber above the $PBcl^{FoxP2}$ cluster. The KF^{FoxP2} cluster is more rostral and ventral, and placing an optical fiber specifically above these neurons would destroy most of the $PBcl$ and $PBel$, which would confound the experiments. Our new data with genetic ablation includes both the $PBcl$ and KF groups, and this is compared with opto-inhibition that largely targeted only the $PBel^{FoxP2}$ neurons.

In general, the figures could be improved by making the size and font of text consistent (some axis labels are very small, some are large, some text is bold, some not, etc.). There is also a lot of color used which can get somewhat confusing, and it is not always consistent. E.g. CO_2 trace is green and $\Delta F/F$ is red in Fig 2d but CO_2 is orange and $\Delta F/F$ is black in Fig 2g.

Response: We agree and have now improved the figures to make them clearer.

Parabrachial anatomy could be more clearly/consistently referenced in the text. E.g. PBL vs LPB vs $PBel/PBcl$

Response: We have now made these changes.

Response to reviewers:

We thank the reviewers for their careful reading of our manuscript. Responses below are in RED.

Reviewer #1 (Remarks to the Author):

The authors addressed my concerns. They added proposed mechanisms for FoxP2 neurons to receive chemosensory information and they improved the display of quantitative information in their figures. I still object to using standard error in place of standard deviation, but since the authors include the original data points, I will leave it to the editors to decide whether the changes are acceptable. In most circumstances I would advocate for showing the original data points (unless the data set has 100's or 1000's of points), using SD to quantify variability, and then plotting effect size (difference in means) with 95% confidence intervals. In this context, adding the original data points to the figures is an important change; editors can decide if that is enough. Overall the paper is well done experimentally and advances understanding of the Parabrachial / Kolliker-Fuse regions that are important for respiratory neural control.

Response: The reviewers comments are well taken. There are a number of ways to display statistical differences, and we are now providing the dot-plots plus the SEM, which is the $SD/(\text{square root of } n)$. We find this useful because the 95% confidence interval for each observed mean is $2 \times \text{SEM}$, and we prefer p values for the differences in mean, because these indicate the likelihood of obtaining this difference due to random variation rather than systematic differences between the two means. Most recent experimental studies published in Nature Communications seem to use this approach, although we do understand that many human studies with large n's use the method that the reviewer prefers. We are willing to display our statistics in whatever way the editors require.

Reviewer #2 (Remarks to the Author):

The authors' revisions addressed most of my initial critiques. They improved their manuscript by adding a DTA ablation experiment and helpful anatomical details, but I must admit some difficulty in understanding quite how the different pieces of the story fit together. After CO₂ exposure, there was a rise in PB-Foxp2 calcium activity at the time of increased RR, and stimulating PB-Foxp2 neurons increased RR (with minimal change in V_t), yet inhibiting or ablating the same neurons did not alter RR but did reduce V_t and MV (after CO₂ exposure). I get that these neurons may be sufficient to increase RR but not necessary for that same effect after CO₂ exposure, but it is difficult to see how a group of neurons with little to no effect on V_t (when they are photostimulated) would nonetheless be necessary for the majority of an increase in V_t after CO₂ exposure. None of that is to say that I don't think the results should be shared/published, and I do think the results will be of interest to investigators in several related fields.

Response: We appreciate this issue, and tried to address it in the previous version of our manuscript, in lines 323-325 in the Results and 479-481 in the Discussion. Our explanation for the lack of an increase in V_t with photostimulation of the PB-FoxP2 neurons was 1.) that the increase in V_t during CO₂ exposure takes >10 sec to develop (Fig. 2), and 2.) that the duration of the photostimulation was limited to 5 or 10 sec (Fig. 4f), so may not have been sufficient for a

change in V_t to develop. We have added a sentence to the text after line 481 (in the previous version), which is highlighted in the revised manuscript, to better explain this idea.

We also draw to the reviewer's attention that 5 sec of photostimulation failed to cause any change in mean V_t at all, but that by 10 sec the mean V_t during 10 sec of stimulation at 10 or 20 Hz was roughly 15% greater (Fig. 4). However, V_t had a larger variance than RR at all time points during both CO₂ exposure and photostimulation, so that the 15% increase in mean did not reach statistical significance. These data are consistent, though, with our hypothesis that the change in V_t may take longer to develop than the change in RR.

On the other hand, in the deletion experiment (Fig. 6), when the remaining respiratory cell groups can increase RR in response to elevated CO₂, the PB/KF-FoxP2 neurons turn out to be required for the elevation of V_t during CO₂ exposure. In these experiments, with exposure to 10% CO₂, the animals typically wake up between 17 and 20 sec into the exposure, and the pre-arousal V_t measurement is made on the last 5 breaths (about 1.5 sec) before the animal wakes up, so at least 15 sec into the CO₂ exposure. By this time, the V_t has already increased substantially. It goes up further after arousal, presumably reflecting forebrain input to the PB/KF-FoxP2 neurons. We hope that this is clearer now.

More importantly, from the abstract through the Discussion, the authors have made overly specific claims about the location(s) of neurons responsible for their results. Ideally, they would have a more precise marker than Foxp2 for their neurons of interest, which appear to be a small minority of neurons in the lateral PB that project axons to the medulla, rather than the forebrain. Separate publications have blamed different aspects of the lateral PB Foxp2 population for itch (Ren et al. Neuron 2023) and for immune-mediated HPA axis activation (Jagot et al. Neuron 2023), and the primary output targets of Foxp2-expressing neurons in the PB region are in the forebrain and midbrain. The present results are interesting, but the anatomical terms for locations targeted and neurons transduced should be provided with more nuance because this tiny brain contains a seemingly continuous population of Foxp2-expressing neurons, spanning several subnuclei (originally named and defined by the senior author of this paper) -- including central lateral, superior lateral, and dorsal lateral PB. Even the medial and external lateral subnuclei have a sprinkling of Foxp2 neurons. Immediately outside the PB, contiguous Foxp2 neurons fill neighboring nuclei, including the "caudal KF" (GABAergic neurons) as well as the KF and most of the NLL (glutamatergic neurons). The authors' AAV injections certainly transduced neurons across multiple PB subnuclei and surrounding nuclei, on one or both sides, in most cases. It is impossible that a stereotaxic injection at the volumes listed, in this tiny region of the mouse brainstem, could have remained confined to a single PB subnucleus like PBcl. Thus, the term PB"cl" is overly restrictive (leaving out, for example, the lateral crescent region), and the authors should refer to their region of neuronal transduction in photometry/imaging, stimulation, ablation, and inhibition experiments as "PBFoxp2" or "PB/KFFoxp2" (or perhaps "LPBN/KFFoxp2"), not PBcl/KFFoxp2.

Response: We appreciate this problem, and had originally called these PB-FoxP2 and KF-FoxP2 neurons in the first version of this manuscript. However, while there are many types of FoxP2 neurons in the lateral PB (see Huang et al. 2021), it is clear from Fig. 1 that the CO₂-responsive neurons in the lateral PB are limited to the most ventral cluster of FoxP2 neurons in the PBcl subnucleus (with a few in the lateral crescent). So, we changed the terminology in the first revision to PBcl^{FoxP2} neurons to reflect that the only cells that show cFos responses to CO₂

and which project to the ventrolateral medulla from lateral PB are in the PBcl group. However, the referee is technically correct that, although the PBcl-FoxP2 group is separated physiologically, connectionally, and spatially from the FoxP2 neurons located more dorsally in the PB (which were the ones studied by Ren et al and Jagot et al), the injections of viral vectors frequently involved the more dorsal parts of the lateral PB as well. We have therefore adopted the terminology suggested by the reviewer (PB^{FoxP2} neurons) for the optogenetic and lesion experiments, where we may have included more dorsally located FoxP2 neurons. This is now explained in the section of “Photo-activation of PB^{FoxP2} neurons...”, with the added sentences highlighted. However, in the fiber photometry experiments we only obtained responses to CO₂ when the optical fiber or endoscope was just above or in the PBcl, which is the only part of the lateral PB that showed cFos responses to CO₂. Hence in those experiments we have a high level of confidence that the results are obtained from PBcl-FoxP2 neurons.

Other suggestions (at the authors' discretion) which may improve the clarity of their manuscript for readers:

- Abstract: I recommend deleting the line “but has little if any effect on the respiratory response,” given that there is a robust respiratory response (associated with arousal). The authors may or may not be correct in guessing that the respiratory response is mechanistically downstream from the arousal response triggered by CGRP neurons in PBel, but it is not clear from their previous publication how this was (or could be) falsified in a way suggested by “little if any”.

Response. The sentence in question reads: “We previously found that inhibition of the glutamatergic neurons expressing calcitonin-gene related protein (CGRP) in the external lateral PB...prevents cortical arousal in response to hypercapnia, but has little if any effect on the respiratory responses.” We stand by that sentence. The referee is misinterpreting it as saying that the PBel-CGRP neurons have little or no effect on respiratory responses. That is not what the sentence says. We agree with the referee that the arousal responses caused by activating the PBel-CGRP neurons result in respiratory responses, but that is not what the sentence was about.

- Line 136: “neuron” should be plural

Response. Agreed. The change has been made.

- Line 153-157: It is difficult to understand how the authors find Calca mRNA co-localizing with a GFP Cre-reporter for Foxp2, and nearly 100% co-localization of FoxP2-ir with this GFP Cre-reporter, yet they cannot find co-localization of CGRP and FoxP2 directly. Instead of addressing this interesting dichotomy (and instead of showing direct co-localization between Foxp2 mRNA and Calca mRNA, or between Foxp2 mRNA and CGRP-ir, or between FoxP2-ir and Calca mRNA), they discussed spatial transcriptomic data from a separate (unpublished) paper. As in my initial critique, I recommend that the authors put their conclusion to the test by directly comparing the mRNA and protein products of each genetic marker -- and show the results.

Response. The FoxP2-ir is very strong, and it colocalizes 100% with the GFP reporter for FoxP2. The latter colocalizes with Calca mRNA in the small, Calca+ KF neurons. Hence these

neurons make both FoxP2 and Calca mRNA, although they do not stain for Calca immunoreactivity.

The reason for this disparity is likely due to the difference in the level of *Calca* mRNA (and hence, CGRP) expressed in the larger PBel-*Calca* neurons compared to the smaller, KF-*Calca* cells. The paper showing the differential expression of *Calca* between these two distinct cell types was cited as a preprint in the previous version, but has since been published in final form, which is now cited (Nardone S, De Luca R, Zito A, Klymko N, Nicoloutsopoulos D, Amsalem O, Brannigan C, Resch JM, Jacobs CL, Pant D, Veregge M, Srinivasan H, Grippo RM, Yang Z, Zeidel ML, Andermann ML, Harris KD, Tsai LT, Arrigoni E, Verstegen AMJ, Saper CB, Lowell BB. A spatially-resolved transcriptional atlas of the murine dorsal pons at single-cell resolution. *Nat Commun.* 2024 Mar 4;15(1):1966. doi: 10.1038/s41467-024-45907-7.) The reviewer is referred to a detailed discussion of this issue is on p. 5, left hand column, of that publication.

As an aside, many peptides are difficult to stain for immunohistochemically, as the peptides are rapidly exported to the axon terminals. Only cells with high levels of expression of a peptide are usually seen by immunohistochemistry, unless one dams up the peptide product in the cell bodies by either cutting the axon or causing disassembly of axonal microtubules with colchicine. For example, in 1985 we showed that CCK neurons in the superior lateral PB project to the VMH (Fulwiler and Saper, *Neurosci Lett* 53:289), by doing a combination of retrograde labeling and immunohistochemistry in colchicine treated animals. In non-colchicine animals, there was little immunostaining for CCK cell bodies in the PB. Zaborszky et al (*Brain Res* 303, 225, 1984) showed the same pathway by placing a lesion in the medial forebrain bundle that caused loss of CCK staining in the VMH, but by preventing transport of CCK down the axon, caused cell body staining in the PB that was not there in intact animals. Garfield et al. (*Cell Metab* 20:1030, 2014) later showed the same PB-CCK neurons, which are not seen by IHC in untreated animals, by using as CCK-Cre mouse crossed with a GFP reporter line. Another example is the presence of AVP and oxytocin in hypothalamic neurons projecting to the spinal cord. Without colchicine, Sawchenko and Swanson (1983) estimated that fewer than 5% of the spinally projecting neurons in the paraventricular nucleus expressed these peptides; with colchicine, Cechetto and Saper (1987) found about 25-35% of these neurons contained each peptide; with in situ hybridization, Hallbeck et al (1981) found that 40% expressed AVP mRNA.

The bottom line is that the large PBel CGRP neurons are exceptional for their immunohistochemical stainability without colchicine, but the small KF-*Calca* neurons are not stained under those conditions (at least in our laboratory or that of Huang et al). We do not think it is worthwhile to do more experiments to show what is widely known (and has repeatedly been shown, e.g., in the experiments cited above): that use of in situ hybridization (or Cre-reporter mice) is more sensitive than immunohistochemistry for identifying peptide expression by brain neurons.

- Line 286: waves of periodic activity, with peaks separated by 10+ seconds, make me think of gastrointestinal peristalsis driven by the MMC. In a resting mouse gut, the frequency of spontaneous GI contractions is in the neighborhood of a few per 30 seconds, and increases upon luminal distention. The lateral PB receives the bulk of its input from the caudal-medial NTS region that receives vagal input from the stomach and other parts of the upper GI tract. Pdyn neurons (most of which co-express Foxp2) in this part of the PB have been implicated in sensing gastric fullness (Kim et al *Nature* 2020), and this may be worth mentioning as an

alternative hypothesis for this unusual activity pattern, which does not look like it has much to do with breathing or arousal.

Response. The reviewer's thoughts on the possible origin of the responses of PBcl-FoxP2 neurons that have long periodicity are interesting. Kim et al were studying PB-*Pdyn* neurons that responded to various oro-pharyngeal and upper gi stimuli. Some of the PB-*Pdyn* neurons are in the PBcl and express FoxP2, although as Kim et al show (and Geerling showed in 2016, *AJPReg*), most are much more dorsally located and belong to the dorsal lateral FoxP2-*Pdyn* population. In addition to Kim et al. observing that some of the dorsal *Pdyn* neurons respond to oropharyngeal and upper gi stimuli, Geerling showed that some dorsal lateral *Pdyn* neurons respond to warming of the skin. It is likely that these constitute two different cell types, but studies to show that have not been done, and convergence of multiple types of stimuli on individual neurons, even of different cell types, is likely. For example, instilling water in the oropharynx in Kim's study may have activated the temperature responsive neurons that Geerling identified, depending upon the temperature of the water.

In our experiments, the GCaMP responses of the PBcl-FoxP2 neurons shown in Fig 3g are clearly CO2 responsive, because they had a sharp increase in their GCaMP fluorescence after onset of a CO2 stimulus. As we said in the previous version of our manuscript (lines 286-288), it is possible that some of the PBcl-FoxP2 neurons may receive other afferents which may give this periodicity to their GCaMP signal when they are activated. It is possible, as the reviewer indicates, that this could be due to convergence of inputs from oropharyngeal or upper gi vagal sources on the same neurons that respond to CO2. We have added this possibility to the Discussion at the end of the section on endomicroscopy of PBcl neurons.

- Lines 347-8: The authors claim to "map the projections of the PBclFoxP2 neurons expressing ChR2-mCherry" but then mention nothing of their (predominantly) rostral projections. Something should be added here to clarify this, at least in passing. Otherwise, non-expert readers would be forgiven for concluding, after reading this paragraph, that the authors' cases had axonal labeling exclusively in the lower brainstem.

Response. The reviewer makes a good point. We have now revised the text in the last paragraph of the section on "Photoactivation of PB^{FoxP2} neurons..." to clarify that this section is only about the PBcl^{FoxP2} projection to the medulla. The ChR2 injections involved the more dorsal lateral PB^{FoxP2} neurons as well as the PBcl^{FoxP2} cells. That was why, after finding medullary projections in those brains, we then did retrograde labeling studies to show that of the lateral PB^{FoxP2} neurons, only those in the PBcl project to the medulla.

- Line 368: delete "such as..." -- no evidence is provided specifically that PBelCGRP neurons survived. That is almost certainly true, but this is stated as if the authors actually labeled and quantified effects on that population, which was not shown or claimed.

Response. The referee is technically correct. The statement has been modified to say that "Immunostaining for mCherry showed that other neuronal populations, such as the PBel neurons,...are intact."

- Line 406: the modifiers "immediate" and "with a latency of less than 2s" seem to be fighting each other. Deleting the former would improve clarity.

Response. Agreed. We have made this change.

- Line 429: “insure” should be “ensure”

Response. Agreed. We have made this change.

Reviewer #3 (Remarks to the Author):

The authors have addressed my comments and have made important additions/revisions that have improved the paper. I have no further concerns.